# APFoam-1.0: integrated CFD simulation of $O_3$–$NO_x$–VOCs chemistry and pollutant dispersion in typical street canyon

Luolin Wu[1], Jian Hang[1], Xuemei Wang[2], Min Shao[2], and Cheng Gong[3]

[1]School of Atmospheric Sciences, Sun Yat-sen University, Guangzhou 510275, P. R. China
[2]Institute for Environmental and Climate Research, Jinan University, Guangzhou 510632, P. R. China
[3]China Aerodynamics Research and Development Center, Mianyang 621000, P. R. China

*Correspondence to*: Xuemei Wang (eciwxm@jnu.edu.cn) and Jian Hang (hangj3@mail.sysu.edu.cn)

**Abstract.** Urban air quality issue is closely related to the human health and economic development. In order to investigate the street-scale flow and air quality, this study developed the Atmospheric Photolysis calculation framework (APFoam-1.0), an open-source CFD code based on OpenFOAM, which can be used to examine the micro-scale reactive pollutant formation and dispersion in the urban area. The chemistry module of the new APFoam has been modified by adding five new types of reactions, which can implement the atmospheric photochemical mechanism (full $O_3$–$NO_x$–VOCs chemistry) coupling with CFD model. Additionally, the model including photochemical mechanism (CS07A), air flow and pollutant dispersion has been validated and shows the good agreement with SAPRC modeling and wind tunnel experimental data, indicating that the APFoam has sufficient ability to study urban turbulence and pollutant dispersion characteristics. By applying the APFoam, $O_3$–$NO_x$–VOCs formation processes and dispersion of the reactive pollutants were analyzed in an example of typical street canyon (aspect ratio $H/W$=1). The comparison of chemistry mechanism shows that $O_3$ and $NO_2$ are underestimated while NO is overestimated if the VOCs reactions are not considered in the simulation. Moreover, model sensitivity cases reveal that 82%–98% and 75%–90% of NO and $NO_2$ are related to the local vehicle emissions which are verified as the dominant contributors to local reactive pollutant concentration in contrast to their background conditions.

Besides, a large amount of $NO_x$ emission, especially NO, is beneficial to reduce the $O_3$ concentration since NO would consume $O_3$. Background precursors (NOx/VOCs) from boundary conditions only contribute 2%–16% and 12%–24% of NO and $NO_2$ concentrations and raise $O_3$ concentration by 5%–9%. Weaker ventilation conditions could lead to the accumulation of $NO_x$ and consequently a higher $NO_x$ concentration, but a lower $O_3$ concentration due to the stronger NO titration effect which would consume $O_3$. Furthermore, in order to reduce the reactive pollutant concentrations under the odd-even license plate policy (reduce 50% of the total vehicle emissions), vehicle VOCs emissions should be reduced by at least another 30% to effectively lower $O_3$, NO and $NO_2$ concentrations at the same time. These results indicate that the examination of the precursors (NOx/VOCs) from both traffic emissions and background boundaries is the key point for better understanding $O_3$–$NO_x$–VOCs chemistry mechanisms in street canyons and providing effective guidelines for the control of local street air pollution.

**Keyword.** CFD model; $O_3$–$NO_x$–VOCs chemistry; Urban pollution; Street canyon; OpenFOAM;

## 1 Introduction

With the worldwide rapid urbanization, air pollution in cities, such as haze and photochemical smog characterized by high-
levels of the particulate matter and/or surface ozone ($O_3$), has become one of the widely concerned environmental problems
(Lu et al., 2019; Wang et al., 2020). Recently, the observational data have shown that $PM_{2.5}$, one of the major pollutants in
cites, has decreased 30%–50% across China due to strict air quality control measures (Zhai et al., 2019). At the same time,
87%, 63%, 93%, 78% and 89% of the observational stations in China have shown the decreasing trend of CO, $NO_2$, $SO_2$, $PM_{10}$
and $PM_{2.5}$ in recent 5 years, respectively (Fan et al., 2020). Various data indicate that air quality in China has been significantly
improved. Unlike other pollutants, however, $O_3$ concentrations has increased in major urban clusters of China (Lu et al., 2018).
Severe $O_3$ pollution episodes still exist and happen frequently (Wang et al., 2017). Therefore, research on reactive pollutants
such as $O_3$, which has adverse effect on human health (Goodman et al., 2015; Liu et al., 2018b; Sousa et al., 2013), the
productivity of crop (Rai and Agrawal, 2012), building material (Massey, 1999) and vegetation (Yue et al., 2017), is of great
significance to the further improvement of air quality especially in urban area.

From the perspective of the cause of urban air pollution, traffic-related emissions are the major part of airborne pollutant
sources, including the precursors of $O_3$, $NO_x$ ($= NO + NO_2$) and volatile organic compounds (VOCs) (Degraeuwe et al., 2017;
Kangasniemi et al., 2019; Keyte et al., 2016; Pu and Yang, 2014; Wild et al., 2017; Wu et al., 2020). It has been believed that
the production of $O_3$ is from the $NO_2$ photolysis. Generally, in a clean atmosphere, the produced $O_3$ would be consumed by
NO titration effect. However, with the involved VOCs, NO concentrations become lower due to the consumption with $RO_2$
(the production of VOCs and OH, $VOCs + OH \rightarrow RO_2 + H_2O$), which weaken the NO titration effect and consequently lead
to the $O_3$ accumulation (Seinfeld and Pandis, 2016). In China, previous studies have shown that 22%–52% of total CO, 37%–
47% of total $NO_x$, and 24%–41% of total VOCs emissions are contributed by the vehicle emissions in urban area (Li et al.,
2017; Zhang et al., 2009; Zheng et al., 2014, 2009).

To investigate the formation pattern and dispersion of reactive pollutants, numerical simulation has been considered as
an effective method by using the air quality models. Based on length scales, the air flow and air quality modeling in cities are
commonly categorized into four groups by length scales, i.e., street scale (~100 m), neighborhood scale (~1 km), city scale
(~10 km), and regional scale (~100 km) (Britter and Hanna, 2003). With the complex geometric and non-uniformity in
building distribution within cities, Computational Fluid Dynamics (CFD) simulation has recently gained popularity in the
urban climate research (Toparlar et al., 2017). Different from the typical meso-scale (~1000km) and regional-scale (~100km)
air quality models, CFD has better performance in micro-scale pollutant dispersion within the urban street canyon (~100m) or
urban neighborhoods (~1 km), which are restricted spaces with more complicated turbulent mixing and poorer ventilation
condition than rural areas (Zhong et al., 2015). Besides the shorter physical processes in micro-scale urban models (~100m-
1km, ~10s-100s), the rather fast chemical processes of $NO_2$ photolysis and NO titration  with the complex chain of VOCs

reactions also require finer resolution model (Vardoulakis et al., 2003). For instance, CFD models with fine grid (~0.1-1m)

and small timestep (~0.1s) have been effectively adopted to simulate these high-resolution spatial and temporal variations in urban areas (Sanchez et al., 2016).

With the rapid growth of the high-performance computing (HPC) platforms, computational power is no longer an obstacle. CFD simulation shows the good application prospect in urban microclimate research (Fernandez et al., 2020; Garcia-Gasulla et al., 2020). Many CFD models coupled with photochemical reaction mechanism have been developed to investigate the

street-scale air quality problem in recent years (See Table 1). More commonly, simple photochemical mechanism with only three reactions (Leighton, 1961) is adapted in CFD models. This mechanism can simulate the $NO_x$-$O_3$ dispersion with a lower computational requirement. Many previous studies have investigated the pivotal factors that affect the reactive pollutant distribution within the street canyon by using CFD model with simple photochemical mechanism, such as street-building aspect ratio (He et al., 2017; Zhang et al., 2020; Zhong et al., 2015), ambient wind conditions (Baker et al., 2004; Merah and

Noureddine, 2019), thermal effects (Baik et al., 2007) or emissions from vehicle (Liu et al., 2018a; Zhang et al., 2019b). However, due to the simple photochemical mechanism ignoring the effect of other nitrogen oxides and VOCs on the photochemistry, some studies recently have applied the full photochemical mechanism in CFD models to reduce the uncertainty of pollutant simulation. Photochemical mechanisms contain $NO_x$-$O_3$-VOCs reactions and photochemistry, such as CBM-IV (Garmory et al., 2009; Kwak et al., 2013; Kwak and Baik, 2012, 2014), GEOS-Chem (Kim et al., 2012; Park et al.,

2016), RCS (Bright et al., 2013; Zhong et al., 2017), and CCM (Sanchez et al., 2016) are successfully coupled with CFD models and applied to analyse the street-scale pollutant dispersion.

Currently, most of the simulation studies have been carried out by the application of commercial CFD software. This software has rather simple operation which is effective to save time when setting up the simulation case. However, the commercial codes are usually closed source, which is a "black box" for users (Chatzimichailidis et al., 2019). In this case, the

adjustment to the equations and parameters or modifications to the model is difficult for some specific simulations. Therefore, an open-source CFD code for atmospheric photolysis calculation, APFoam-1.0, was developed in this study. Open Source Field Operation and Manipulation (OpenFOAM) was selected as the platform for the APFoam framework, as OpenFOAM has good performance in computing scalability and low uncertainty levels which shows good ability in large-scale CFD simulation with million-level grid number (Robertson et al., 2015). Additionally, the solvers in OpenFOAM for specific CFD problem

can be developed by using the appropriate packaged functionality, which simplifies the difficulty of programming. Furthermore, OpenFOAM has been also well-developed with various pre- and post-processing utilities which are convenient for data manipulation (OpenFOAM Foundation, 2018).

This paper is organized as follows: besides this Sect. 1 of introduction, Sect. 2 presents a full description of the new chemistry module and simulation solver. Model validation for the photochemical mechanism, turbulence simulation and

pollutants dispersion compared with the chemical box model and several wind tunnel experiments is discussed in Sect. 3. Subsequently, in Sect. 4, a series of sensitive case has been set up to investigate the contribution of the key factors on the

reactive pollutants in a typical street canyon (aspect ratio, building height/street width, *H/W*=1). Finally, the conclusion and future research plan for the APFoam framework are summarized in Sect. 5 and Sect. 6.

## 2 Model description

### 2.1 General overview

APFoam framework has been developed based on OpenFOAM which is an open-source code for CFD simulation. For the numerical solution, APFoam uses finite volume method (FVM) to discretize the governing equations and adopts arbitrary three-dimensional structured or unstructured meshes. All variables of the same cell are stored at the center of the control volume (CV), and complex geometries can be easily handled with FVM (Chauchat et al., 2017). In APFoam, Laminar, Reynolds-averaged Navier–Stokes equations (RANS) and Large eddy simulation (LES) method are available for turbulence solution. Additionally, APFoam also has complete boundary conditions to choose for numerical simulation.

Based on the OpenFOAM, APFoam has been developed to conduct the photochemical simulation within the atmosphere. Different from the general chemical reaction type, there are some new types of the gaseous reactions describing the photochemical processing. More details will be introduced in Sect. 2.2.

To make it easier to get started with APFoam, the structure of the simulation case folder is consistent with the OpenFOAM. Figure 1 shows the flow diagram of simulation set-up in APFoam. The solver of APFoam, APChemFoam for one-dimension (1D) chemistry solving is modified from the solver ChemFoam which will be introduced in detail in Sect. 2.3. Other three-dimension (3D) solvers are modified from the solver reactingFoam, including APreactingFoam for online solving coupled with flow and chemical reaction, APonlyChemReactingFoam for solving only chemical reaction with a certain flow field and APsteadyReactingFoam for online solving coupled with flow and chemical reaction in steady-state. More details will be presented in Sect. 2.4.

For the simulation running (see Figure 1), mesh files, configure files, initial and boundary condition files should be prepared before the simulation. Mesh files can be made by various ways, such as blockMesh application executable by using data from blockMeshDict or fluentMeshToFoam by converting the .msh file to OpenFOAM format. For APFoam simulation, all required configure files are also listed in Figure 1. Besides, user-defined function can be loaded at run-time without recompiling the program, via writing related configuration files in the system folder. As for initial conditions, the initial state of turbulence, environment (e.g. temperature, pressure) and chemical species are necessary for the simulation. The results of APFoam would contain the wind flow and pollutant concentrations, which can be processed by the Paraview in OpenFOAM or any other CFD post-processing tools.

## 2.2 Chemistry module

For photochemical calculation, there are five types of reactions in the $NO_x$-$O_3$-VOCs mechanism. In the original version of OpenFOAM, these types of reactions are not included and should be added to the chemistry module prior to simulation. These types of the reactions are described as followed:

(1) Arrhenius reactions:
Arrhenius reaction is the basic reaction in the mechanism, and the rate of the Arrhenius reaction is calculated as

$$k = A \cdot (\frac{T}{300})^B \cdot \exp\left(-\frac{E}{T}\right) \tag{1}$$

where $A$, $B$ and $E$ are the parameters of the reaction rates, and $T$ is the temperature of mixture in Kelvin.

(2) Photolysis reactions:
Photolysis reactions are first-order reactions, and the photolysis rate is calculated as

$$k_{phot} = \int_{\lambda_1}^{\lambda_2} J(\lambda) \cdot abs(\lambda) \cdot QY(\lambda) d\lambda \tag{2}$$

where $k_{phot}$ is the first order rate for the photolysis reaction; $\lambda_1$ and $\lambda_2$ are the photolysis wavelength ranges according to the specific species; $J(\lambda)$, $abs(\lambda)$ and $QY(\lambda)$ are the intensity of the light source, absorption cross section and the quantum yield for the reaction at wavelength $\lambda$, respectively.

Actually, the photolysis rate could be calculated by other photolysis rate model such as Fast-J (Wild et al., 2000), TUV (Madronich and Flocke, 1999), or obtained from photolysis data set such as IUPAC (Atkinson et al., 2003). Since the photolysis rate does not depend on temperature, in the current version of APFoam, the model does not consider the variation of light intensity, the photolysis rates are obtained from the literature (Carter, 2010) rather than online calculation in order to improve the calculation efficiency.

(3) Falloff reactions:
Rate of Falloff reactions is a function of temperature and pressure which is calculated as

$$k(T, M) = \left\{ \frac{k_0(T) \cdot [M]}{1 + \frac{k_0(T) \cdot [M]}{k_{inf}(T)}} \right\} \cdot F^z \tag{3}$$

where $z = \{1 + [\frac{\log_{10}\{\frac{k_0(T) \cdot [M]}{k_{inf}(T)}\}}{N}]^2\}^{-1}$; [M] is the concentration of third body, which depends on total pressure; $F$ is broadening factors. $k_0$ and $k_{inf}$ are the rates of in Arrhenius form at low-pressure limit and high-pressure limit, respectively.

(4) "Three $k$" reactions:
The rate of "Three $k$" reactions depends on three reaction rates in Arrhenius form. The rate is calculated as

$$k(T, M) = k_0(T) + k_3(T) \cdot [M] \cdot \left(1 + \frac{k_3(T) \cdot [M]}{k_2(T)}\right) \tag{4}$$

where $k_0$, $k_2$ and $k_3$ are three reaction rates and [M] is the concentration of third body.

(5) "Two $k$" reactions:
The rate of "Two $k$" reactions depends on two reaction rates in Arrhenius form. The rate is calculated as

$$k(T, M) = k_1(T) + k_2(T) \cdot [M] \tag{5}$$

where $k_1$ and $k_2$ are two reaction rates and [M] is the concentration of third body.

In the current version of the APFoam, two atmospheric photochemical mechanisms are included in the model, which was SAPRC07 (Carter, 2010) and CB05 (Yarwood et al., 2005). For SAPRC07, two versions of the chemical mechanism are available, which are CS07A and SAPRC07TB. CS07A is one of the condensed versions of the mechanism, which contains 52 species and 173 reactions. SAPRC07TB is a more complicated version and even contains toxics species, with 141 species and 436 reactions. As for CB05, basic version with 51 species and 156 reactions is optional in the model. In the section 3.1, CS07A has been validated while the other two mechanisms are not verified in this study but are still the available option to the users.

## 2.3 One-dimension chemical solver (APchemFoam)

In APFoam framework, one-dimension chemistry solver (i.e. chemistry box model) called APchemFoam is included in the model. This solver only concerns the chemical concentration and reaction heat variation during simulation, and calculations are started from initial conditions within a single cell mesh. The concentration and energy equation are described as (OpenFOAM Foundation, 2018):

$$\frac{\partial \rho Y_i}{\partial t} = k_i(Y_i, T) \tag{6}$$

$$h = u_0 + \frac{p}{\rho} + \int_0^t \frac{\dot{q}}{\rho} d\tau \tag{7}$$

$$h = \sum Y_i \left( \Delta h_{f,i}^0 + \int_{T_0}^T C_{p,i}(T') dT' \right) \tag{8}$$

$$p = \frac{\rho R T}{M_{ave}} = \sum p_i = \sum \frac{Y_i}{M_i} \rho R T \tag{9}$$

where $Y_i$ is the species mass fraction; $k_i$ is the reaction rate; $T$ is the temperature of the mixture; $h$ is the specific enthalpy; $u_0$ is the initial energy; $p$ is the pressure; $\rho$ is the density of the mixture; $\dot{q}$ is the heat from reaction; $\Delta h_{f,i}^0$ and $C_{p,i}$ are the enthalpy of formation at reference temperature $T_0$ and the constant-pressure specific heat (a function of temperature) of species $i$; $R$ is the gas constant and $M_{ave}$ is the average molar weight; $p_i$ and $M_i$ are partial pressure and the molar mass of species $i$. Besides, one of the $p$ and $\rho$ should be set as a constant for simulation according to the needs of research. The other is calculated by Eq. 9.

## 2.4 Three-dimensional (3D) CFD solver with photochemical reaction

As mentioned above, three 3D solvers for atmospheric photochemical CFD calculation, including APreactingFoam, APonlyChemReactingFoam and APSteadyReactingFoam, are developed in the APFoam framework.

For APreactingFoam, flow field, chemical reaction and pollutant dispersion are solved simultaneously in the same time step in this solver. Firstly, the continuity governing equation in this solver is:

$$\frac{\partial \rho}{\partial t} + \nabla \cdot (\rho U) = 0 \tag{10}$$

Besides, the momentum governing equation is:

$$\frac{\partial \rho U}{\partial t} + \nabla \cdot (\rho U U) - \nabla \cdot \tau = -\nabla p \tag{11}$$

Additionally, the energy governing equation is:

$$\frac{\partial \rho h}{\partial t} + \nabla \cdot (\rho U h) + \frac{\partial \rho K}{\partial t} + \nabla \cdot (\rho U K) - \nabla \cdot (\alpha_{eff} \nabla h) = \frac{\partial p}{\partial t} + \dot{q} \tag{12}$$

where $U$ is the velocity vector of the air flow; $\tau$ is the viscous stress tensor; $K$ is the specific kinetic energy; $\alpha_{\text{eff}}$ is the effective thermal diffusivity coefficient.

Pressure-velocity coupling schemes for solving the flow field is PIMPLE algorithm, a merged PISO–SIMPLE algorithm in OpenFOAM toolkit. This algorithm uses the steady-state solution (SIMPLE algorithm) for the flow field within the time step. When defined tolerance criterion is reached, this algorithm uses PISO algorithm in the outer correction loop and moves on in time (Holzmann, 2017). The PIMPLE algorithm allows the larger Courant numbers (Co >1) so that time step can be increased to reduce the computation time. Even so, the time step ($\Delta t$) generally follows the Courant–Friedrichs–Lewy (CFL) condition to maintain numerical stability, which is:

$$Co = \frac{U \Delta t}{\Delta x} \leq 1 \tag{13}$$

where $\Delta x$ is the grid size.

Besides, the governing equation for the reactive species transportation is:

$$\frac{\partial \rho Y_i}{\partial t} = -\nabla \cdot (\rho U Y_i) + \nabla \cdot (\mu \nabla Y_i) + [\Delta Y_i]_{chem} + E_i \tag{14}$$

where $\mu$ is the kinematic viscosity; $[\Delta Y_i]_{chem}$ is the concentration change of species $i$ from the chemical reaction; As mentioned above, $[\Delta Y_i]_{chem}$ is calculated following the equations of (6)–(9). $E_i$ is the emission source of species $i$. The chemistry is solved by the ordinary deferential equation (ODE) solvers in OpenFOAM library, in which the chemical reactions can be integrated by dividing the flow time step into serval sub-time steps, automatically. The APreactingFoam is only designed to solve compressible fluids because the simulation results are more likely to be unstable and divergent when the chemistry and flow field are solved simultaneously under the incompressible fluids. It is the limitation of OpenFOAM code which is widely known by its users.

APonlyChemReactingFoam is only capable of solving the chemical reaction and species dispersion in the same time step under a certain flow field. The solution of turbulent fluids governing equation is switch off. The purpose of developing this solver is to save the computation time and reduce repetitive simulation. In general, the atmospheric chemical reactions have negligible effect on the flow field. Therefore, when the example cases to be studied do not involve the flow field change, this

solver is suitable for this kind of simulation. The governing equation for the reactive species transportation is consistent with APreactingFoam (Eq. (14)).

APSteadyReactingFoam is developed for solving the chemical reaction and species dispersion under the steady-state flow field. This solver is only designed to solve compressible fluids for the same reason as APreactingFoam which is mentioned above. In this solver, pressure-velocity coupling scheme switches to SIMPLE algorithm for steady-state solution. The

220 continuity governing equation in this solver is:

$$\nabla \cdot \rho U = 0 \tag{15}$$

Besides, the momentum governing equation is:

$$\nabla \cdot (\rho U U) = -\nabla p + \nabla \cdot (\mu \nabla U) \tag{16}$$

Also, the energy governing equation is:

$$\nabla \cdot (\rho U h) + \nabla \cdot (\rho U K) - \nabla \cdot \left(\alpha_{eff} \nabla h\right) = \dot{q} \tag{17}$$

As for reactive species, the governing equation for the transportation still applies Eq. (14) as well, in order to ensure the stability of chemical reaction.

## 3 Model validation

### 3.1 Photochemical reaction mechanism

To verify the accuracy of chemical reaction solution and specie concentration calculation, APFoam results are compared with the results from SAPRC box modeling software (Carter, 2010). For the chemical mechanism, CS07A is selected for validation in this study, and simulation time is set as 24h without diurnal variation (i.e., chemical reaction rate is constant during simulation), allowing the reactants to fully react and verifying the stability of the model.

Figure 2 shows the concentrations of 52 species from two models at 24h which is the last time step of the simulation. In general, APFoam results have a good agreement with the SAPRC box model. Except that some species have large errors when the magnitude is very small (Figure 2f–h), the simulation results for other species from two models are basically consistent.

For further investigation, relative error (*RE*, %) for each specie at each time step and mean relative error (*MRE*, %) are calculated for the selected species with large bias. These statistics are calculated as followed:

$$RE_{i,\ t} = \frac{C_{APFoam,\ i,\ t} - C_{SAPRC,\ i,\ t}}{C_{SAPRC,\ i,\ t}} \times 100\% \tag{16}$$

$$MRE_i = \frac{\sum_{t=1}^{n} \left| \frac{C_{APFoam,\ i,\ t} - C_{SAPRC,\ i,\ t}}{C_{SAPRC,\ i,\ t}} \right|}{n} \times 100\% \tag{17}$$

where $RE_{i,\,t}$ is the relative error of specie $i$ at time step $t$; $C_{APFoam,\,i,\,t}$ is the concentrations of specie $i$ at time step $t$ from APFoam; $C_{SAPRC,\,i,\,t}$ is the concentrations of specie $i$ at time step $t$ from SAPRC box model; $MRE_{i,}$ is the mean relative error of specie $i$ and $n$ is the total number of the time step.

Overall, most of the $RE_{i,t}$ are less than 1% in the concentrations range between 0 to $10^{-20}$ ppmv (i.e., the concentrations under realistic conditions), indicating that simulation error of APFoam is less than 1% during the whole simulation period. However, there are 6 species with $RE$ and $MRE$ greater than 1%, which are TERP, ISOPRENE, OLE1, OLE2, IPRD and ARO2. The $MRE$ of these 6 species are 44.0%, 40.7%, 7.74%, 38.5%, 7.71% and 1.20%, respectively. Additionally, Figure 3 shows time series of $RE$ and concentrations for these high $RE$ and $MRE$ value species. In Figure 3a, $RE$ values in the early stage of simulation ($t$ = 0–180 min) are less than 1% for these species. However, the $REs$ of TERP, ISOPRENE and OLE2 increase dramatically after $t$ = 180 min. The $REs$ can even up to respective 190.4%, 297.0% and 867.4% in the following simulation. It should be noted that, at the later time, the $REs$ of these 3 species have no values because they are consumed up during the chemical reaction and their concentrations from SARPC box model become zero. The significant increase in the $RE$ values of OLE1 and IPRD begins at $t$ = 1020 min, with the maximum $RE$ values of 60.1% and 60.9%, respectively. Relatively, the $RE$ of ARO2 is smaller with the value of 5.0% only.

Figure 3b illustrates the concentrations variation of these 6 species with worst agreement from two models. It can be found that the concentrations of these 6 species keep dropping during the whole simulation period. Combined with the result of Figure 3a, the dramatical increase of $RE$ is due to the significant concentration decrease of these 6 species. In this study case, these 6 species are continuously consumed without complementation, which results in the concentrations of these species tending to be 0. For extremely small number, the processing of different model is diverse. Thus, when the reduction of magnitude excesses $10^{-5}$ ppbv to $10^{-6}$ ppbv, the $RE$ would become much larger between two models. In the realistic situation, the concentrations of the species would not be completely consumed up with continuous emission source and boundary conditions. Therefore, the photochemical reaction simulation results of APFoam could be reliable and the overall errors might be less than 1%.

**3.2 Numerical settings and validation study in urban flow modelling**

It is well known that large eddy simulations (LES) perform more accurately in simulating urban turbulent characteristics than the Reynolds-Averaged Navier-Stokes (RANS). However, RANS models (e.g., $k$-$\varepsilon$ models) are still more widely utilized because of the disadvantages of LES, such as the much more computational time and resource requirements, the difficulties in setting appropriate wall boundary conditions and defining the time-dependent domain inlet, the challenges to develop advanced sub-grid scale models. Among the RANS turbulence models, in contrast to the modified $k$-$\varepsilon$ models (e.g., realizable and RNG $k$-$\varepsilon$ models), although the standard $k$-$\varepsilon$ model performs worse in predicting turbulence in the strong wind region of urban districts (e.g., separate flows near building corner), the prediction accuracy is better in simulating the low-wind-speed region

(e.g. weak wind in 2D street canyon sheltered by buildings at both sides) (Tominaga and Stathopoulos, 2013; Yoshie et al., 2007). Hence, as one of the widely adopted RANS methods, the standard $k$-$\varepsilon$ model is selected to solve the incompressible steady-state turbulent flows in 2D street canyon.

To further evaluate the numerical accuracy of the turbulence flow simulation, a scaled CFD case is performed under the estimation of wind tunnel data. In the wind tunnel experiments (Figure 4a), 25 rows of building model in total are set along
the wind direction with the working section of 11 m long, 3 m wide and 1.5 m tall. For each row, building height ($H$), building width ($B$) and street width ($W$) are 12 cm, 5 cm and 5 cm (i.e., aspect ratio $H/W$=2.4), respectively. The span-wise (or lateral) length is $L$ = 1.25 m > 10$H$ which is sufficient long to ensure the 2D flow characteristics in the street canyon (Hang et al., 2020; Oke, 1988; Zhang et al., 2019a), i.e. the flow in the targeted street region is determined by the external flow above it but with little impacts from the lateral boundaries. Free flow wind speed in the wind tunnel experiment is 13 m s$^{-1}$.

Figure 4b and 4c show the schematic diagrams of the CFD simulation domain setting for a single full-scale street canyon simulation. $H$ and $W$ of the street canyon are set as 24 m and 10 m, with spatial scale ratio of 200:1 compared with the wind tunnel experiment. The corresponding Reynolds number ($Re = \frac{U_{ref}H}{v}$) in full-scale flow CFD validation ($H/W = 2.4$, $H = 24$m) is about $2.14 \times 10^7$, and that in wind-tunnel-scale experiments ($H/W = 2.4$, $H = 0.12$ m) is $1.9 \times 10^5$, which satisfy the requirement of Reynolds number independence (the critical is about $8.7 \times 10^4$ with the $H/W$ of 2) (Chew et al., 2018; Yang et
al., 2020, 2021). The normalized wind profiles with two scales can be compared for the validation purpose. Such validation technique has been adopted in the literature (Hang et al., 2020; Yang et al., 2021). Besides, the building width $B$ and $Ly$ in the CFD simulation are 10 m and 3.2 m ($2H/15$), respectively, assuming that only a section ($Ly$=$2H/15$) of long street canyons adopted with symmetry conditions is applied at two lateral boundaries. The minimum grid size in this case is 0.2 m with expansion ratio of 1.2 from the wall surface toward the surrounding, which refers to the grid independence tests from our
previous research (Zhang et al., 2019a). The upstream domain inlet profiles along Line E and comparison of profiles along Line F (Figure 4c) are measured by the Laser Doppler Anemometry (LDA) System in wind tunnel tests. Additionally, CFD inlet profiles of stream-wise velocity (u) and turbulent kinetic energy (TKE) are fitted following the profiles in experimental data (Figure 5).

All governing equations for the flow and turbulent quantities are discretized by FVM and SIMPLE scheme is used for the
300 pressure and velocity coupling. The under-relaxation factors for pressure term, momentum term, k and ε terms are 0.3, 0.7, 0.8 and 0.8 respectively. CFD simulations do not stop until all residuals become constant. Typical residuals at convergence are $1\times10^{-6}$, $1\times10^{-9}$ and $1\times10^{-6}$ for $Ux$, $Uy$ and $Uz$, respectively, $1\times10^{-7}$ for continuity, $1\times10^{-6}$ for $k$, and $1\times10^{-6}$ for $\varepsilon$.

Figure 6 shows the stream-wise velocity profiles of simulation results and experimental data along the centerline (Line F) of the street canyon. The predicted wind profile agrees well with the wind tunnel data. One main vortex structure is formed in
the street canyon. The center of the main velocity (i.e. stream-wise velocity is 0) also matches well between simulation and experiment.

Furthermore, some statistical parameters, including normalized mean square error (NMSE), fractional bias (FB) and correlation coefficient (R) are calculated by the following equations:

$$NMSE = \frac{\sum_{i=1}^{n}(O_i - P_i)^2}{\sum_{i=1}^{n} O_i P_i} \tag{18}$$

$$FB = \frac{2(\bar{O} - \bar{P})}{\bar{O} + \bar{P}} \tag{19}$$

$$R = \frac{\sum_{i=1}^{n}[(O_i - \bar{O})(P_i - \bar{P})]}{[\sum_{i=1}^{n}(O_i - \bar{O})^2]^{0.5}[\sum_{i=1}^{n}(P_i - \bar{P})^2]^{0.5}} \tag{20}$$

where $n$ is the total number of measurement points; $O_i$ is the experimental data at measurement point $i$; $P_i$ is the CFD results at measurement point $i$; $\bar{O}$ is the mean value of experimental data at all points; $\bar{P}$ is the mean value of CFD results at all points. According to the previous works (Chang and Hanna, 2005; Sanchez et al., 2016), the model acceptance criteria for urban configuration are NMSE<1.5, −0.3<FB<0.3 and R>0.8. In this simulation case, the respective NMSE, FB and R are 0.01, -0.04 and 0.99 (Table 2), which shows the good performance of the APFoam in flow field simulation.

## 3.3 Pollutant dispersion in 2D street canyon

Currently, there are rarely wind tunnel experiments with chemical reactions. Thus, the pollutant dispersion accuracy in 2D street canyon is validated by wind tunnel experimental data with tracer gas (Meroney et al., 1996), following the pervious study (He et al., 2017; Zhang et al., 2020). The wind tunnel, also the CFD domain configuration is presented in Figure 7. 28 rows of the wooden bar with 27 street canyons are set from upstream toward downstream along the inflow and the street axis is perpendicular to the wind direction. Both the height (*H*) and width (*B*) of the bar are 0.06 m and the street canyon width (*W*) is also 0.6 m, i.e. aspect ratio (*H/W*) is 1 in this study case. A pollutant line source of ethane ($C_2H_6$) is set to emit the pollutant in targeted street canyon. Following the wind tunnel configuration, there are 20 bars upstream and 8 bars downstream of the targeted street canyon. Eight measurement points are set in the targeted street canyon, and four (P4, P5, P6, P7) of them are on the leeward side and the rest of four (P11, P12, P13, P14) are on the windward side. The positions of the measurement points are demonstrated in Figure 7. Pollutant concentrations at each measurement point are normalized with respect to that of the P7 ($C_i/C_7$) within the street canyon (Sanchez et al., 2016; Santiago and Martín, 2008). For CFD simulation, APreactingFoam solver with the standard $k$-$\varepsilon$ model is applied to solve the compressible unsteady-state turbulent flow field and pollutant dispersion. In order to be consistent with the wind tunnel experiments setting, photochemical mechanism is not used in the simulation. The minimum grid size in this case is 0.5 mm with expansion ratio of 1.1 from the wall surface toward surrounding, and inlet velocity is constant as 3 m s$^{-1}$ in the simulation. Time step of simulation is set as $1\times10^{-4}$ s in this validation case.

As the comparison results, Figure 8 shows the normalized concentrations between the CFD simulation and experimental data. In general, the model slightly overestimates the $C_2H_6$ concentrations on windward side. However, at P4, the model

concentrations for the top of the leeward side are lower than the experimental data, and the simulation results at P5 overestimates the concentrations of pollutant. In this simulation case, the respective values of NMSE, FB and R are 0.06, -0.13 and 0.95 (Table 3), which shows the good performance of the APFoam in 2D pollutant dispersion simulation.

## 3.4 Pollutant dispersion in 3D street canyon

As mentioned in section 3.3, 3D pollutant dispersion validation with tracer gas is conducted in this study, following the pervious study (Zhang et al., 2019b). Simulation results also compares with the wind tunnel experimental data (Chang and Meroney, 2001). CFD domain configuration is presented in Figure 9a. In this case, six buildings are set in the domain. Building
height ($H$) and street canyon width ($W$) is 0.08 m with the $H/W = 1$. Building length ($Lx$) and building width ($Ly$) is 0.276 m and 0.184 m, respectively. The distance between buildings and domain inlet, side boundary, top boundary and domain outlet are respective $5H$, $5H$, $10H$ and $15H$, for simulating a realistic results (Tominaga et al., 2008). Within the target street canyon, there are also 8 measurement points (4 of which on the leeward side and 4 on the windward side) for measuring the concentrations (Figure 9b). Besides, 6 more measurement points are also set on the top of the downstream building. Pollutant
concentrations at each measurement point in this simulation case are normalized with respect to the P5 ($C_i/C_5$) within the street canyon. The source of the $C_2H_6$ is set as an inlet at the bottom of the target street canyon. The size of the source is 0.005 m in width and 0.092 m in length setting in the middle of canyon. The release velocity is 0.01 m s$^{-1}$ toward top boundary and the mass fraction of the $C_2H_6$ is 1 (pure gas of $C_2H_6$). For 3D pollutant dispersion simulation, APreactingFoam solver with Standard k-ε model is applied to solver compressible unsteady-state turbulent flow and pollutant dispersion as well. Photochemical
mechanism is not used in the simulation. The minimum grid size in this case is 0.0005 m with expansion ratio of 1.1 from the wall surface toward surrounding. Time step of simulation is set as $1\times10^{-4}$ s in this validation case as well. Meanwhile, the inlet velocity and TKE profile are also retrieved from and fitted by the experimental data (Figure 10).

Figure 11 shows the comparison results between CFD simulation and experimental data. Overall, CFD simulation in 3D dispersion case slightly overestimates the concentrations in the street canyon. As for P23 and P24, the simulated results also
overestimate, effected by the higher concentrations predicted within street canyon. Similarly, Statistical variables such as NMSE, FB and R are calculated to evaluate the performance of the model. As shown in Table 4, the value of NMSE, FB and R are 0.16, -0.21 and 0.93 in the 3D dispersion case, respectively, which agrees with acceptance criteria. In general, APFoam also shows the good performance in the 3D pollutant dispersion simulation.

## 4 Numerical results in case study

### 4.1 Simulation configuration and CFD setting

In this study, APFoam with CS07A photochemical mechanism is applied for street air quality simulation. As shown in Figure 12a, the street aspect ratio ($H/W$) is one with building height ($H$=24 m), street width ($W$=24 m) and span-wise street length ($L$=30 m). Telescoping multigrid approach is adopted in the simulation with minimum grid size of 0.2 m and expansion ratio of 1.2 from building walls to the surrounding. The total grid number is about 87300 for the whole CFD domain. Top and two lateral boundaries of the domain are set up as the symmetry boundary condition.

Emissions area is set up at the bottom of street canyon with the pollutant source size of 18 m (width, $W_E$) × 30 m (length, $L_E$) × 0.3 m (height, $H_E$), representing the traffic emission near street ground, and emission data are obtained from the our previous work (Wu et al., 2020). In this study, the emissions of $NO_x$, VOCs and CO are $4.37 \times 10^{-8}$, $2.34 \times 10^{-8}$ and $2.03 \times 10^{-7}$ kg m$^{-3}$ s$^{-1}$ (i.e., ~35, ~200 and ~170 ppbv s$^{-1}$), respectively. The NO and $NO_2$ are separated from $NO_x$ by the ratio of 9:1, which is similar with the previous study (Baik et al., 2007). VOCs are speciated following SAPRC mechanism and the emission fraction of the species is obtained from the literature (Carter, 2015).

Figure 12b shows the probe point locations for numerical case, where temporal variations of reactive pollutant concentrations are monitored, which include three points at pedestrian height of $z = 1$ m (near street bottom, at leeward side (LB), center (CB) and windward side (WB)) and two other probe points near street top (ST, $z = 23$ m) and street center (SC, $z = 12$ m $= 0.5H$).

Power-law velocity vertical profile is adopted for the inflow boundary condition, which is described as following:

$$U_{in}(z) = U_{ref} \times \left(\frac{z - H}{z_{ref}}\right)^{\alpha} \tag{21}$$

$$k_{in}(z) = (U_{in}(z) \times I_{in})^2 \tag{22}$$

$$\varepsilon_{in}(z) = \frac{C_\mu^{\frac{3}{4}} k(z)_{in}^{\frac{3}{2}}}{\kappa z} \tag{23}$$

Here reference velocity $U_{ref}$ is 3 m s$^{-1}$; reference height $z_{ref}$ is 24 m; turbulence intensity $I_{in}$ is 0.1; power-law exponent $\alpha$ is 0.22 (He et al., 2017; Zhang et al., 2019a, 2020); Von Karman constant $\kappa$ is 0.41 and $C_\mu$ is 0.09.

Besides, the initial and inlet background concentrations for $O_3$, NO, $NO_2$, VOCs and CO are 60 ppbv, 5 ppbv, 15 ppbv, 40 ppbv and 400 ppbv, respectively, which are obtained from an observation campaign (Liu et al., 2008). For meteorological conditions, the temperature is 300 K and the operating pressure is 1013.25 hPa.

In all simulation cases, steady state turbulence field is first solved in advance. The result of turbulent flow would drive the chemistry solution from $t = 0$. During $t = 0$–30 min (1800 s), the emission and chemistry solution are turned on under the statistically steady turbulent flow, reaching a quasi- dynamic and photostationary steady state. Data from next 60 minutes ($t = 30$–90 min, 1800–5400 s) are used for analysis. Time step of chemistry solution is set as 0.1 s in all numerical cases.

The description of all simulation cases is listed in Table 5. To investigate the effect of chemical mechanism, background condition of the precursors (BC), emission (Emis) and wind condition ($U_{ref}$) on the reactive pollutant concentrations in the street canyon, the cases of BC_zero_out, Emis_zero_out and Uref0.5 are set up in numerical simulations. In the Case_BC_zero and Case_Emis_zero, the precursors of $O_3$ (i.e. $NO_x$ and VOCs) are removed from domain inlet (background boundary conditions) and pollutant source emissions, respectively, and then we compare the results with Base. In the Case_Uref50%, the $U_{ref}$ is reduced by 50% to investigate the contribution of wind condition on the chemical reaction. However, in the Case_simple_mech, only three photochemical reactions (Leighton, 1961) are considered in the simulation:

$$NO_2 + hv \rightarrow NO + O(^3P)$$
$$O(^3P) + O_2 \rightarrow O_3 + M$$
$$O_3 + NO \rightarrow NO_2 + O_2$$

In order to improve the air quality within the urban area, some cities have tried to implement traffic control policies to reduce the pollutants from vehicle emission sources. Thus, four emission control scenarios are carried out to investigate the effect of emission reduction. Case_Emis_Ctrl50% is the scenario that reducing 50% of the traffic volume by applying such as odd-even license plate policy (i.e., reduce 50% of the total vehicle emissions). Case_Emis_Ctrl_VOC20%, Case_Emis_Ctrl_VOC30% and Case_Emis_Ctrl_VOC40% are the scenarios which apply the stricter VOCs control measures (corresponding to 20%, 30% and 40% more VOCs emission reduction which are 60%, 65% and 70% reduction of total VOCs emission, respectively) on the vehicles with traffic control policies.

Additionally, the change rate is used to reveal the effect of different factors on pollutant concentrations in street canyon. For each pollutant, the change rate ($CR_p$) for different cases is defined as

$$CR_p(\%) = \frac{C_{case} - C_{base}}{C_{base}} \times 100\% \tag{24}$$

where $C_{case}$ and $C_{base}$ are the concentrations regarded as condition change case and base case, respectively.

**4.2 The comparison of pollutant distribution among the 3D CFD solvers**

To investigate the difference of APonlyChemReactingFoam, APreactingFoam and APSteadyReactingFoam results, the comparisons of $O_3$, NO, $NO_2$ and CO distribution are conducted in $H/W = 1$ street canyon in this study. For APonlyChemReactingFoam, the flow field is treated as the incompressible steady-state flow and pre-solved using the SIMPLE method. The under-relaxation factors and residuals threshold for convergence are same as the setting in section 3.2. Chemical reaction and pollutant dispersion are solved under the steady-state flow for 90 minutes. For APreactingFoam and APSteadyReactingFoam case, turbulence flow, chemical reaction and pollutant dispersion are solved simultaneously for 90 minutes. The results in Figure 13 and all subsequent Figure are the pollutant dispersion at 90 minutes.

As depicted in Figure 13, the wind speed in APonlyChemReactingFoam case (Figure 13a) is lower than that in APreactingFoam (Figure 13b) and APSteadyReactingFoam (Figure 13c) case. The reason for the difference is most likely due to the different turbulence flow algorithm, where the turbulence is treated as incompressible steady flow, compressible unsteady flow and compressible steady flow in APonlyChemReactingFoam, APreactingFoam and APSteadyReactingFoam, respectively. Because of the slightly difference in wind speed, the concentrations of APonlyChemReactingFoam (Figure 13d, g, j, m) for pollutants are higher (due to the lower wind speed) than that in APreactingFoam (Figure 13e, h, k, n) and APSteadyReactingFoam (Figure 13f, i, l, o) case.

Table 6 shows the elapsed time of these three simulations in same $H/W = 1$ street canyon for 90 minutes simulation. Totally, the elapsed time of APonlyChemReactingFoam case (226 minutes) is slightly longer than that of APreactingFoam (214 minutes) and APSteadyReactingFoam (217 minutes) case while employing 192 CPU cores (16 × Intel$^®$ Xeon$^®$ E5-2692) for simulation. However, if the flow filed has been determined and no need to recalculate in the simulation case, the APonlyChemReactingFoam only takes 191 minutes for solving the chemical reaction and pollutant dispersion, which is 11% less time than APreactingFoam.

Many previous studies have treated the urban air turbulence as incompressible steady-state flow and investigate the pollutant dispersion successfully(He et al., 2017; Ng and Chau, 2014; Zhang et al., 2019a, 2020, 2019b). With less time spending, the APonlyChemReactingFoam is applied in the study to analyse the photochemical reaction process in the street canyon.

### 4.3 Pollutant concentration distribution with full chemistry mechanism vs simple chemistry

As shown in Figure 13a, one main clock-wise vortex is formed in the street canyon with $H/W = 1$. The wind speed (WS) is small near the vortex center, i.e. the minimum wind speed is approximated 0.03 m s$^{-1}$, which is only 1% of the speed at the domain inlet. The distributions of pollutants, such as $O_3$, NO and $NO_2$ in street canyon are also swirling (Figure 13a, d, g, j).

Leeward-side $O_3$ concentration in Base is less than the windward side, while NO and $NO_2$ are opposite. At the corner of leeward side, the minimum value of $O_3$ and maximum value of $NO_x$ appear, with less than 20 ppbv for $O_3$, more than 200 ppbv for NO and 140 ppbv for $NO_2$ (Figure 13a, d, g, j), respectively. Meanwhile, due to the higher NO emissions, the ratio of NO and $NO_2$ are higher at the bottom of the street canyon (Figure 14a). The larger NO/$NO_2$ values indicate that the titration effect of from NO ($O_3 + NO \rightarrow NO_2 + O_2$) and ozone depletion would be stronger, leading to the lower $O_3$ concentrations in this area.

While on the windward side, $NO_x$ concentrations are less than that on the leeward side. This is because that the $NO_x$ from emission source first affects the leeward side which leads to the high concentrations in this area. As the wind flows, the concentrations of $NO_x$ gradually decrease due to the wind diffusion and dilution effect. With the comparison of background, the windward NO and $NO_2$ concentrations increase approximately 35 ppbv and 55 ppbv, respectively. On one hand, pollutants

from emissions are transported along the flow which increases the concentrations. On the other hand, VOCs in street canyon react with OH via the chemical reactions and generate $HO_2$ and $RO_2$. These $RO_2$ and $HO_2$, $O_3$ would react with NO and generate $NO_2$, leading a higher increment for $NO_2$ (Figure 14b-14c). It should be noted that the $O_3$ concentrations could be higher due to the less depletion reaction with NO compared to that of the leeward side.

Figure 15 shows the change rate of pollutant concentrations (Figure 15a–15c) and NO to $NO_2$ ratio change rate of Case_simple_mech compared with the Base (Figure 15d). Without the consideration of the VOCs-related reactions in the mechanism, there is no consumption of NO by $RO_2$ in the mechanism (VOCs + OH → $RO_2$ + $H_2O$, $RO_2$ + NO → RO + $NO_2$), and NO titration effect ($O_3$ + NO → $NO_2$ + $O_2$) would be stronger in this case. For the $O_3$, the concentrations are 36%–58% less than that in Base within the street canyon; $NO_2$ concentrations are also 15%–40% less and NO could be up to 90% higher than that in the simple chemistry case. Thus, the NO to $NO_2$ ratio would be 60%–150% higher in simple chemistry case.

## 4.4 Influence of background precursors of $O_3$ on reactive pollutant concentrations

In the Case_BC_zero, all background precursors of $O_3$ (i.e., $NO_x$ and VOCs) from upstream domain inlet are removed. As depicted in Figure 16a, the change rates of $O_3$ are negative confirming that $O_3$ concentration becomes lower without the background $NO_x$ and VOCs. In addition, the $O_3$ reduction rate on the windward side (-5% to -8%) is smaller than that on the leeward side (-9%). The influencing mechanisms of $O_3$ reduction are complicated and will be explained later.

On the one hand, by analysing Figure 16b-16c, such reduction rates on the windward side for NO (-12% to -16%) and $NO_2$ (-20% to -24%) are greater than those on the leeward side (-2% to -6% for NO and -12% to -15% for $NO_2$). Therefore, the ratio of NO/$NO_2$ increases for about 9% to 14% in the street canyon (Figure 16d). Overall, this increment of NO/$NO_2$ enhances $O_3$ depletion because the main source of ozone is the photolysis reaction of $NO_2$, meanwhile, the main sink is the titration effect of $O_3$ and NO.

On the other hand, the $RO_2$ from the oxidation of VOCs with OH will consume the NO (VOCs + OH → $RO_2$ + $H_2O$, $RO_2$ + NO → RO + $NO_2$), which would affect the $NO_x$–$O_3$ circulation. Figure 16e shows that the reduction of $RO_2$/OH on the windward side is more than that on the leeward side, which indicates that the background VOCs and OH reaction on the windward side is more active. However, due to the slower reaction rate of $RO_2$ and NO compared with that of $HO_2$ and $O_3$ with NO, the conversion of $RO_2$ to RO by reacting with NO would require more time. The reduction rate of RO/$RO_2$ (Figure 16f) on the leeward side (-18% to -23%) are slightly greater than that of on windward side (-16% to -17%). Therefore, the influence of $RO_2$ on $NO_x$–$O_3$ circulation would gradually appear on the leeward side, along with the flow transportation.

Additionally, Figure 16g shows the reaction rate of $RO_2$ ($\frac{dRO_2}{dt}$) at the bottom, center and top point on the centerline of the street canyon with (base) and without (case) background conditions. At the bottom point (CB), the reduction rate of $RO_2$ is lower in Base. This is because that the background VOCs and OH reaction consumes a portion of NO on the windward side which leads to a lower consumption of $RO_2$ with NO. As the simulation continues, NO concentrations would increase due to

the continuous release of large amounts of NO from source emissions, and the reduction rate of $RO_2$ become lower in Case_BC_zero due to the lack of background VOCs.

At top (ST) and center point (SC), however, as the reaction goes on and pollutants mix up, the NO concentration could become higher. Therefore, the $RO_2$ consumption rate is less without the background $RO_2$. This reduction indicates that the background conditions mainly consume NO in the street canyon which leads to the increase of $O_3$ due to the weakening titration effect.

## 4.5 Effects of vehicular source emissions on reactive pollutants

In the Case_Emis_zero, the precursors of $O_3$ ($NO_x$ and VOCs) from near-ground emissions are removed. As shown in Figure 17a, $O_3$ concentrations increase by over 30%-120% in the whole street canyon compare to the Base. In particular, $O_3$ increment on the leeward side is from 80% to 250% (not shown here) which is much higher than that on the windward side (30% to 40%). However, $NO_x$ concentrations decrease significantly, i.e., the reduction rates are -84% to -98% for NO and -76% to -90% for $NO_2$, showing that the $NO_x$ from source emissions are dominant part of $NO_x$ in the street canyon (Figure 17b-17c). The large reduction of $NO_x$ concentrations induces the increase of $O_3$ concentrations with weaker titration effect of $O_3$. Particularly, both the maximum increase for $O_3$ and minimum reduction of NO and $NO_2$ appear at the near-ground corner of the leeward side, where is the downwind area of the pollutant source. Besides, due to a larger amount of NO emissions than $NO_2$ (emission ratio of NO to $NO_2$ is 9:1), the concentration ratio of NO/$NO_2$ considerably decreases with the reduction rates of -30% to -70% (Figure 17d) if vehicular pollutant sources are removed.

Additionally, due to the large reduction of NO and $NO_2$ concentrations, more OH would react with VOCs instead of $NO_x$, which increases the $RO_2$ concentration (Figure 17e-17f). Meanwhile, with the reduction of NO concentration, the consumption of $RO_2$ significantly decreases, which leads to the dramatic increase of $RO_2$ concentration. Thus, the ratio of $RO_2$/OH rises by 115%-205% and the ration of RO/$RO_2$ decreases by -60% to -88%.

In Figure 17g, the reaction rate of $RO_2$ at three points in Case_Emis_zero is positive, which means that the $RO_2$ keeps being generated but not consumed among these three points. As mentioned above, $RO_2$ (the production of VOCs and OH) will cosume the NO and weaken the $O_3$ titration effect with NO. In Base case (Figure 17g), the reaction rate of $RO_2$ is negative, which means that $RO_2$ consumes the NO. However, in Case_Emis_zero, reaction rate of $RO_2$ is positive during the whole simulation period which means that there is not enough NO to react with $RO_2$ or even $O_3$ without the vehicular source. Therefore, the source emissions provide a large amount of NO which enhances the $O_3$ depletion in the street canyon.

## 4.6 Influence of wind velocity reduction on reactive pollutants

Figure 18 shows the change rates of $O_3$, $NO_x$, and ratios when the background wind speed decreases from $U_{ref}$ =3 m s$^{-1}$ (Base) to $U_{ref}$=1.5 m s$^{-1}$ (Case_Uref50%). In Figure 18a, there is no significant change of $O_3$ concentration at the center of street canyon. However, in the downwind area of the near-ground pollutant source, $O_3$ concentration decreases by 5% to 30%, compared with that of Base. Interestingly, at the bottom of the leeward side, $O_3$ has an increase up to 6% under the half inlet wind speed condition.

Due to the weaker capacity of pollutant dilution caused by the smaller wind speed, the concentrations of NO and $NO_2$ almost double (i.e. rising by 80%-98% in Figure 18b-18c) but the NO/$NO_2$ change rate has no significant change (-1% to -3% in Figure 18d). Besides, a higher $NO_x$ concentration would react with more OH which consequently weakens the $RO_2$ production from VOCs (-8% to -20% in Figure 18e). Meanwhile, the increase of NO concentration consumes more $RO_2$ to RO, which leads to 180% to 340% increase of RO/$RO_2$ (Figure 18f).

Additionally, Figure 18g illustrates the $RO_2$ reduction in street canyon. Because of the higher concentration of NO in Case_Uref50%, the $RO_2$ reduction rates in three monitoring points are higher than that in Base, particularly at the bottom of the street canyon (CB). In the early stage of the reaction, the reduction rate of $RO_2$ at the top point (ST) is slightly lower in Case_Uref50%. This is because that the $RO_2$ concentration at ST is firstly affected by the background. As the NO concentrations increases in the whole street canyon, the $RO_2$ consumptions become higher than that in Base.

## 4.7 Emission control strategy on reactive pollutant concentrations

Figure 19 shows the concentrations of $O_3$, NO and $NO_2$ in different NOx and VOCs emission control scenarios at 90 minutes. In Case_Emis_ctrl50% (the emission of $NO_x$ and VOCs reduces 50%, i.e. 50% reduction of traffic volume), the $O_3$
concentration increases from 19-47 ppbv to 29-54 ppbv (Figure 19a). On the contrary, this control measure for NO and $NO_2$ are very effective (Figure 19b-19c), and NO and $NO_2$ concentrations reduce 47% to 54% and 37% to 40% in Case_Emis_ctrl50%, respectively.

This indicates that the simple traffic control measures cannot effectively reduce $O_3$ concentration. It is because that most of the urban areas are the VOC-sensitive regions (Ye et al., 2016). When the total number of vehicles decreases under the
545 traffic control measures, the reduction of $NO_x$ is higher than that of the VOCs (due to the larger $NO_x$ emission from vehicles), which leads to a higher VOCs-to-NOx ratio, consequently resulting in a higher $O_3$ concentration in the street canyon (Sillman and He, 2002). Thus, in order to reduce the concentrations of $O_3$, the stricter VOCs control measures on vehicle should be conducted. Based on the results shown from three other emission control scenarios (Case_Emis_ctrl_VOCs20%, Case_Emis_ctrl_VOCs30% and Case_Emis_ctrl_VOCs40%) in Figure 19a, the emission of VOCs needs be reduced by

550 another 30% under traffic control (Case_Emis_ctrl50%) to bring the $O_3$ concentrations back to the level when no traffic control measures have been taken (Base).

  As for NO and $NO_2$, when the additional VOCs control measures are carried on, the concentrations are higher than those in Case_Emis_ctrl50%. Even so, their concentrations do not still exceed the concentration level before the traffic control (Base), which means that such emission control scenario is still effective for the NO and $NO_2$. In Summary, the control policies of
555 reactive pollutants require the comprehensive consideration of the relationship between precursors and pollutants, so that the purpose to improve air quality can be achieved.

## 5 Conclusions

A detailed description of the Atmospheric Photolysis calculation framework APFoam-1.0 is presented in this paper, and this
CFD model is coupled with multiple full atmospheric photochemical mechanisms, including SAPRC07 (CS07A and SAPRC07TB) and CB05. In order to simulate the photochemical process of reactive pollutants, five new types of the reactions, including the new form of the (1) Arrhenius reactions; (2) Photolysis reactions; (3) Falloff reactions; (4) "Three $k$" reactions; and (5) "Two $k$" reactions have been modified and added into the APFoam. Additionally, to verify the model performance, several validations, including photochemical mechanism (CS07A) with SAPRC box modelling, flow field, 2D and 3D
pollutant dispersion with wind tunnel experimental data have been conducted in this study. The model results show a good agreement with the SAPRC box modelling and wind tunnel experimental data, indicating that the APFoam can be applied in the analysis of micro-scale urban pollutant dispersion.

  By applying the APFoam with CS07A mechanism in the simulation of reactive pollutants in typical street canyon ($H/W$ = 1) with the VOCs to $NO_x$ emission ratio ~ 5.7 in ppbv s$^{-1}$, key factors of chemical processes are investigated. In the
570 comparison of chemical mechanism, $O_3$ and $NO_2$ are underestimated by 36%–58% and 15%–40%, respectively, while NO is overestimated by 30%–90% without the consideration of the VOCs reactions. Other numerical sensitivity cases (Case_BC_zero, Case_Emis_zero and Case_Uref50%) reveal that vehicle emission is the main source of the NO and $NO_2$, with the contribution of 82%–98% and 75%–90%, respectively. The resident part of the $NO_x$ in street canyon is contributed by the background concentration. However, vehicle emissions with a large amount of emitted $NO_x$, especially NO, is a main
reason for the decrease of $O_3$ due to the stronger NO titration effect within the street canyon. In contrast, 5%–9% of the $O_3$ are contributed by the boundary conditions. Ventilation condition is another reason for the $NO_x$ concentrations increment, and the increase of $NO_x$ can be up to 98% when the wind speed is reduced to the half. If no chemical reactions, NOx concentration should rise 100% when the wind velocity decreases 50% (i.e. ventilation capacity reduces 50%) since the Re-independence requirement is satisfied. However, at the downwind of the emissions, $O_3$ is reduced due to the increase of NO concentrations.
In order to control and improve the air quality in the street canyon, traffic control policies are effective for $NO_x$. However, our results indicate that at least another 30% reduction in vehicle VOCs emissions can reduce the $O_3$ concentrations under the odd-

even license plate policy with 24%-32%, 25%-28% and -6%-2% reduction rates of NO, NO$_2$ and O$_3$, respectively. Overall, APFoam-1.0, a fully coupled CFD model, can be employed to investigate the atmospheric photolysis calculation in urban areas, and provide reliable and useful suggestions for the improvement of urban air quality.

## 6 Future plans

However, in the current version of the APFoam, aerosol chemistry is not included in the model, and it is necessary to couple with the aerosol processes, such as MOSAIC (Zaveri et al., 2008) or ISORROPIA (Fountoukis and Nenes, 2007; Nenes et al., 1998) in the future work. Besides, the photolysis rates in the current model have been fixed without the diurnal variation,

which means that the model is not suitable for a long-term simulation, and needs to be updated in the subsequent versions. Moreover, the interaction between radiation and chemical reaction will be investigated by APFoam and validated by the scaled outdoor experiment (Chen et al., 2020a, 2020b) in the future.

*Code availability.* The source code of the APFoam-1.0 model and examples are available on GitHub (https://github.com/vnuni23/APFoam, last access: 18 November 2020) and Zenodo (https://doi.org/10.5281/zenodo.4279172). More information and help are also available by contacting the authors.

*Author contributions.* LW and XW designed the experiments. LW and CG developed the model code. LW and JH performed

the simulations and organized the results of model cases. LW and JH prepared the article with contributions from all coauthors. XW and MS proposed revision suggestions for the article.

Competing interests. The authors declare that they have no conflict of interest.

*Acknowledgements.* We acknowledge the technical support and computational time from the Tianhe-2 platform at the National Supercomputer Center in Guangzhou, and the support from Collaborative Innovation Center of Climate Change, Jiangsu province, China.

*Financial support.* This research has been supported by the National Key Research and Development Program of China (grant

no. 2016YFC0202206), and the National Nature Science Fund for Distinguished Young Scholars (grant no. 41425020).

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

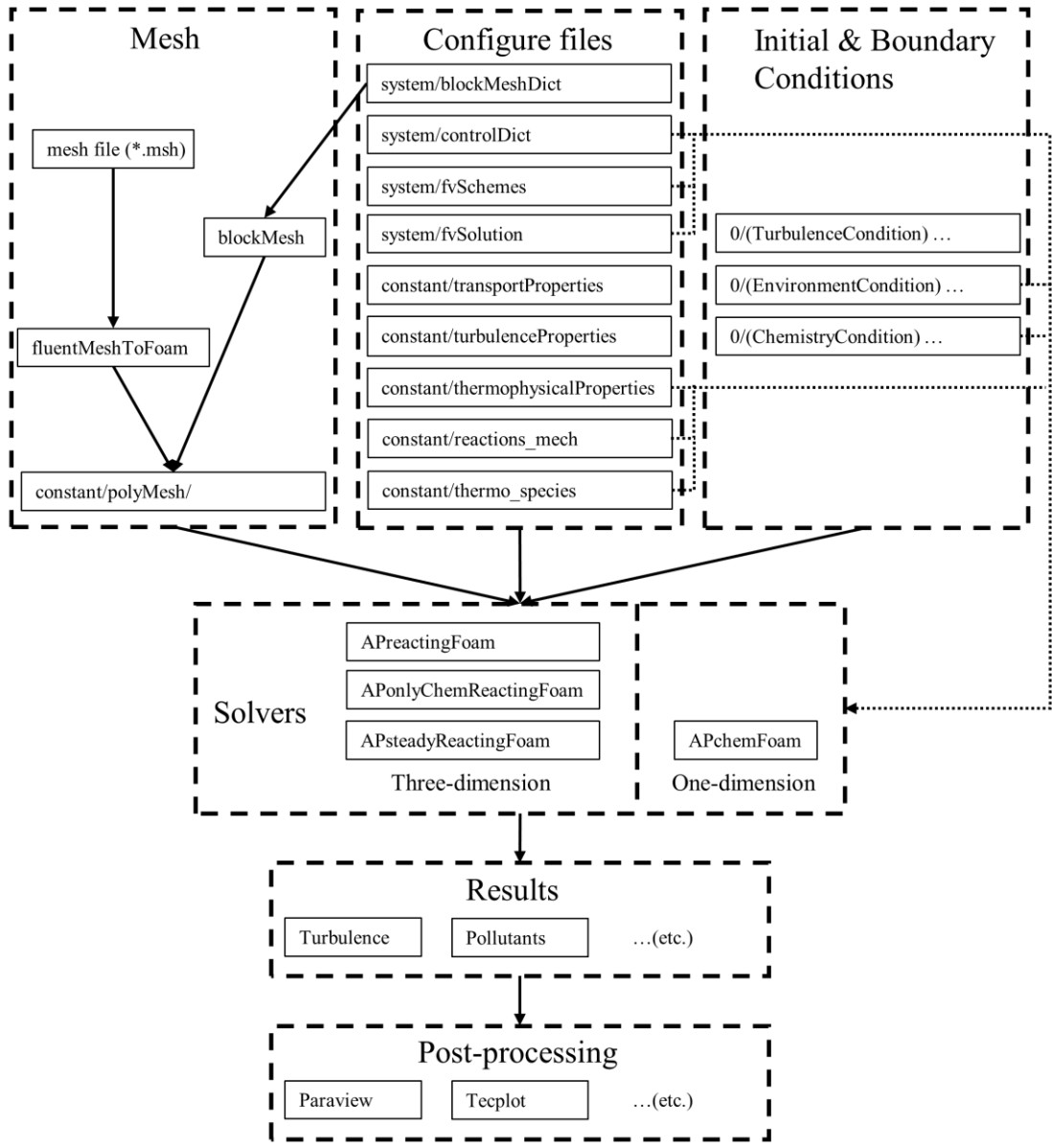

**Figure 1. Flow diagram of simulation set-up in APFoam framework**

(a)                                                                (b)

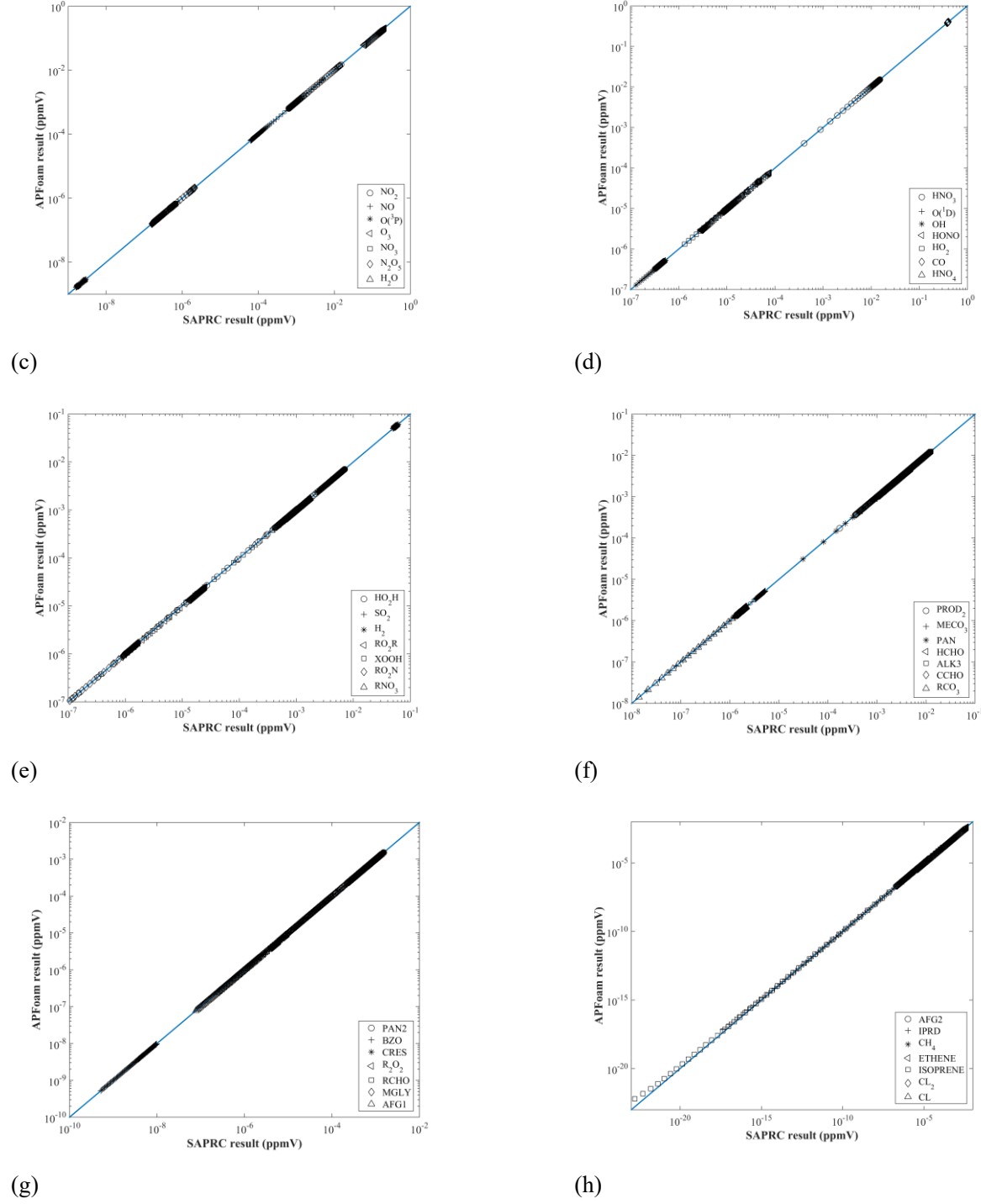

(c)

(d)

(e)

(f)

(g)

(h)

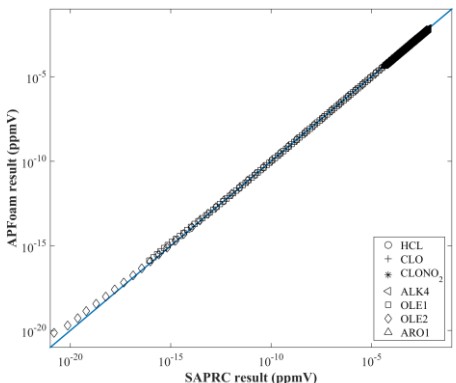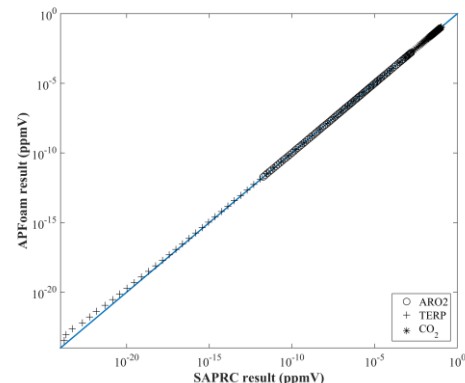

**Figure 2.** The comparison of concentration result between APFoam and SAPRC box model

(a)                                      (b)

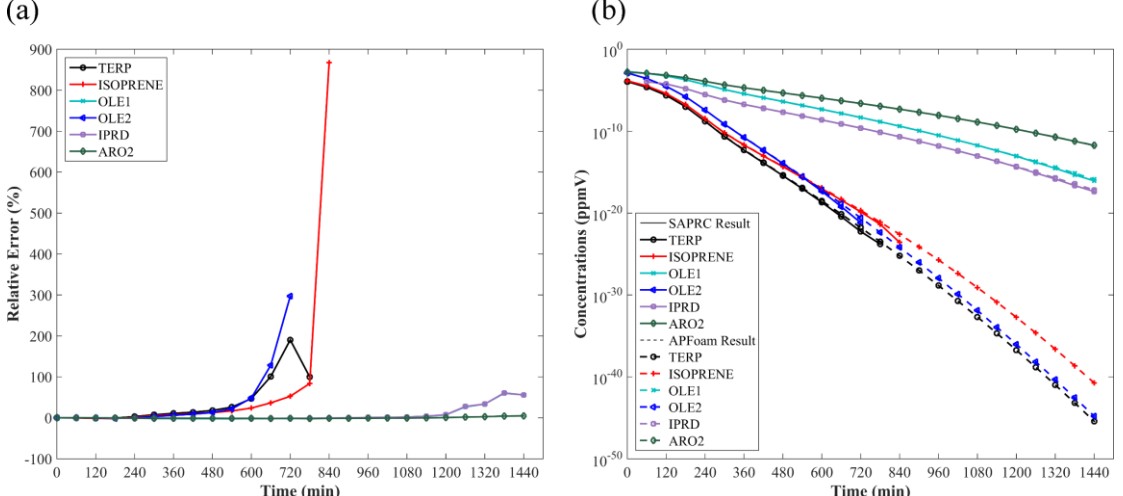

**Figure 3. The time series of (a) relative error (%) and (b) concentrations (ppmv) of 6 species with largest bias**

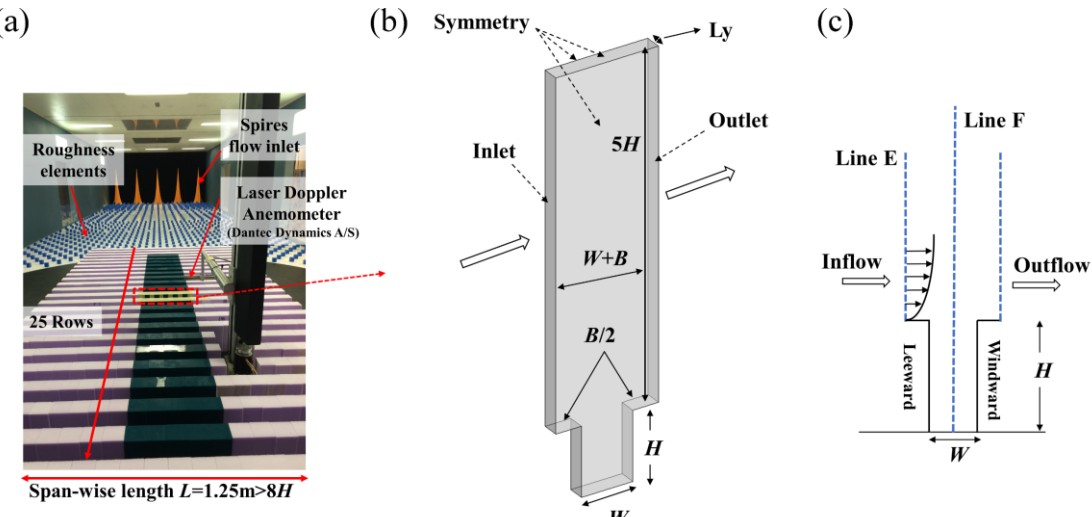

**Figure 4. (a) Wind tunnel experiment in 2D street canyon; (b) the single street canyon CFD domain setups in scaled model and (c) the inlet profile (Line E) and measurement profiles (Line F) in the street canyon.**

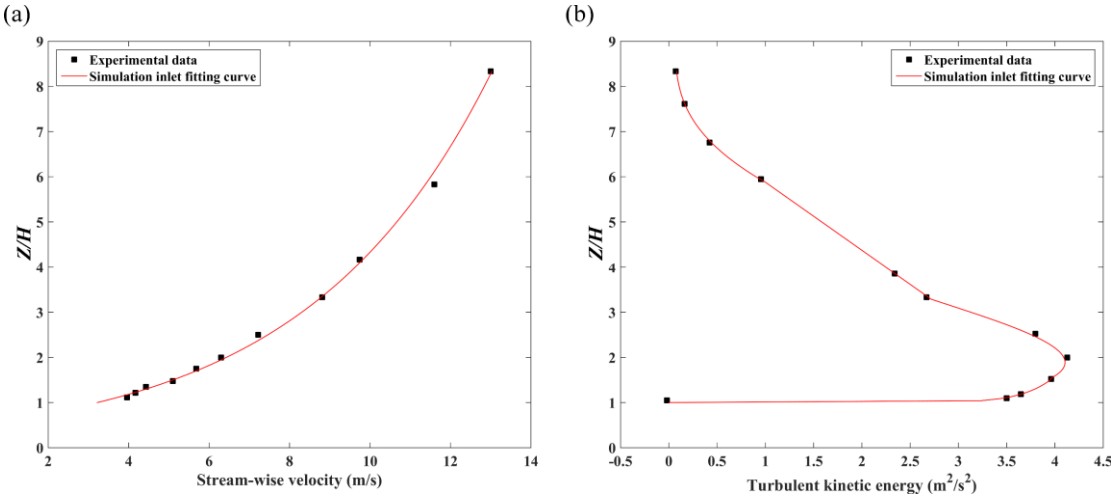

**Figure 5. The inlet profile of (a) stream-wise velocity and (b) turbulent kinetic energy in single street canyon case**

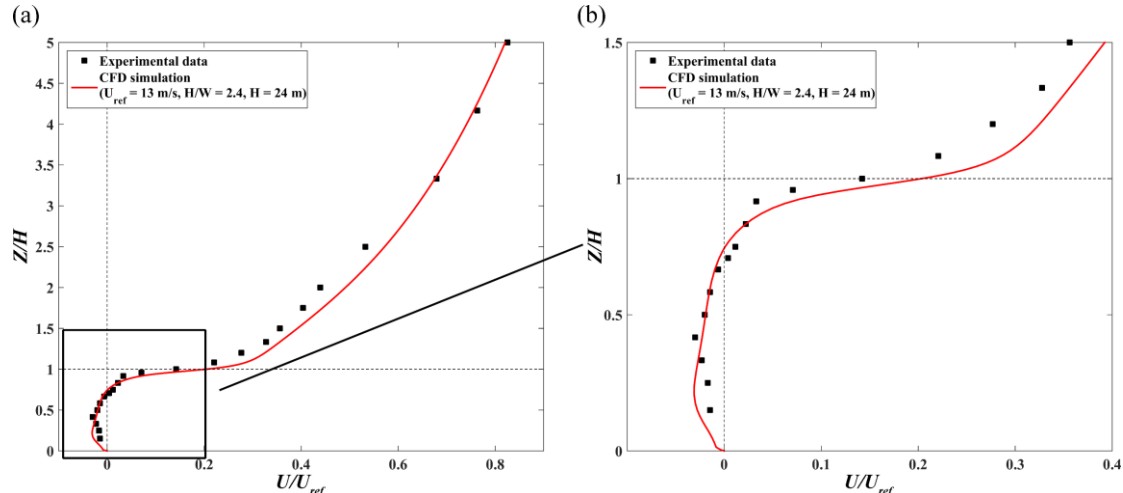

**Figure 6. The stream-wise velocity profiles along the Line F**

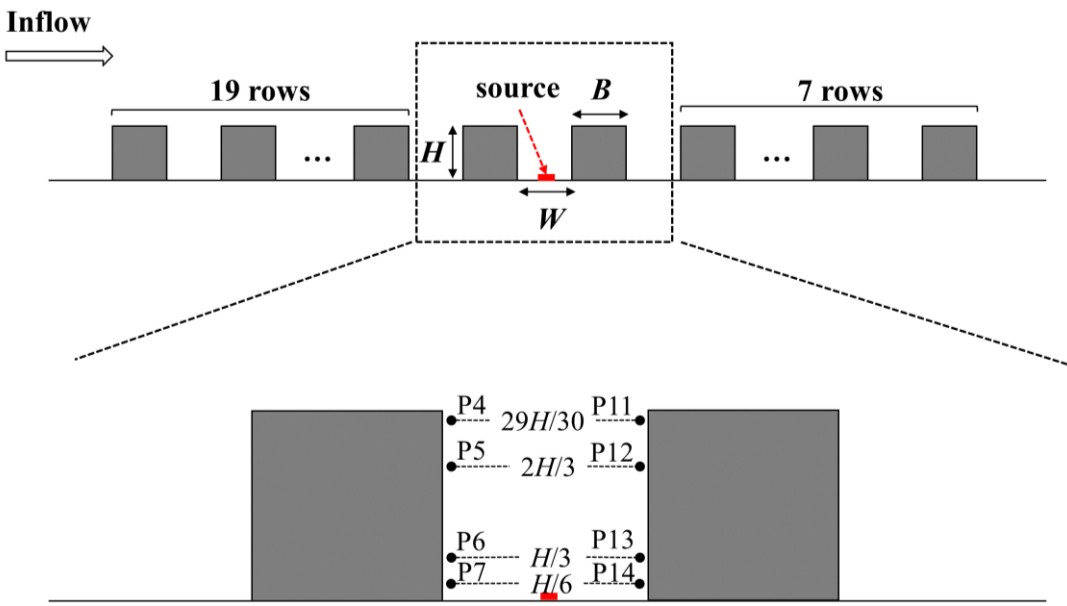

**Figure 7. the schematic diagram of the 2D pollutant dispersion simulation setting**

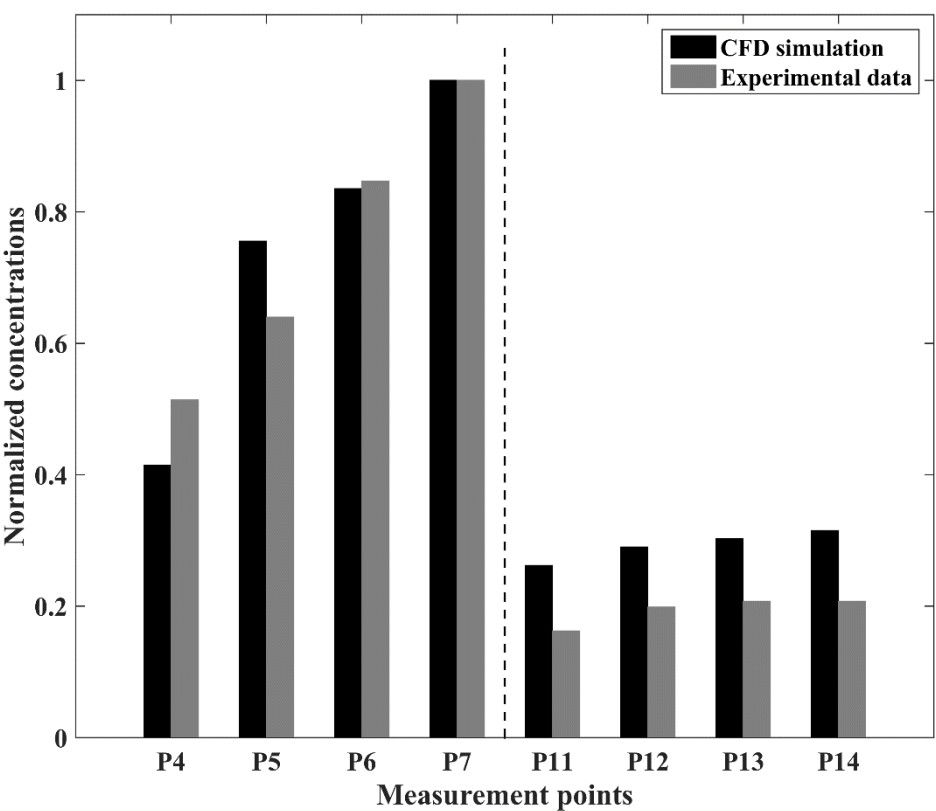

**Figure 8. Normalized concentrations of CFD and experimental data at each measurement point in 2D dispersion case**

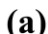

**(a)**

Symmetry

Outlet

Inlet

10*H*

*H*

source

*W*

15*H*

Ly/2

*Ly*

*Lx*

5*H*

5*H*

**(b)**

source

115.5*Lx*/276

263.5*Lx*/276

62.5*Lx*/276

213.5*Lx*/276

12.5*Lx*/276

160.5*Lx*/276

P23  P24  P25    P26  P27  P28

P2   25*H*/32  P13

P3   23*H*/40  P14

P4   3*H*/8  P15

P5   5*H*/32  P16

**Figure 9. (a) The simulation domain of 3D pollutant dispersion and (b) the measurement points setting in the street canyon**

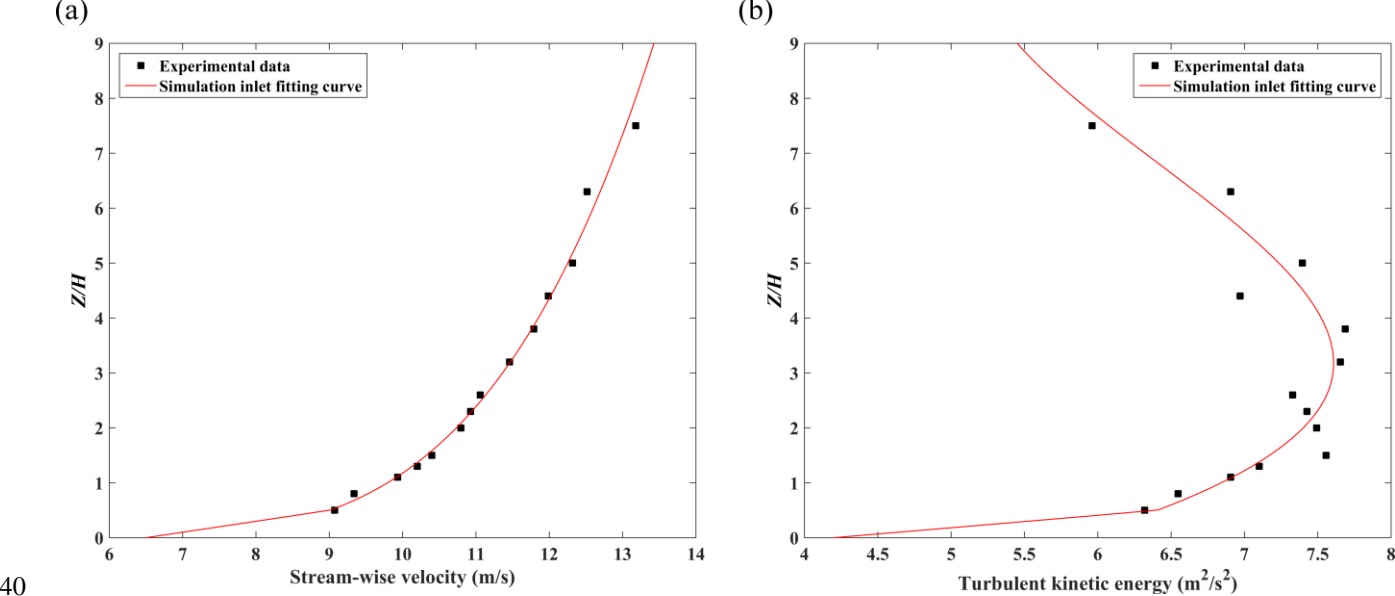

**Figure 10. The inlet profile of (a) stream-wise velocity and (b) turbulent kinetic energy in 3D dispersion case**

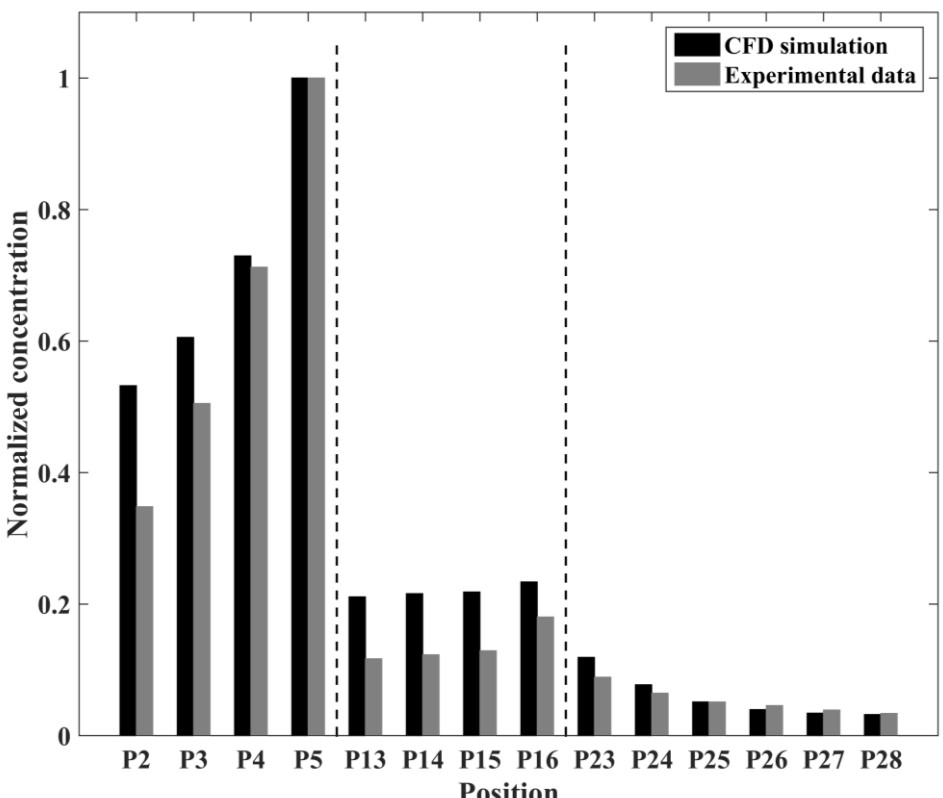

**Figure 11. Normalized concentrations of CFD and experimental data at each measurement point in 3D dispersion case**

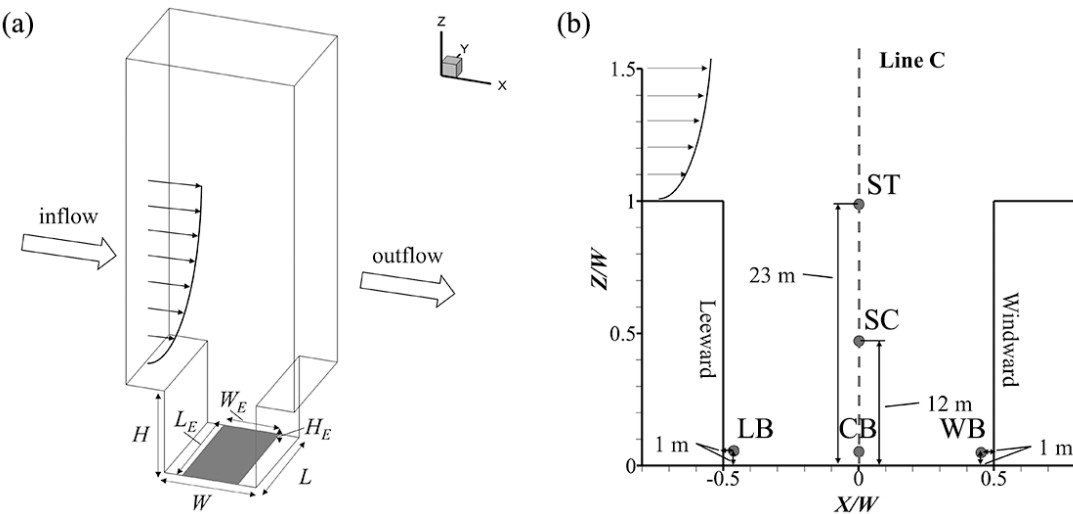

**Figure 12. Schematic diagram of (a) the CFD simulation domain and (b) the probe points locations**

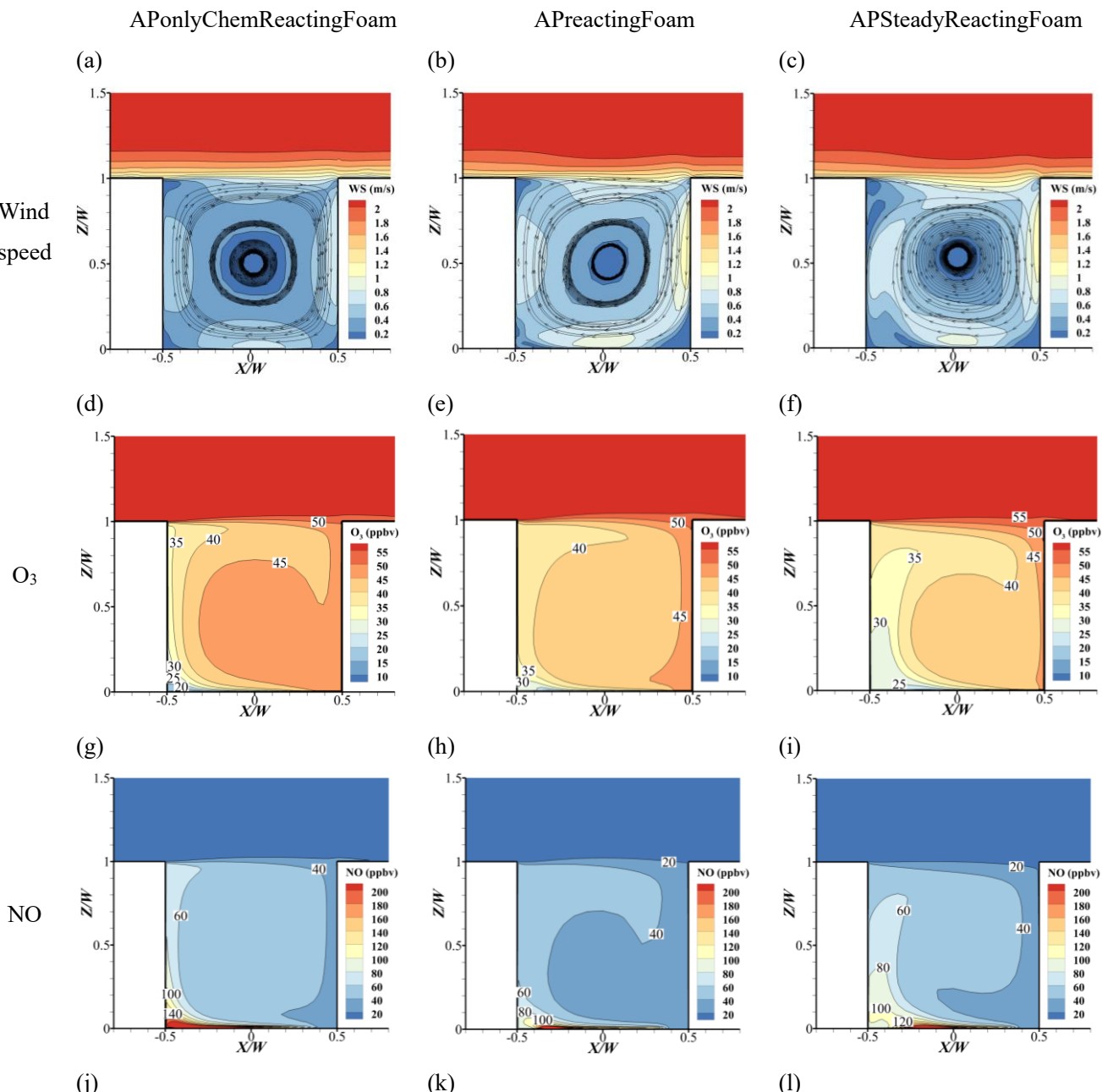

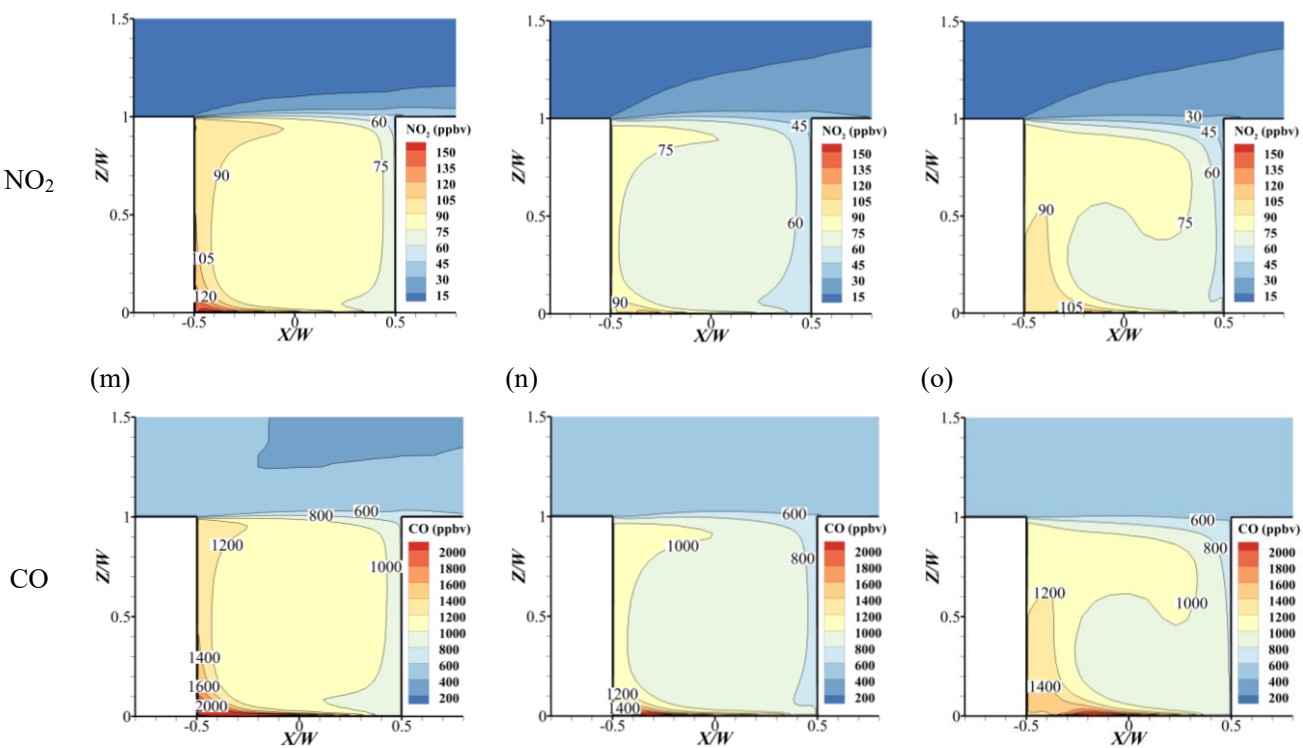

NO₂ (m) (n) (o)

CO

**Figure 13. The comparison of (a-c) wind speed, (d-f) O₃, (g-i) NO, (j-l) NO₂ and (m-o) CO between APonlyChemReactingFoam, APreactingFoam and APSteadyReactingFoam**

(a)       (b)

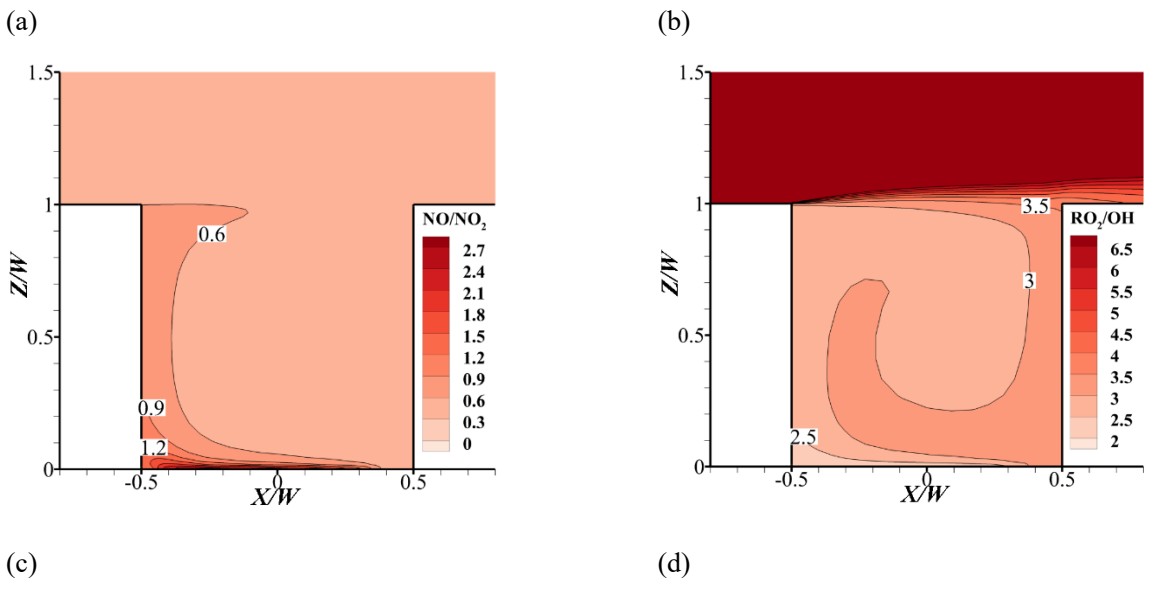

(c)       (d)

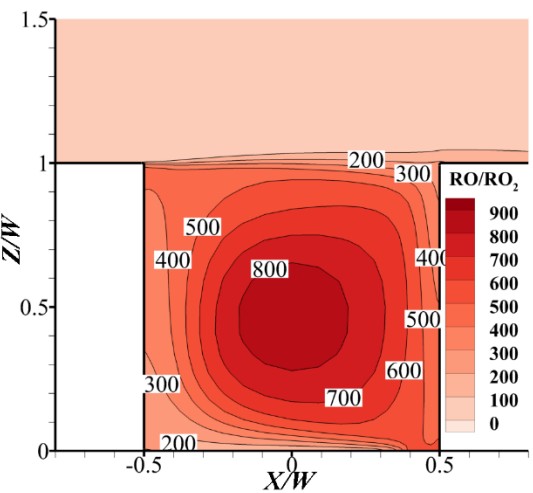

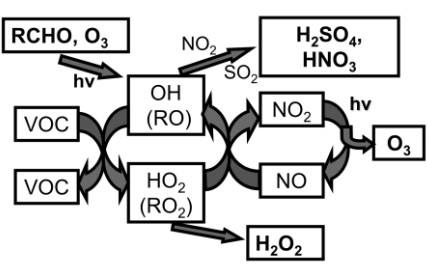

**Figure 14. Pollutant distribution of (a)NO/NO₂, (b) RO₂/OH, (c) RO/RO₂ in Base and (d) Schematic diagram of the formation mechanism of photochemical reactions (Tang et al., 2006)**

(a)

(b)

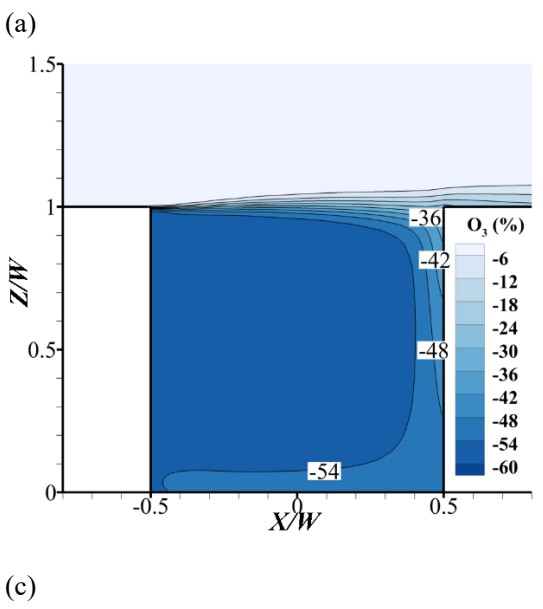

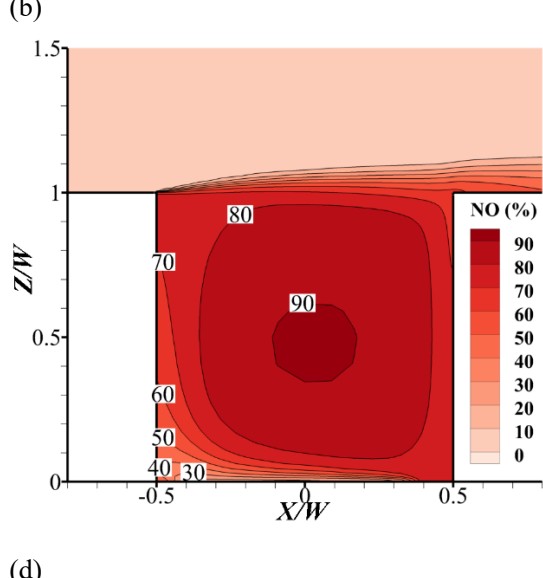

(c)

(d)

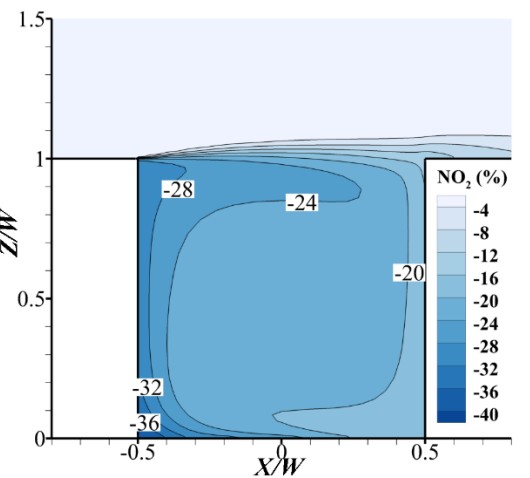

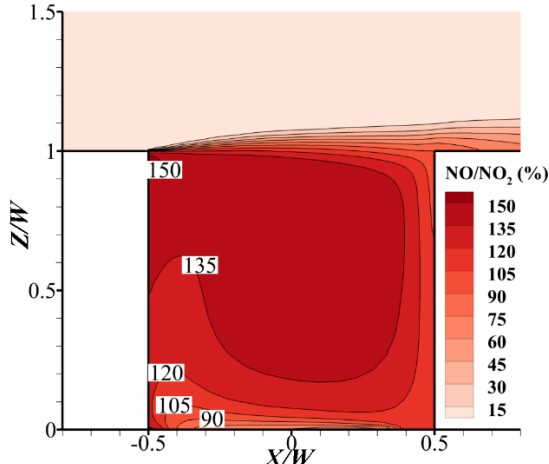

**Figure 15. Change rate of (a) O₃, (b) NO, (c) NO₂, (d) NO/NO₂ between simple chemistry and full chemistry mechanism**

(a)

(b)

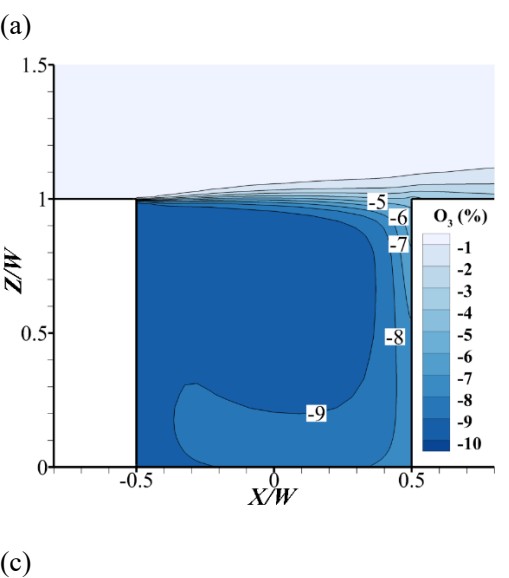

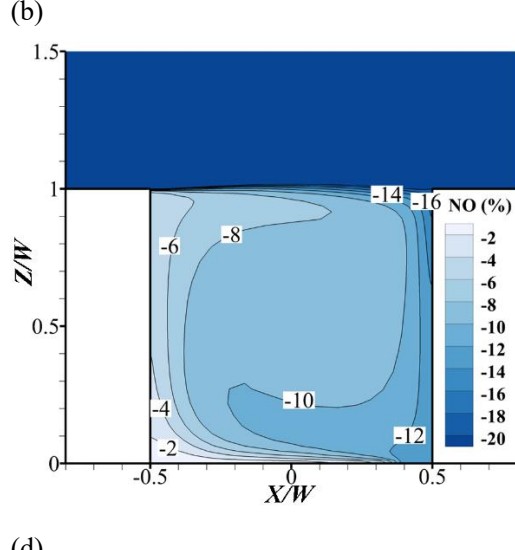

(c)

(d)

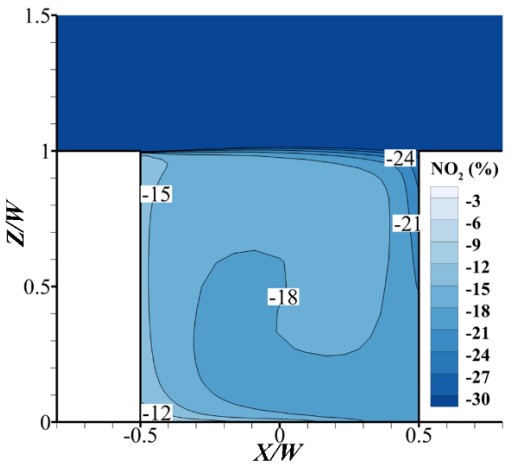

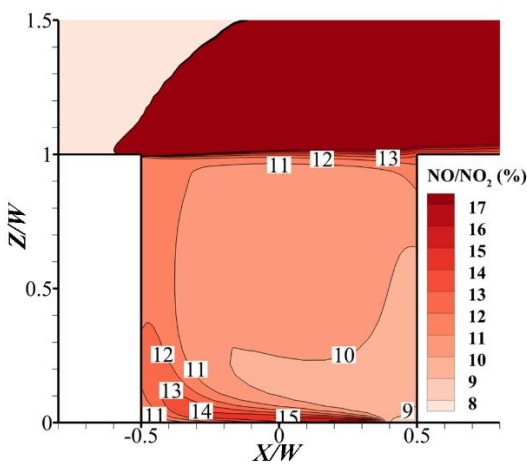

(e)

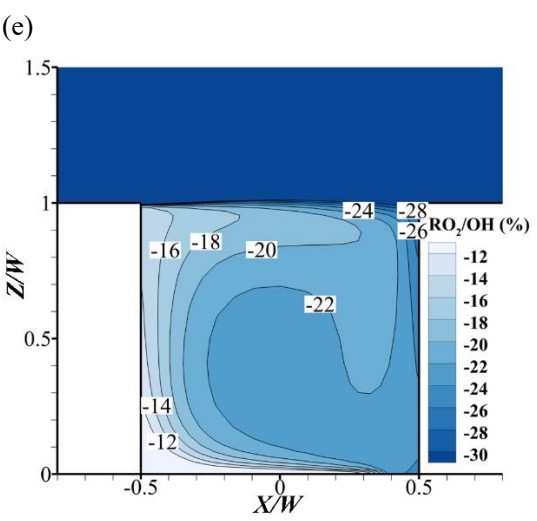

(f)

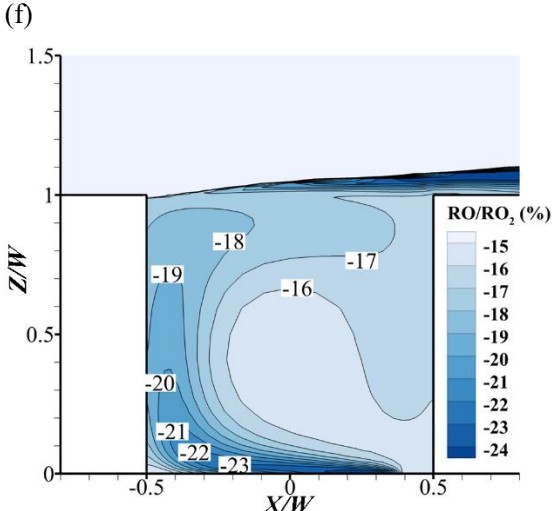

(g)

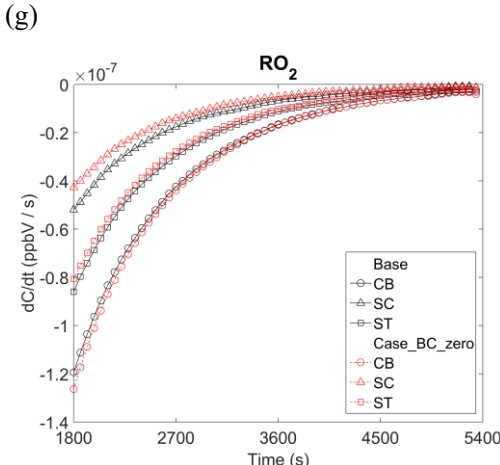

**Figure 16. Change rate of (a) O₃, (b) NO, (c) NO₂, (d) NO/NO₂, (e) RO₂/OH, (f) RO/RO₂ at t = 5400 s and (g) time series of reaction rate of RO₂ ($\frac{dRO_2}{dt}$) of Case_BC_zero**

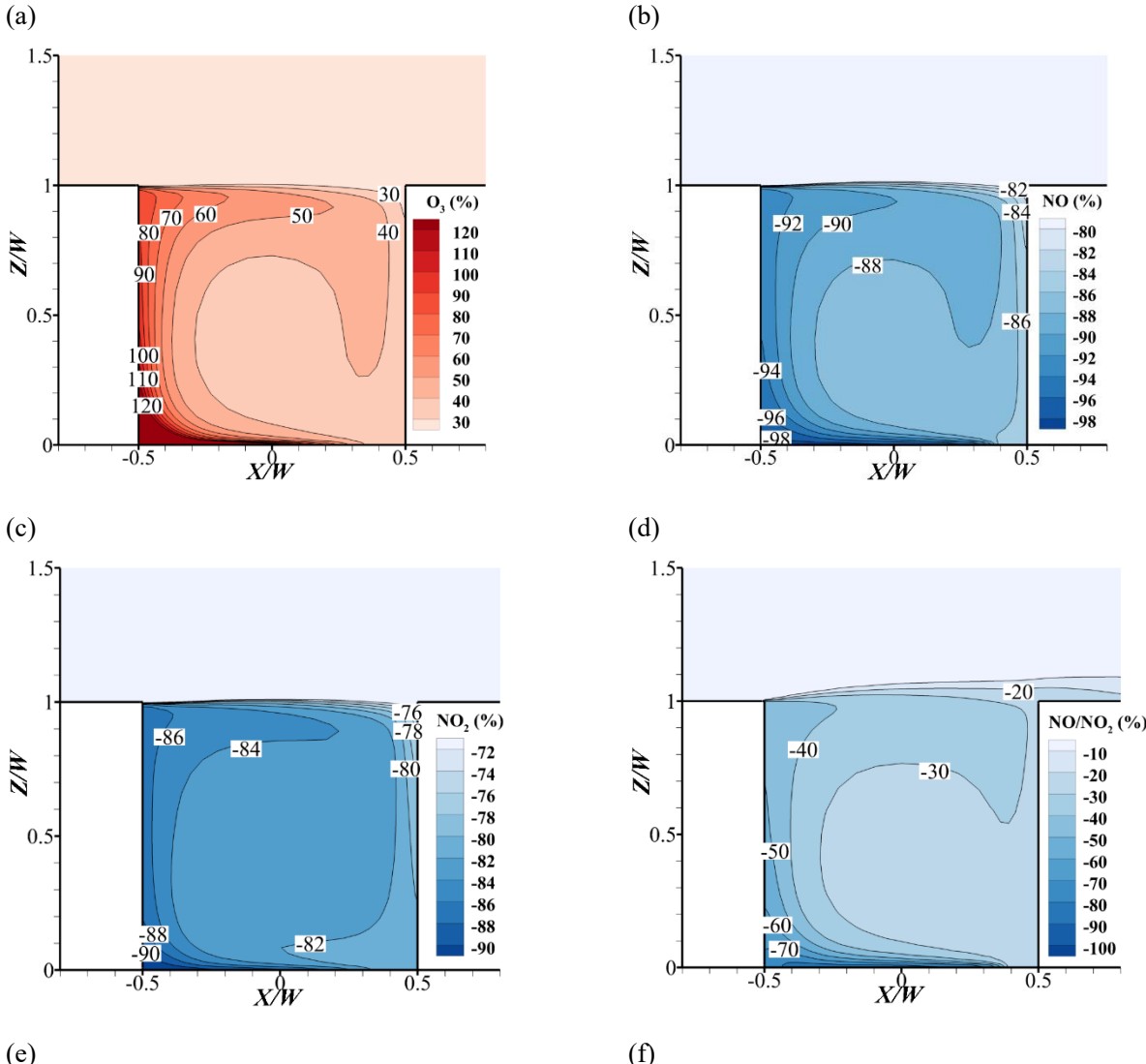

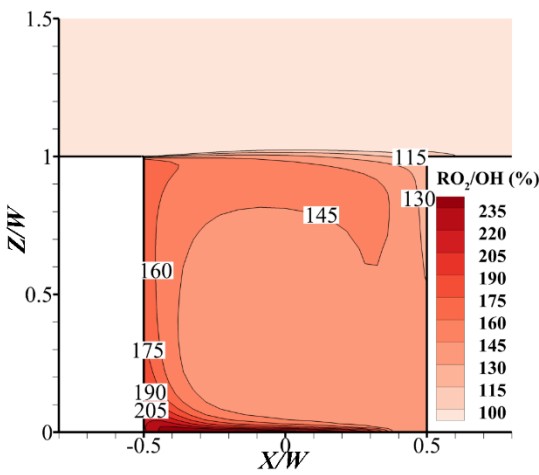 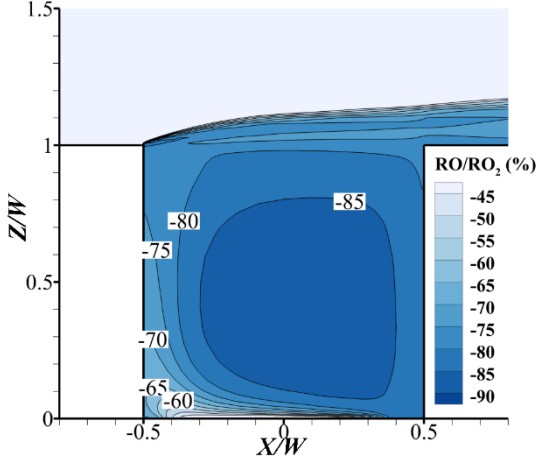

(g)

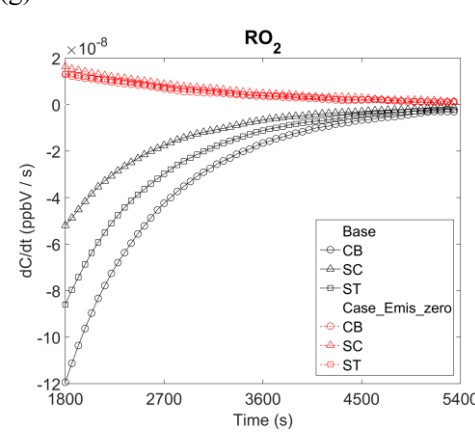

**Figure 17.** Change rate of (a) $O_3$, (b) NO, (c) $NO_2$, (d) $NO/NO_2$, (e) $RO_2/OH$, (f) $RO/RO_2$ at t = 5400 s and (g) time series of reaction rate of $RO_2$ ($\frac{dRO_2}{dt}$) of Case_Emis_zero

(a)                                                                              (b)

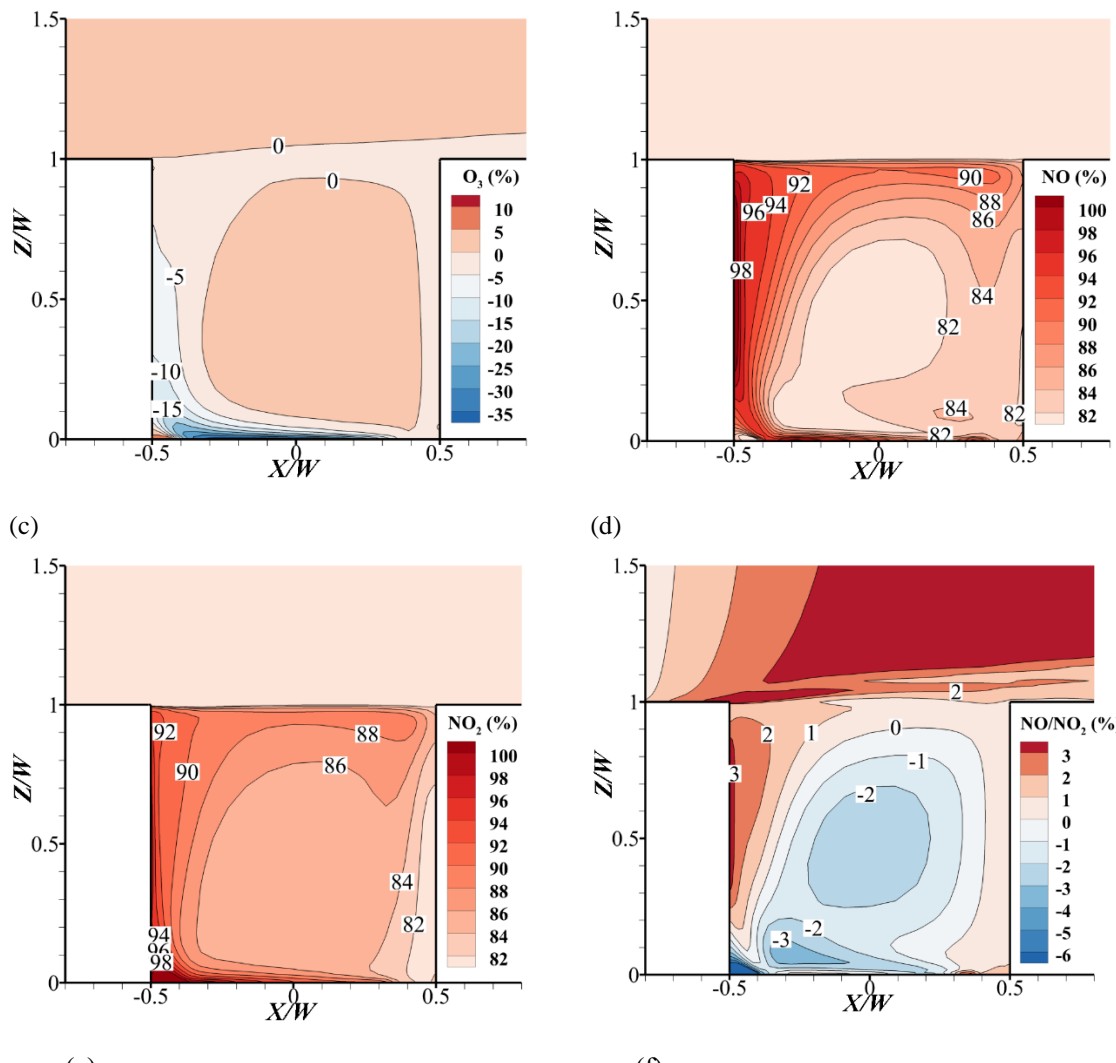

(c)

(d)

(e)

(f)

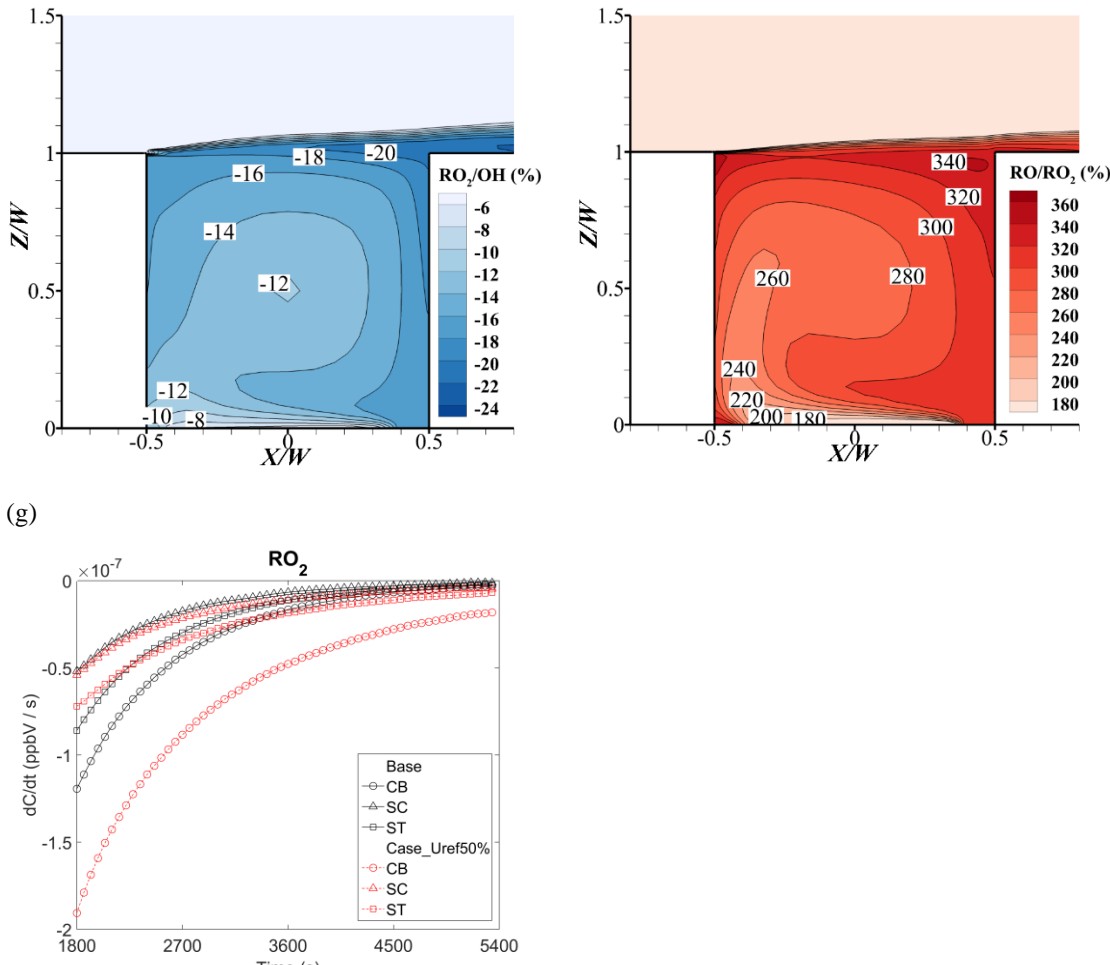

(g)

**Figure 18.** Change rate of (a) $O_3$, (b) NO, (c) $NO_2$, (d) $NO/NO_2$, (e) $RO_2/OH$, (f) $RO/RO_2$ at t = 5400 s and (g) time series of reaction rate of $RO_2$ ($\frac{dRO_2}{dt}$) of Case_Uref50%

(a)

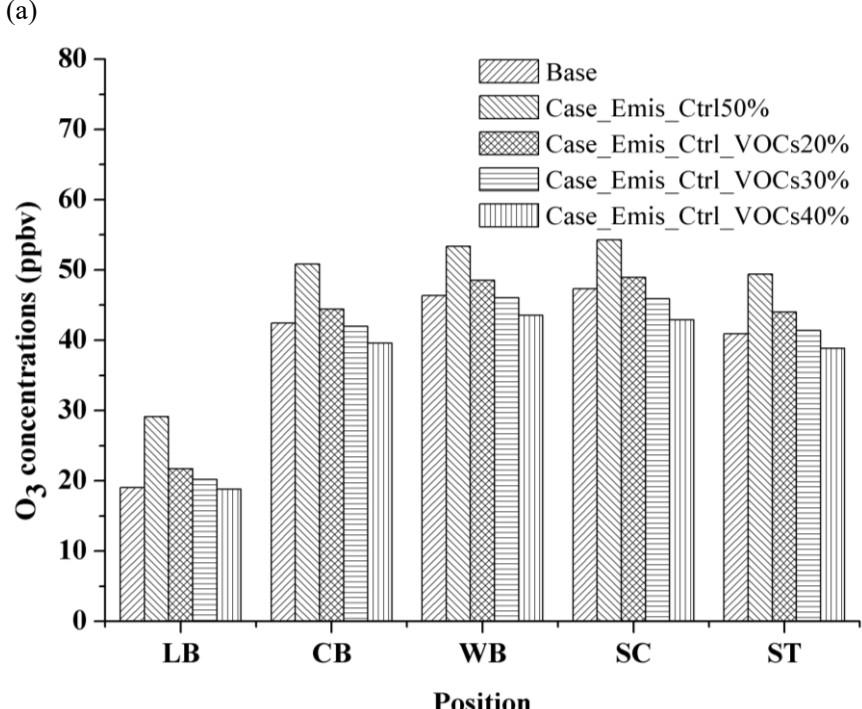

(b)

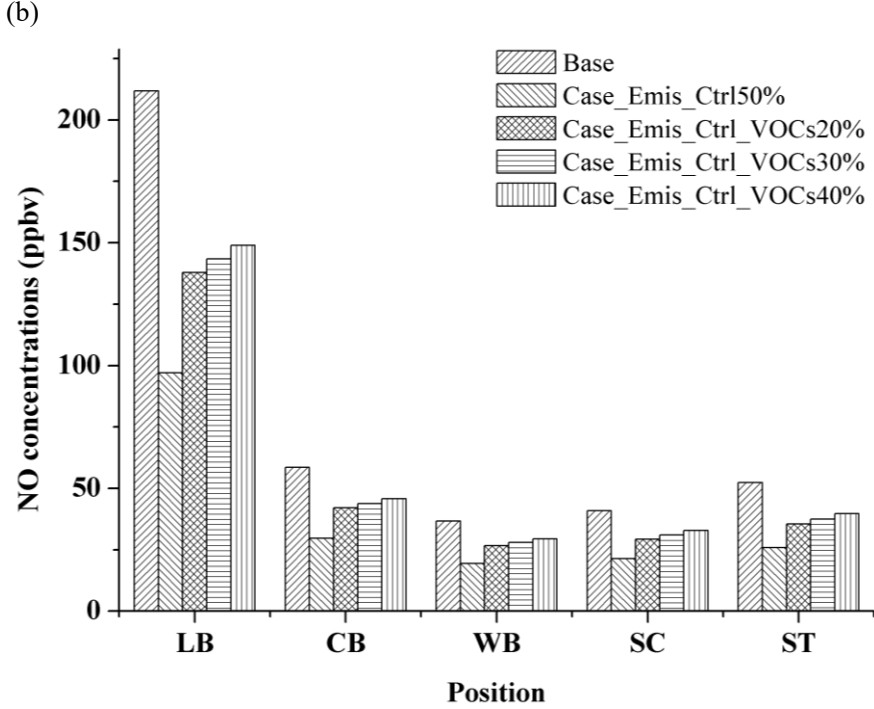

(c)

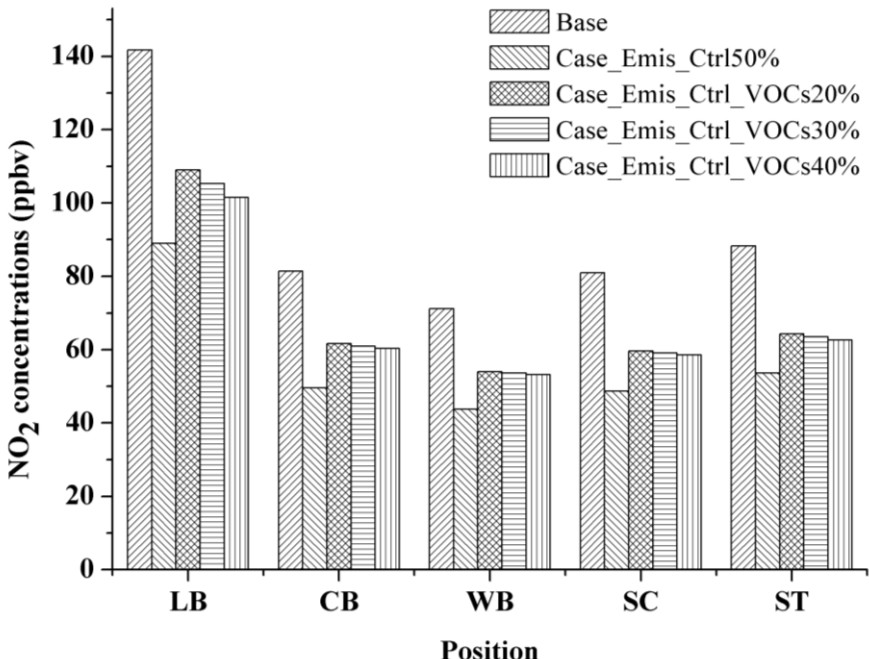

**Figure 19.** The (a) $O_3$, (b) NO and (c) $NO_2$ concentrations at 90 minutes in different emission control scenarios

**Table 1. Overview of the CFD studies with photochemical mechanism**

| study | photochemical mechanism | Parameter | Platform |
|---|---|---|---|
| Baker et al., 2004 | simple | wind conditions | RAMS |
| Baik et al., 2007 | simple | thermal effects | Own code |
| Zhong et al., 2015 | simple | aspect ratio | OpenFOAM |
| He et al., 2017 | simple | aspect ratio | Fluent |
| Liu et al., 2018a | simple | emissions | Own code |
| Merah and Noureddine, 2019 | simple | wind conditions | Ansys-CFX |
| Zhang et al., 2019b | simple | emissions | Fluent |
| Zhang et al., 2020 | simple | aspect ratio | Fluent |
| Garmory et al. 2009 | CBM-IV | chemical mechanism | Fluent |
| Kim et al., 2012 | GEOS-Chem | emissions | Own code |
| Kwak and Baik, 2012 | CBM-IV | emissions | Own code |
| bright et al., 2013 | RCS | chemical mechanism | RAMS |
| Kwak et al., 2013 | CBM-IV | wind conditions | Own code |
| Kwak and Baik, 2014 | CBM-IV | thermal effects | Own code |
| Park et al., 2016 | GEOS-Chem | thermal effects | Own code |
| Sanchez et al., 2016 | CCM | chemical mechanism | STAR-CCM+ |
| Zhong et al., 2017 | RCS | chemical mechanism | OpenFOAM |

**Table 2. Statics values of the turbulence flow simulation**

|      | Acceptance criteria | This study |
|------|---------------------|------------|
| NMSE | < 1.5               | 0.01       |
| FB   | (-0.3,0.3)          | -0.04      |
| R    | >0.8                | 0.99       |

**Table 3. Statics values of the 2D pollutant dispersion simulation**

|      | Acceptance criteria | This study |
|------|---------------------|------------|
| NMSE | < 1.5               | 0.06       |
| FB   | (-0.3,0.3)          | -0.13      |
| R    | >0.8                | 0.95       |

**Table 4. Statics values of the 3D pollutant dispersion simulation**

|      | Acceptance criteria | This study |
|------|---------------------|------------|
| NMSE | < 1.5               | 0.16       |
| FB   | (-0.3, 0.3)         | -0.21      |
| R    | > 0.8               | 0.93       |

**Table 5.** The description of all simulation cases

|                   | Mechanism     | Boundary conditions                                                                 | Emissions                                                                                                     | Wind condition            |
|-------------------|---------------|-------------------------------------------------------------------------------------|--------------------------------------------------------------------------------------------------------------|---------------------------|
| Base              | Full (CS07A)  | $BC\_NO_x = 20$ ppbv; $BC\_VOCs = 40$ ppbv; $BC\_O_3 = 60$ ppbv                     | $E\_NO_x = 4.37 \times 10^{-8}$ kg m$^{-3}$ s$^{-1}$; $E\_VOCs = 2.34 \times 10^{-8}$ kg m$^{-3}$ s$^{-1}$    | $U_{ref} = 3$ m/s         |
| Case_simple_mech  | Simple        | *Same as base*                                                                      | *Same as base*                                                                                               | *Same as base*            |
| Case_BC_zero      | *Same as base*| $BC\_NO_x \times 0$; $BC\_VOCs \times 0$                                             | *Same as base*                                                                                               | *Same as base*            |
| Case_Emis_zero    | *Same as base*| *Same as base*                                                                      | $E\_NO_x \times 0$; $E\_VOCs \times 0$                                                                        | *Same as base*            |
| Case_Uref50%      | *Same as base*| *Same as base*                                                                      | *Same as base*                                                                                               | $U_{ref} \times 0.5$      |
| Case_Emis_Ctrl50% | *Same as base*| *Same as base*                                                                      | $E\_NO_x \times 0.5$; $E\_VOCs \times 0.5$                                                                    | *Same as base*            |

| Case_Emis_Ctrl_VOCs20% | *Same as base* | *Same as base* | $E\_NO_x \times 0.5$;<br>$E\_VOCs \times 0.4$ | *Same as base* |
|---|---|---|---|---|
| Case_Emis_Ctrl_ VOCs30% | *Same as base* | *Same as base* | $E\_NO_x \times 0.5$;<br>$E\_VOCs \times 0.35$ | *Same as base* |
| Case_Emis_Ctrl_ VOCs40% | *Same as base* | *Same as base* | $E\_NOx \times 0.5$;<br>$E\_VOCs \times 0.3$ | *Same as base* |

**Table 6.** The elapsed time of three solvers

|  | APonlyChemReactingFoam | APreactingFoam | APSteadyReactingFoam |
|---|---|---|---|
| elapsed time<br>(minute) | 191 + 35 (for turbulence) | 214 | 217 |