# Peer review of "APFoam-1.0: integrated CFD simulation of O3–NOx–VOCs chemistry and pollutant dispersion in typical street canyon"

_Geoscientific Model Development, 2020_

## Referee Comment (RC1) · Anonymous Referee #1 · 26 Dec 2020

**1    General comments**

This paper developed a calculation framework (APFoam-1.0), based on open-source CFD code OpenFOAM, for atmospheric photolysis to examine the micro-scale reactive pollutant formation and dispersion in the urban area. Five new types of reaction are added to the chemistry module, which is coupled with full O3–NOx–VOCs chemistry and CFD model. The model was validated against SAPRC box modeling software and wind tunnel experimental data. The framework was applied to case studies investigating O3–NOx–VOCs formation processes and dispersion of the reactive pollutants in an example of a typical street canyon. APFoam provides a numerical simulation tool

based on the general purpose open source solver, which gives researchers the full capability to have control not only out of the box but also inside of the "box". So APFoam can be a useful tool of broad interest in atmosphere science.

However, the merits of APreactingFoam and APSteadyReactingFoam modules (two important modules of the APFoam-1.0) are not well demonstrated/articulated. Fully coupling (two way coupling) is adopted in both APreactingFoam and APSteadyReactingFoam, where the reaction heat is considered to have impact on fluid flow. Intuitively speaking, the concentration of pollutants is too low to have a significant impact on the fluid field.Whether the heat source from reaction is considered or not might NOT have any significant effect in terms of simulation accuracy: whether to consider or ignore such tiny effects won't change the simulation results too much. But it will have a significant effect on computational cost affecting the speed of simulations. Two way fully coupled model requires solving coupled governing equations with more unknown variables, which usually requires much more computational resources. In addition, the fully coupled system has more constraints on time steps and hence needs smaller time steps. The numerical algorithms become more complicated, too. As a general tool (calculation framework) for doing simulation, computational efficiency is important. Complicated model also makes results analysis more complicated as there are more factors that need to be considered.In that sense, the simplest 3D module, APonlyChemReactingFoam, might be the best choice for most situations due to the following reasons:

- Faster in term of simulation and reduce computational cost significantly as explained above.

- Flow field data can be calculated offline and reused if the CFD set up does not change. For example, in certain case study, you keep the geometry, boundary condition and initial condition unchanged and only vary the source of pollutant (such as the locations, the pollutant release rate et al), then you only need to

do the CFD simulation for once and reuse it for different case studies, this will significantly speed up your case study.

**Validation of the calculation framework is validated only using APonlyChemRe-actingFoam. APreactingFoam and APSteadyReactingFoam are not validated in the validation section. Not clear if APreactingFoam and APSteadyReactingFoam have ever been used in the case study section or not, the author did not explicitly mention that.** Do these two modules (with much more complicated governing equations) really have certain advantages in any situation? If so, I would encourage the author to justify it in a proper way, either based on literature review or ideally with a real case study. The authors need to show that considering and not considering the heat effect from reaction will have significant effect on simulation results for certain cases. With the above being said, It would also be necessary to compare the simulation speed of those three modules. In addition, in the validation section (section 3), the only validated 3D module APonlyChemReactingFoam is not validated in a fully coupled manner. The reaction model is validated alone (section 3.1). The CFD model is validated alone (section 3.2), too. And the pollutant species transportation and dispersion validation (section 3.3) is , at least, decoupled from reaction. The model is fully coupled, but the validation is done in a decoupled manner. Maybe the coupling between reaction and pollutants is also very weak and can be decoupled as well?

In summary, the models seem more complicated than necessary and lack sufficient validation. I would recommend accepting this manuscript to publish on GMD only after the major concern and specific comments (see next section) being properly addressed.

**2 Specificl comments**

Abstract, line 9. Numerical "resolution", in the context of mesh-based methods, such as the finite volume method used in the paper, depends on grid size, which is purely a choice in preprocess. Not clear to me how the framework developed in this paper can "improve the resolution". By "resolution", do you mean to say that the framework you developed targets at modeling small scale phenomena (such as street scale flow)?

Abstract, line 14. The framework is also validated against SAPRC box modeling software, which is an essential validation, why only mention the wind tunnel validation in the abstract? Worthwhile to mention both validations.

Abstract, general comments. Please double check the grammar in the abstract.

Section 1, general comments. It would be helpful for readers to have a better understanding on the major contribution of this paper if you have a little bit more detailed discussion regarding existing CFD based simulation studies in this area. For example, do they consider atmospheric photochemical in their simulation? Are those five reactions recently added in this paper already been studied in other research? In the existing studies, do they consider two way coupling or one way coupling? What tool do they use in their simulation study, OpnFoam or commercial softwares?

Section 2, question. How do you determine the time step for this coupled system?

Section 2.1, line 113. Missing" boundary condition", which is included in Fig. 1 but not mentioned in this paragraph.
$T$ in Eq.1 is supposed to be the temperature of the mixture? In Kelvin or Celsius?

In Eq. 2, Question. How are the lower and upper limit of the integration determined?

Eq. 1 - Eq. 5 use $k$ to represent "reaction rates", while in Eq. 6, $w$ is used as the reaction rate. Are those two reaction rates the same thing? If so, why using different symbols?

In Eq. 8, should the average of molecular weight also be an unknown variable that needs to be calculated based on the mass fraction of each species? Imaging that chemical reaction changes the mass fraction of different species and then leads to changes of the average molecular weight. Any justification why the average molecular weight is assumed to be constant. Actually the assumption that average molecular weight is constant sounds reasonable to me, as the average molecular weight change caused by reaction might be ignorable. Then the following up question is, how significant is the reaction heat source? Seems can be ignorable as well. Authors may need to prove whether they are significant or not.

Section 2.3. For the governing equations (Eq. 6 - Eq. 8) of APChemForm, there are in total n+2 governing equations, I suppose the primitive unknowns in the governing equation are: $T$, $\rho$, $p$, $h$, $Y_i$, there are in total n + 4 unknowns. number of equations < number of unknowns, Seems some other equations missing or not mentioned? Mathematically, the system is not closed. Or the author assumes that two of those $T$, $\rho$, $p$, $h$ are not unknown? Which two?

Section 2, general comments.

- Symbols that are used in this paper only need to be specified for once, for example, $\rho$ is density, you only need to explain it the first time it appears.

- Need to make sure that the same symbol has consistent meaning across the manuscript, make sure the same variable is only represented by one symbol.

- There are several dummy assumptions made in the model: such as, the mixture (mixture of air and pollutants, air itself is mixture) is in a thermal dynamic and dynamic quasi static state, they all share the same temperature and velocity. Maybe worthwhile to explicitly state the assumptions that you made when you establish the model.

- Would be better to explicitly specify that, $\rho$ is density of the **mixture**. Similar for **U** and $T$.

Eq. 10 and Eq. 20, the physical viscosity is also considered. For air, physical viscosity is much smaller than turbulent viscosity and usually ignored in air flow simulation.

Eq. 20, $k$ shows up again here, what does $k$ represent in this equation, recall that $k$ was used as reaction rate in section 2. This is really confusing.

Section 3.2. May worthwhile to explicitly state that the Eq. 18 - Eq. 21 are not a new set of governing equations, they are essentially governing equations Eq. 13 - Eq. 15 with turbulence model.

Section 3.2, line 256. Do you treat the air as incompressible flow in the simulation?

Section 3. In the validation case, what is the Renolds number for the CFD simulation set up? It is not clear to me why you choose to scale the geometry in CFD simulation?

Why not use the same geometry as whatever in the experiment?

Section 3. Why no 3D simulation attempted? Any difficulties or due to computational cost?

Section 3. The validation mainly validates the pollutant transportation and dispersion in the fluids field coupled with reaction, no reaction-flow-transportation coupling situation be tested. Namely, APSteadyReactingFoam, APreactingFoam are not tested. So do the authors also believe that the effect of reaction to flow is too weak that no need to be overly concerned about it?

Section 3.3, general question. Could you clarify which of the following strategies is used in your validation?

- Solve the turbulent fluids governing equation (Eq. 18 - Eq. 21), pollutant species transportation equation (Eq. 12) and reactions (Eq.1 - Eq. 5) simultaneously.

- First do the fluids field simulation without considering pollutant species and reaction, that is, first solve Eq. 18 - Eq. 21. Then do species transportation and dispersion together with reactions based on fluids field solution obtained in the first step? Namely, solve Eq. 12 and Eq. 1 - Eq. 5 in second step.

Section 3.3, general comments. What is the time scale for reactions and what is the time scale for species transportation and dispersion? The motivation to ask this question is to see if there is possibility to decouple reaction from species transportation in such micro-scale simulation. In this section, both the experiment and the numerical simulation do not consider chemical reaction. Does this indicate that the coupling between reaction and species transportation and dispersion is ignorable as well? That

is to say $E_i$ in Eq. 12 can be removed.

Section 3, general comments. Comparison between those three modules: AP-SteadyReactingFoam, APreactingFoam and APonlyChemReactingFoam would be very helpful.

Section 4, general comments/questions. Could you explicitly specify which one of the three (APonlyChemReactingFoam, APSteadyReactingFoam and APreactingFoam ) is used in case studies. Any discussions to compare APonlyChemReactingFoam with APSteadyReactingFoam/APreactingFoam.

General comments regarding the associated source code. "README" has detailed instructions for compilation, but does not specify any dependencies, such as third party libraries that needed? Any requirement in terms of the version of these dependencies? In addition, I would recommend adding detailed instructions about how to run APFoam, either in "README" or in a separate user manual. Imagining I am a fresh user of APFoam, I would need to know which executable to execute. Before executing that executable, what kind of preparation work is needed. For example, no need to instruct users how to generate mesh, but would be necessary to mention that users would need to get the meshed geometry ready before using APFoam. A little bit more Instructions about how to make use the "APFoam_tutorials" would also be very helpful.

[Figure]

**3 Technical corrections**

Abstract, line 9. Change "newly" to "new".

Abstract, line 13. Change "reaction" to "reactions".

In Eq. 1, please specify the meaning of $T$?

In Eq. 7, what is the $h$? Specific enthalpy?

Fig. 9, caption. "The probe points **locations**."

Line 307. "... in **the** targeted street canyon ."

Line 334. Change "obtain" to "obtained".

Line 414. "... **is** slightly greater than that of ..."

Line 418. Again "... **the** street canyon .."

Line 419. "This is because **that** the background "

---

## Referee Comment (RC2) · Anonymous Referee #2 · 5 Jan 2021

General comments:

Wu et al. present the development of an open-source CFD code based on OpenFOAM for Atmospheric Photolysis calculation to study the dispersion of reactive pollutants at the microscale. Full O3-NOx-VOCs chemistry has been implemented in the CFD model and compared with data from a box model simulation. Additionally, the accuracy of the model to predict the flow field and pollutant dispersion is evaluated in a 2D street canyon against wind tunnel measurements. This coupled system is applied to perform a comprehensive sensitivity test and examine the influence of the background precursors of O3, traffic emissions, and wind speed on pollutant concentrations in the

street.

I think this work provides valuable information on the field of urban air quality modeling at the microscale and I suggest carefully addressing the following comments before publication in GMD.

1.Although distinct photochemical mechanisms have been implemented in the model, this paper just present results from the full chemical mechanism "CS07A". Has the implementation of the rest of the photochemical schemes been properly evaluated? It should be mentioned in the manuscript, at least.

2.One of the limitations is that despite the model is fully coupled, the evaluation is performed separately (chemistry, flow, and dispersion of pollutants). It is probably because of the lack of measurements to validate this system; however, it should also be mentioned in the manuscript.

3. Conclusions. I would recommend improving this section and being more precise in giving the outcomes.

Specific comments: #Line 9. How can this development improve the resolution? #Line 13. The implementation of the atmospheric photochemical mechanism in the CFD model is evaluated with box model results. It should also be mentioned in the abstract. #Line 130. Similar to the reaction rates depending on T, might the photolysis rates be modified according to an input of the intensity of the light? #Line 151-156. It should be mentioned (here or in Section 3.1.) whether the implementation of these three photochemical mechanisms has been evaluated. #Line 213. Why is the simulation time set at 24h if no diurnal variation is considered? #Line 215. "Figure 2 shows the concentrations of 52 species..." Is that average concentration over 24h or concentration at a specific time? #Line 229. The concentrations are extremely low ($10^{-40}$ ppmV). I would recommend focusing on the comparison of CFD outputs and box model results just under realistic conditions since the largest differences occur when concentrations are almost zero and are mainly related to the different processing of these two models. #Line 254. "..., the prediction accuracy is better in simulating the low-wind-speed region". Please add a reference. #Line 296. The model acceptance criteria were previously defined in Chang and Hanna (2004) and Hanna and Chang (2012). Chang, J., Hanna, S., 2004. Air quality model performance evaluation. Meteorology and Atmospheric Physics 87 (1), 167-196. Hanna, S., Chang, J., 2012. Acceptance criteria for urban dispersion model evaluation. Meteorology and Atmospheric Physics 116 (3-4), 133-146. #Line 298. "...the respective NMSE, FB and R are 0.06, -0.13 and 0.95 (Table 1)...". These values do not correspond to the values presented in Table 1. #Line 302. Is any photochemical mechanism used in this simulation? If not, it should be clarified that this evaluation is performed with no chemical reactions included. #Line 325. Please use the same nomenclature for the aspect ratio (H/W). It is also referred to as H/W=1 in the abstract. #Line 331. Since the pollutant concentrations are presented in ppbv, could you also provide the emissions of NOx, VOCs and CO in ppbv s-1? #Line 351. Could you explain why that time step is selected for the chemistry? #Line 368. Please also provide the percentage of VOC reduction over the total VOC emissions (as shown in Table 3). #Line 385. I do not agree with the sentence "While on the windward side, NOx concentrations are more affected by the background conditions rather than emissions". NOx concentration on the windward side is more affected by the background conditions than that on the leeward side. However, the influence of the NOx emission on NOx concentration on the windward side is still larger than the background concentrations. Based on the results in Section 4.3., the influence of background concentrations on NOx concentration on the windward side is just around 10-20%. #Line 396. "...NO could be up to 90%" higher in the simple chemistry case. #Line 409-412. "...from the oxidation of background VOCs with OH will consume". Why is that oxidation only occur with background VOCs? Not sure how to distinguish the background VOCs from the emitted VOCs on the concentration in the street since pollutants are already well mixed and chemical reactions are non-linear. #Line 420-423. I think it occurs in the Base case as well. #Line 446. "In summary...". Please explain this better. #Line 468. Is average or total concentration in the street? Please

modify the caption in Fig. 15 as well. #Line 490. This paper presents results from the coupling of the chemical mechanism CS07A and CFD model. It should also be clarified in this section. #Line 494. "..., 2D and 3D pollutant dispersion...". The simulations are only performed in a 2D street canyon. #Line 497. Add aspect ratio. #Line 498. Please provide the VOC-to-NOx emission ratio used in these simulations. #Line 499. "Other numerical sensitivity cases,...". Please clarify what cases. #Line 503. Due to the non-linearity of chemical reactions, how is the contribution from the boundary conditions to O3 concentration computed? #Line 504. "Ventilation condition is another reason for the NOx concentrations increment, and the increase of NOx can be up to 98%". I think that is to be expected. In steady state (or quasi-steady state) conditions, the concentration of a non-reactive pollutant is double when wind speed is divided by 2 (if emissions do not change). Despite NOx is not truly a non-reactive pollutant, due to the influence of the VOC reactions with NO and NO2, it might be almost considered as non-reactive as the sum of NO and NO2.

Technical comments:

**Line 44. "material" instead of "materiel" #Line 297. Change "pervious" to "previous". #Line 371. Add "...change rate (CR_p)..." #Line 485. "polies" to "policies" #Line 392. Change "Figure 11 shows the changes rates of pollutant concentrations and NO to NO2 ratio..."**

Table 3. Add the name of the "full chemical mechanism" used in the simulations and mentioned in the manuscript (CS07A photochemical mechanism). Figure 11-14. Please use the defined CRp (Eq. 28) to show the change rate of each pollutant in %.

---

## Author Comment (AC1) · 10 Mar 2021

Thank you for the comments. Please see our responses in the attached supplement.

Please also note the supplement to this comment:
https://gmd.copernicus.org/preprints/gmd-2020-387/gmd-2020-387-AC1-supplement.pdf

---

## Author Response (AR1)

**Response to anonymous Referee #1**

We sincerely thank the reviewers for their constructive and thoughtful suggestions, which improve the quality of this paper. We have made the revisions and responses following your comments point by point.

The referee comments are shown in black.
The responses to the comments are shown in blue. The line numbers refer to the clean version of our revised manuscript.
The changes included in the revised manuscript are shown in red.

**1 General comments**

**General comment 1:**
This paper developed a calculation framework (APFoam-1.0), based on open-source CFD code OpenFOAM, for atmospheric photolysis to examine the micro-scale reactive pollutant formation and dispersion in the urban area. Five new types of reaction are added to the chemistry module, which is coupled with full O3–NOx–VOCs chemistry and CFD model. The model was validated against SAPRC box modeling software and wind tunnel experimental data. The framework was applied to case studies investigating O3–NOx–VOCs formation processes and dispersion of the reactive pollutants in an example of a typical street canyon. APFoam provides a numerical simulation tool based on the general purpose open source solver, which gives researchers the full capability to have control not only out of the box but also inside of the "box". So APFoam can be a useful tool of broad interest in atmosphere science.

However, the merits of APreactingFoam and APSteadyReactingFoam modules (two important modules of the APFoam-1.0) are not well demonstrated/articulated. Fully coupling (two way coupling) is adopted in both APreactingFoam and APSteadyReactingFoam, where the reaction heat is considered to have impact on fluid flow. Intuitively speaking, the concentration of pollutants is too low to have a significant impact on the fluid field. Whether the heat source from reaction is considered or not might NOT have any significant effect in terms of simulation accuracy: whether to consider or ignore such tiny effects won't change the simulation results too much. But it will have a significant effect on computational cost affecting the speed of simulations. Two way fully coupled model requires solving coupled governing equations with more unknown variables, which usually requires much more computational resources. In addition, the fully coupled system has more constraints on time steps and hence needs smaller time steps. The numerical algorithms become more complicated, too. As a general tool (calculation framework) for doing simulation, computational efficiency is important. Complicated model also makes results analysis more

complicated as there are more factors that need to be considered. In that sense, the simplest 3D module, APonlyChemReactingFoam, might be the best choice for most situations due to the following reasons:

• Faster in term of simulation and reduce computational cost significantly as explained above.

• Flow field data can be calculated offline and reused if the CFD set up does not change. For example, in certain case study, you keep the geometry, boundary condition and initial condition unchanged and only vary the source of pollutant (such as the locations, the pollutant release rate et al), then you only need to do the CFD simulation for once and reuse it for different case studies, this will significantly speed up your case study.

**Response:**

We are very thankful to reviewer for his/her constructive criticisms and valuable comments, which were of great help in improving the quality of the manuscript.

1. The APFoam is based on the OpenFOAM. In the original chemistry solver of OpenFOAM, the reaction heat source is considered in the solver. Actually, except the photochemical reaction, APFoam can also calculate other reaction problems which has greater reaction heat source. Therefore, the as one of the main characteristics of chemical reactions, we still keep this feature in the model.

2. The time step for this coupled system mainly follows the CFL condition from the previous studies (Bright et al., 2013; Garmory et al., 2009; Kim et al., 2012; Kwak et al., 2013; Sanchez et al., 2016; Zhong et al., 2017). The chemistry is solved by the ordinary deferential equation (ODE) solvers in OpenFOAM library, in which the chemical reactions can be integrated by dividing the flow time step into serval sub-time steps, automatically.

3. The simulations with chemical mechanism (using APreactingFoam or APSteadyReactingFoam) are initialized by the convergent flow fields (e.g., velocity and temperature, turbulence) simulated by the original flow solver in OpenFOAM. The flow fields keep almost unchanged during the simulation with chemistry. Furthermore, the simulation cost for flow fields equations (only 6 equations) is pretty small compared to the cost for the pollutant equations (about 52 equations in CS07A). Actually, only 11% of cpu time is consumed for the flow fields during each time step. Therefore, the present the fully coupled system does not affect the simulation efficiency of the solvers.

**General comment 2:**

Validation of the calculation framework is validated only using APonlyChemReactingFoam. APreactingFoam and APSteadyReactingFoam are not validated in the validation section. Not clear if APreactingFoam and APSteadyReactingFoam have ever been used in the case study section or not, the author did not explicitly mention that. Do these two modules (with much more complicated

governing equations) really have certain advantages in any situation? If so, I would encourage the author to justify it in a proper way, either based on literature review or ideally with a real case study. The authors need to show that considering and not considering the heat effect from reaction will have significant effect on simulation results for certain cases. With the above being said, It would also be necessary to compare the simulation speed of those three modules. In addition, in the validation section (section 3), the only validated 3D module APonlyChemReactingFoam is not validated in a fully coupled manner. The reaction model is validated alone (section 3.1). The CFD model is validated alone (section 3.2), too. And the pollutant species transportation and dispersion validation (section 3.3) is, at least, decoupled from reaction. The model is fully coupled, but the validation is done in a decoupled manner. Maybe the coupling between reaction and pollutants is also very weak and can be decoupled as well?

In summary, the models seem more complicated than necessary and lack sufficient validation. I would recommend accepting this manuscript to publish on GMD only after the major concern and specific comments (see next section) being properly addressed.

**Response:**

1. We had rechecked the 2D validation case and noted that the simulation case uses APreactingFoam. We apologize for the writing mistake in the manuscript. Generally, it should use the fully coupled solver (turbulence, pollutant dispersion and chemical reactions) to obtain the accurate results when considering the case with chemical reaction. Thus, in the original chemistry solver of OpenFOAM, the turbulence equation, pollutant transport equation and energy equation are coupled in the solver due to this reason. APonlyChemReactingFoam and APSteadyReactingFoam are the solvers which are developed based on the characteristics of atmospheric photochemical reactions (the chemical reactions are almost not affect the air flow) to save the simulation time (especially APonlyChemReactingFoam).

2. In all solvers of the APFoam, the reaction heat effect is considered. As mentioned above, APFoam is based on the OpenFOAM. In the original chemistry solver of OpenFOAM, the reaction heat source is considered in the solver. Actually, except the photochemical reaction, APFoam can also calculate other reaction problems which has greater reaction heat source. Therefore, the as one of the main characteristics of chemical reactions, we still keep this feature in the model.

3. We also compared the elapsed time between three solvers and found that the total elapsed time of APonlyChemReactingFoam is the longest. However, if the flow field has been determined and no need to recalculate, APonlyChemReactingFoam can save 11% of elapsed time compared with APreactingFoam while running the same setting case.

4. Due to the rarely wind tunnel experiments with chemical reactions, the model is only validated separately. We will continue to follow up the research on model accuracy in the future.

**2 Specific comments**

**Specific comment 1:**

Abstract, line 9. Numerical "resolution", in the context of mesh-based methods, such as the finite volume method used in the paper, depends on grid size, which is purely a choice in preprocess. Not clear to me how the framework developed in this paper can "improve the resolution". By "resolution", do you mean to say that the framework you developed targets at modeling small scale phenomena (such as street scale flow)?

**Response:**

Thanks a lot for pointing out this. The "improve the resolution" here means that the model can obtain the flow and pollutant dispersion in smaller scale. To make the expression clearer, we revised the sentence in the manuscript line 8-11:

Urban air quality issue is closely related to the human health and economic development. In order to investigate the street-scale flow and air quality, this study developed the Atmospheric Photolysis calculation framework (APFoam-1.0), an open-source CFD code based on OpenFOAM, which can be used to examine the micro-scale reactive pollutant formation and dispersion in the urban area.

**Specific comment 2:**

Abstract, line 14. The framework is also validated against SAPRC box modeling software, which is an essential validation, why only mention the wind tunnel validation in the abstract? Worthwhile to mention both validations.

**Response:**

Thanks a lot for the useful hint. The revision was done in the Abstract line 13-15:

Additionally, the model including photochemical mechanism (CS07A), air flow and pollutant dispersion has been validated and shows the good agreement with SAPRC modeling and wind tunnel experimental data, indicating that the APFoam has sufficient ability to study urban turbulence and pollutant dispersion characteristics.

**Specific comment 3:**

Abstract, general comments. Please double check the grammar in the abstract.

**Response:**

Thanks a lot for your mention. We had polished the grammar in the abstract and some grammatical

mistakes are corrected in the abstract.

**Specific comment 4:**

Section 1, general comments. It would be helpful for readers to have a better understanding on the major contribution of this paper if you have a little bit more detailed discussion regarding existing CFD based simulation studies in this area. For example, do they consider atmospheric photochemical in their simulation? Are those five reactions recently added in this paper already been studied in other research? In the existing studies, do they consider two way coupling or one way coupling? What tool do they use in their simulation study, OpnFoam or commercial softwares?

**Response:**

Thanks a lot for pointing out this. We have revied the introduction and added the information including photochemical mechanism, research parameter and CFD platform about the previous studies in Table 1. Many studies have not clearly pointed out whether their coupling method is one-way or two-way, so we have not sorted it out yet. The revision was done in the manuscript line 67-81:

With the rapid growth of the high-performance computing (HPC) platforms, computational power is no longer an obstacle. CFD simulation shows the good application prospect in urban microclimate research (Fernandez et al., 2020; Garcia-Gasulla et al., 2020). Many CFD models coupled with photochemical reaction mechanism have been developed to investigate the street-scale air quality problem in recent years (See Table 1). More commonly, simple photochemical mechanism with only three reactions (Leighton, 1961) is adapted in CFD models. This mechanism can simulate the NOx-O3 dispersion with a lower computational requirement.   Many previous studies have investigated the pivotal factors that affect the reactive pollutant distribution within the street canyon by using CFD model with simple photochemical mechanism, such as street-building aspect ratio (He et al., 2017; Zhang et al., 2020; Zhong et al., 2015), ambient wind conditions (Baker et al., 2004; Merah and Noureddine, 2019), thermal effects (Baik et al., 2007) or emissions from vehicle (Liu et al., 2018a; Zhang et al., 2019b). However, due to the simple photochemical mechanism ignoring the effect of other nitrogen oxides and VOCs on the photochemistry, some studies recently have applied the full photochemical mechanism in CFD models to reduce the uncertainty of pollutant simulation. Photochemical mechanisms contain $NO_x$-$O_3$-VOCs reactions and photochemistry, such as CBM-IV (Garmory et al., 2009; Kwak et al., 2013; Kwak and Baik, 2012, 2014), GEOS-Chem (Kim et al., 2012; Park et al., 2016), RCS (Bright et al., 2013; Zhong et al., 2017), and CCM (Sanchez et al., 2016) are successfully coupled with CFD models and applied to analyse the street-scale pollutant dispersion.

Table 1. Overview of the CFD studies with photochemical mechanism

| study | photochemical mechanism | Parameter | Platform |
|---|---|---|---|
| Baker et al., 2004 | simple | wind conditions | RAMS |
| Baik et al., 2007 | simple | thermal effects | Own code |
| Zhong et al., 2015 | simple | aspect ratio | OpenFOAM |
| He et al., 2017 | simple | aspect ratio | Fluent |
| Liu et al., 2018a | simple | emissions | Own code |
| Merah and Noureddine, 2019 | simple | wind conditions | Ansys-CFX |
| Zhang et al., 2019b | simple | emissions | Fluent |
| Zhang et al., 2020 | simple | aspect ratio | Fluent |
| Garmory et al. 2009 | CBM-IV | chemical mechanism | Fluent |
| Kim et al., 2012 | GEOS-Chem | emissions | Own code |
| Kwak and Baik, 2012 | CBM-IV | emissions | Own code |
| bright et al., 2013 | RCS | chemical mechanism | RAMS |
| Kwak et al., 2013 | CBM-IV | wind conditions | Own code |
| Kwak and Baik, 2014 | CBM-IV | thermal effects | Own code |
| Park et al., 2016 | GEOS-Chem | thermal effects | Own code |
| Sanchez et al., 2016 | CCM | chemical mechanism | STAR-CCM+ |
| Zhong et al., 2017 | RCS | chemical mechanism | OpenFOAM |

**Specific comment 5:**

Section 2, question. How do you determine the time step for this coupled system?

**Response:**

Thanks a lot for the useful hint. Generally, the time step for this coupled system follows the CFL condition. For reference, the time step of the simulation is between ~$10^{-3}$ to ~$10^{0}$ second from the previous studies (Bright et al., 2013; Garmory et al., 2009; Kim et al., 2012; Kwak et al., 2013; Sanchez et al., 2016; Zhong et al., 2017). The revision was done in the manuscript line 199-202:

Even so, the time step ($\Delta t$) generally follows the Courant–Friedrichs–Lewy (CFL) condition to maintain numerical stability, which is:

$$Co = \frac{U\Delta t}{\Delta x} \leq 1 \tag{13}$$

where $\Delta x$ is the grid size.

**Specific comment 6:**

Section 2.1, line 113. Missing" boundary condition", which is included in Fig. 1 but not mentioned in this paragraph.

**Response:**

Thanks a lot for your mention. The revision was done in the manuscript line 119-120:

For the simulation running (see Figure 1), mesh files, configure files, initial and boundary condition files should be prepared before the simulation.

**Specific comment 7:**

*T* in Eq.1 is supposed to be the temperature of the mixture? In Kelvin or Celsius?

**Response:**

Thanks a lot for pointing out this. *T* is the temperature of mixture in Kelvin. The revision was done in the manuscript line 135:

where *A*, *B* and *E* are the parameters of the reaction rates, and *T* is the temperature of mixture in Kelvin.

**Specific comment 8:**

In Eq. 2, Question. How are the lower and upper limit of the integration determined?

**Response:**

Thanks a lot for your attention. The lower and upper limit of the integration ($\lambda_1$ and $\lambda_2$) in Eq. 2 are the photolysis wavelength ranges according to the specific species. The revision was done in the manuscript line 139-141:

where $k_{\text{phot}}$ is the first order rate for the photolysis reaction; $\lambda_1$ and $\lambda_2$ are the photolysis wavelength ranges according to the specific species; $J(\lambda)$, *abs* $(\lambda)$ and $QY(\lambda)$ are the intensity of the light source, absorption cross section and the quantum yield for the reaction at wavelength $\lambda$, respectively.

**Specific comment 9:**

Eq. 1 - Eq. 5 use *k* to represent "reaction rates", while in Eq. 6, *w* is used as the reaction rate. Are those two reaction rates the same thing? If so, why using different symbols?

**Response:**

Thanks a lot for pointing out this. The *w* in Eq.6 is replaced by *k* to represent "reaction rates". The revision was done in the manuscript line 172:

$$\frac{\partial \rho Y_i}{\partial t} = k_i(Y_i, T) \qquad (6)$$

**Specific comment 10:**

In Eq. 8, should the average of molecular weight also be an unknown variable that needs to be calculated based on the mass fraction of each species? Imaging that chemical reaction changes the mass fraction of different species and then leads to changes of the average molecular weight. Any justification why the average molecular weight is assumed to be constant. Actually the assumption that average molecular weight is constant sounds reasonable to me, as the average molecular weight change caused by reaction might be ignorable. Then the following up question is, how significant is the reaction heat source? Seems can be ignorable as well. Authors may need to prove whether they are significant or not.

**Response:**

Thanks a lot for your mention. The average of molecular weight is a variable that can be calculated based on the mass fraction of each specie in each time step. As the reviewer said, the average molecular weight change of the photochemical reaction is too small and can be assumed as a constant. It should be noted that, this is not a constant variable because the mass fraction of each specie is changing during the simulation.

In all solvers of the APFoam, the reaction heat effect is considered. As mentioned above, APFoam is based on the OpenFOAM. In the original chemistry solver of OpenFOAM, the reaction heat source is considered in the solver. Actually, except the photochemical reaction, APFoam can also calculate other reaction problems which has greater reaction heat source. Therefore, the as one of the main characteristics of chemical reactions, we still keep this feature in the model. The revision was done in the manuscript line 176-181:

where $Y_i$ is the species mass fraction; $k_i$ is the reaction rate; $T$ is the temperature; $h$ is the specific enthalpy; $u_0$ is the initial energy; $p$ is the pressure; $\rho$ is the density; $\dot{q}$ is the heat from reaction; $R$ is the gas constant and $M_{ave}$ is the average molar weight which can be calculated based on the mass fraction of each species during the simulation.

**Specific comment 11:**

Section 2.3. For the governing equations (Eq. 6 - Eq. 8) of APChemForm, there are in total n+2 governing equations, I suppose the primitive unknowns in the governing equation are: $T, \rho, p, h$, Yi, there are in total n + 4 unknowns. number of equations < number of unknowns, Seems some other equations missing or not mentioned? Mathematically, the system is not closed. Or the author assumes that two of those $T, \rho, p, h$ are not unknown? Which two?

**Response:**

Thanks a lot for pointing out this. The governing equation has been added in the manuscript. Besides, in the model, one of the $\rho$ and $p$ can be set as a constant according to the needs of research (the pressure was set as constant during the present study). The other can be calculated by ideal gas

equation of state.

The expression of the governing equations is rewritten to make it clearer and the revision was done in the manuscript line 172-181:

$$\frac{\partial \rho Y_i}{\partial t} = k_i(Y_i, T) \tag{6}$$

$$h = u_0 + \frac{p}{\rho} + \int_0^t \frac{\dot{q}}{\rho} d\tau \tag{7}$$

$$h = \sum Y_i \left( \Delta h_{f,i}^0 + \int_{T_0}^T C_{p,i}(T') dT' \right) \tag{8}$$

$$p = \frac{\rho R T}{M_{ave}} = \sum p_i = \sum \frac{Y_i}{M_i} \rho R T \tag{9}$$

where $Y_i$ is the species mass fraction; $k_i$ is the reaction rate; $T$ is the temperature of the mixture; $h$ is the specific enthalpy; $u_0$ is the initial energy; $p$ is the pressure; $\rho$ is the density of the mixture; $\dot{q}$ is the heat from reaction; $\Delta h_{f,i}^0$ and $C_{p,i}$ are the enthalpy of formation at reference temperature $T_0$ and the constant-pressure specific heat (a function of temperature) of species $i$; $R$ is the gas constant and $M_{ave}$ is the average molar weight; $p_i$ and $M_i$ are partial pressure and the molar mass of species $i$. Besides, one of the $p$ and $\rho$ should be set as a constant for simulation according to the needs of research. The other is calculated by Eq. 9.

**Specific comment 12:**

Section 2, general comments.

- Symbols that are used in this paper only need to be specified for once, for example, $\rho$ is density, you only need to explain it the first time it appears

- Need to make sure that the same symbol has consistent meaning across the manuscript, make sure the same variable is only represented by one symbol.

- There are several dummy assumptions made in the model: such as, the mixture (mixture of air and pollutants, air itself is mixture) is in a thermal dynamic and dynamic quasi static state, they all share the same temperature and velocity. Maybe worthwhile to explicitly state the assumptions that you made when you establish the model.

- Would be better to explicitly specify that, $\rho$ is density of the mixture. Similar for $U$ and $T$.

**Response:**

Thanks a lot for the useful hint.

1. We had checked the symbols and removed the repeated description in the manuscript.

2. We had double checked and make sure that the same symbol has consistent meaning across the manuscript. The $w$ in Eq.6 is replaced by $k$ to represent "reaction rates".

3. Currently, the assumption that the mixture share the same temperature and velocity is a commonly used method in CFD model for reacting flows (Haworth, 2010).

4. The revision has done in the manuscript. All the variables have added detailed and necessary descriptions.

**Specific comment 13:**
Eq. 10 and Eq. 20, the physical viscosity is also considered. For air, physical viscosity is much smaller than turbulent viscosity and usually ignored in air flow simulation.

**Response:**
Thanks a lot for your mention. We choose to keep the physical viscosity term to ensure that the equation is complete.

**Specific comment 14:**
Eq. 20, $k$ shows up again here, what does $k$ represent in this equation, recall that $k$ was used as reaction rate in section 2. This is really confusing.

**Response:**
Thanks a lot for pointing out this. the $k$ in Eq. 20 is the turbulent kinetic energy in $k$-$\varepsilon$ model. As mentioned in the question below, Eq.18 – Eq.21 are essentially governing equations Eq. 13 - Eq. 15. We had removed the Eq.18 – Eq.21 in order to reduce repetition and make the manuscript clearer.

**Specific comment 15:**
Section 3.2. May worthwhile to explicitly state that the Eq. 18 - Eq. 21 are not a new set of governing equations, they are essentially governing equations Eq. 13 - Eq. 15 with turbulence model.

**Response:**
Thanks a lot for your mention. We had removed the Eq.18 – Eq.21 in order to reduce repetition and make the manuscript clearer.

**Specific comment 16:**
Section 3.2, line 256. Do you treat the air as incompressible flow in the simulation?

**Response:**
Thanks a lot for your attention. The air flow is treated as incompressible flow in the simulation. The revision was done in the manuscript line 296:

The air flow is assumed as incompressible steady-state turbulent flow in the simulation.

**Specific comment 17:**

Section 3. In the validation case, what is the Reynolds number for the CFD simulation set up? It is not clear to me why you choose to scale the geometry in CFD simulation? Why not use the same geometry as whatever in the experiment?

**Response:**

Thanks a lot for the useful hint. The *Re* number in full-scale flow CFD validation (*H/W* = 2.4, *H* = 24m) is about $2.14 \times 10^7$, and that in wind-tunnel-scale experiments (*H/W* = 2.4, *H* = 0.12 m) is $1.9 \times 10^5$. From the previous study, the critical of the Reynolds-number-independence is about $8.7 \times 10^4$ with the aspect ratio (*H/W*) of 2 (Chew et al., 2018; Yang et al., 2021). Thus, the *Re* number in both wind-tunnel-scale and full-scale models satisfy the requirement of Reynolds number independence. The normalized wind profiles with two scales can be compared for the validation purpose. Such validation technique has been adopted in the literature (Hang et al., 2020; Yang et al., 2021). The revision was done in the manuscript line 284-288:

The corresponding Reynolds number ($Re = \frac{U_{ref}H}{\upsilon}$) in full-scale flow CFD validation (*H/W* = 2.4, *H* = 24m) is about $2.14 \times 10^7$, and that in wind-tunnel-scale experiments (*H/W* = 2.4, *H* = 0.12 m) is $1.9 \times 10^5$, which satisfy the requirement of Reynolds number independence (the critical is about $8.7 \times 10^4$ with the *H/W* of 2) (Chew et al., 2018; Yang et al., 2021). The normalized wind profiles with two scales can be compared for the validation purpose. Such validation technique has been adopted in the literature (Hang et al., 2020; Yang et al., 2021).

**Specific comment 18:**
Section 3. Why no 3D simulation attempted? Any difficulties or due to computational cost?

**Response:**

Thanks a lot for your mention. The 3D simulation is added in section 3.4. The revision was done in the manuscript line 340-364:

3.4 Pollutant dispersion in 3D street canyon
As mentioned in section 3.3, 3D pollutant dispersion validation with tracer gas is conducted in this study, following the pervious study (Zhang et al., 2019b). Simulation results also compares with the wind tunnel experimental data (Chang and Meroney, 2001). CFD domain configuration is presented in Figure 9a. In this case, six buildings are set in the domain. Building height (H) and street canyon width (W) is 0.08 m with the *H/W* = 1. Building length (Lx) and building width (Ly) is 0.276 m and 0.184 m, respectively. The distance between buildings and domain inlet, side boundary, top boundary and domain outlet are respective 5*H*, 5*H*, 10*H* and 15*H*, for simulating a realistic results (Tominaga et al., 2008). Within the target street canyon, there are also 8 measurement points (4 of which on the leeward side and 4 on the windward side) for measuring the concentrations (Figure 9b). Besides, 6 more measurement points are also set on the top of the downstream building.

Pollutant concentrations at each measurement point in this simulation case are normalized with respect to the P5 ($C_i/C_5$) within the street canyon. The source of the $C_2H_6$ is set as an inlet at the bottom of the target street canyon. The size of the source is 0.005 m in width and 0.092 m in length setting in the middle of canyon. The release velocity is 0.01 m/s toward top boundary and the mass fraction of the $C_2H_6$ is 1 (pure gas of $C_2H_6$). For 3D pollutant dispersion simulation, APreactingFoam solver with Standard k-ε model is applied to solver compressible unsteady-state turbulent flow and pollutant dispersion as well. Photochemical mechanism is not used in the simulation. The minimum grid size in this case is 0.0005 m with expansion ratio of 1.1 from the wall surface toward surrounding. Time step of simulation is set as $1\times10^{-4}$ s and ODE solvers for chemistry is used in this validation case as well. Meanwhile, the inlet velocity and TKE profile are also retrieved from and fitted by the experimental data (Figure 10).

Figure 11 shows the comparison results between CFD simulation and experimental data. Overall, CFD simulation in 3D dispersion case slightly overestimates the concentrations in the street canyon. As for P23 and P24, the simulated results also overestimate, effected by the higher concentrations predicted within street canyon. Similarly, Statistical variables such as NMSE, FB and R are calculated to evaluate the performance of the model. As shown in Table 3, the value of NMSE, FB and R are 0.16, -0.21 and 0.93 in the 3D dispersion case, respectively, which agrees with acceptance criteria. In general, APFoam also shows the good performance in the 3D pollutant dispersion simulation.

[Figure]

**Figure 9. (a) The simulation domain of 3D pollutant dispersion and (b) the measurement points setting in the street canyon**

[Figure]

**Figure 10. The inlet profile of (a) stream-wise velocity and (b) turbulent kinetic energy in 3D dispersion case**

[Figure]

**Figure 11. Normalized concentrations of CFD and experimental data at each measurement point in 3D dispersion case**

**Specific comment 19:**

Section 3. The validation mainly validates the pollutant transportation and dispersion in the fluids field coupled with reaction, no reaction-flow-transportation coupling situation be tested. Namely, APSteadyReactingFoam, APreactingFoam are not tested. So do the authors also believe that the

effect of reaction to flow is too weak that no need to be overly concerned about it?

**Response:**

Thanks a lot for the hint. We had rechecked the 2D validation case and noted that the simulation case uses APreactingFoam. We apologize for the writing mistake in the manuscript.

Also the comparison among three solvers are carried and can be refer in section 4.2. The result of the comparison can refer to the response to the follow-up question.

The revision was done in the manuscript line 327-330:

For CFD simulation, APreactingFoam solver with the standard k-ε model is applied to solve the compressible unsteady-state turbulent flow field and pollutant dispersion. In order to be consistent with the wind tunnel experiments setting, photochemical mechanism is not used in the simulation.

**Specific comment 20:**

Section 3.3, general question. Could you clarify which of the following strategies is used in your validation?

- Solve the turbulent fluids governing equation (Eq. 18 - Eq. 21), pollutant species transportation equation (Eq. 12) and reactions (Eq.1 - Eq. 5) simultaneously.

- First do the fluids field simulation without considering pollutant species and r action, that is, first solve Eq. 18 - Eq. 21. Then do species transportation and dispersion together with reactions based on fluids field solution obtained in the first step? Namely, solve Eq. 12 and Eq. 1 - Eq. 5 in second step.

**Response:**

Thanks a lot for your attention. In the manuscript, turbulent fluids, chemical reaction and pollutant dispersion are solved simultaneously when the simulation cases use APreactingFoam. In the cases with APonlyChemReactingFoam, the solution of turbulent fluids governing equation is switch off, only chemical reaction and pollutant dispersion are solved during simulation. Summarily, the solution strategy is that all the equations are solve in the same time step. The revision was done in the manuscript line 186-187 and line 209-210:

For APreactingFoam, flow field, chemical reaction and pollutant dispersion are solved simultaneously in the same time step in this solver.

APonlyChemReactingFoam is only capable of solving the chemical reaction and species dispersion in the same time step under a certain flow field. The solution of turbulent fluids governing equation is switch off.

**Specific comment 21:**

Section 3.3, general comments. What is the time scale for reactions and what is the time scale for species transportation and dispersion? The motivation to ask this question is to see if there is possibility to decouple reaction from species transportation in such micro-scale simulation. In this section, both the experiment and the numerical simulation do not consider chemical reaction. Does this indicate that the coupling between reaction and species transportation and dispersion is ignorable as well? That is to say $E_i$ in Eq. 12 can be removed.

**Response:**

Thanks a lot for the useful hint. Actually, the chemical reactions and species transportation are solved decoupled during the solvers. During each flow time step (time step for transportation), the chemistry is solved by the ordinary deferential equation (ODE) solvers in OpenFOAM library, in which the chemical reactions can be integrated by dividing the flow time step into serval sub-time steps, automatically. Then, the mean reaction rates during the flow time step will be calculated and be used in the solving of transport equations. Currently, there are rarely wind tunnel experiments with chemical reactions. Thus, the 2D and 3D validation in this study are only carried with tracer gas. We are also curious about the couple and decouple problem about the species reaction and transportation. But in generally, both chemical reaction and transportation affect the pollutants distribution.

$E_i$ is the pollutant emitted from the source, it cannot be removed from the equation because there are emission source in validation cases. The revision was done in the manuscript line 331-333 and line 317-319:

Time step of simulation is set as $1 \times 10^{-4}$ s in this validation case. The chemistry is solved by the ordinary deferential equation (ODE) solvers in OpenFOAM library, in which the chemical reactions can be integrated by dividing the flow time step into serval sub-time steps, automaticall

Currently, there are rarely wind tunnel experiments with chemical reactions. Thus, the pollutant dispersion accuracy in 2D street canyon is validated by wind tunnel experimental data with tracer gas (Meroney et al., 1996), following the pervious study (He et al., 2017b; Zhang et al., 2020).

**Specific comment 22:**

Section 3, general comments. Comparison between those three modules: APSteadyReactingFoam, APreactingFoam and APonlyChemReactingFoam would be very helpful.

**Response:**

Thanks a lot for your mention. Comparison between APSteadyReactingFoam, APreactingFoam and APonlyChemReactingFoam has been carried in section 4.2 for the $H/W = 1$ street canyon case. The

result of the comparison can refer to the response to the next question.

**Specific comment 23:**

Section 4, general comments/questions. Could you explicitly specify which one of the three (APonlyChemReactingFoam, APSteadyReactingFoam and APreactingFoam) is used in case studies. Any discussions to compare APonlyChemReactingFoam with APSteadyReactingFoam/APreactingFoam.

**Response:**

Thanks a lot for your mention. Comparison between APSteadyReactingFoam, APreactingFoam and APonlyChemReactingFoam has been carried in section 4.2 for the *H/W* = 1 street canyon case. We have compared the flow field and the pollutant distribution for these three solver results. The comparison has shown that due to the different flow algorithm, the pollutant dispersion has slightly different in the street canyon. We also compared the elapsed time between three solvers and found that the total elapsed time of APonlyChemReactingFoam is the longest. However, if the flow field has been determined and no need to recalculate, APonlyChemReactingFoam can save 11% of elapsed time compared with APreactingFoam while running the same setting case. The revision was done in the manuscript line 417-441:

4.2 The comparison of pollutant distribution among the 3D CFD solvers

To investigate the difference of APonlyChemReactingFoam, APreactingFoam and APSteadyReactingFoam results, the comparisons of $O_3$, NO, $NO_2$ and CO distribution are conducted in *H/W* = 1 street canyon in this study. For APonlyChemReactingFoam, the flow field is treated as the incompressible steady-state flow and pre-solved using the SIMPLE method. The under-relaxation factors and residuals threshold for convergence are same as the setting in section 3.2. Chemical reaction and pollutant dispersion are solved under the steady-state flow for 90 minutes. For APreactingFoam and APSteadyReactingFoam case, turbulence flow, chemical reaction and pollutant dispersion are solved simultaneously for 90 minutes. The results in Figure 13 and all subsequent Figure are the pollutant dispersion at 90 minutes.

As depicted in Figure 13, the wind speed in APonlyChemReactingFoam case (Figure 13a) is lower than that in APreactingFoam (Figure 13b) and APSteadyReactingFoam (Figure 13c) case. The reason for the difference is most likely due to the different turbulence flow algorithm, where the turbulence is treated as incompressible steady flow, compressible unsteady flow and compressible steady flow in APonlyChemReactingFoam, APreactingFoam and APSteadyReactingFoam, respectively. Because of the slightly difference in wind speed, the concentrations of APonlyChemReactingFoam (Figure 13d, g, j, m) for pollutants are higher (due to the lower wind speed) than that in APreactingFoam (Figure 13e, h, k, n) and APSteadyReactingFoam (Figure 13f, i, l, o) case.

Table 5 shows the elapsed time of these three simulations in same case setting. Totally, the elapsed time of APonlyChemReactingFoam case (226 minutes) is slightly longer than that of APreactingFoam (214 minutes) and APSteadyReactingFoam (217 minutes) case while employing 192 CPU cores (16 × Intel® Xeon® E5-2692) for simulation. However, if the flow filed has been determined and no need to recalculate in the simulation case, the APonlyChemReactingFoam only takes 191 minutes for solving the chemical reaction and pollutant dispersion, which is 11% less time than APreactingFoam.

Many previous studies have treated the urban air turbulence as incompressible steady-state flow and investigate the pollutant dispersion successfully(He et al., 2017b; Ng and Chau, 2014; Zhang et al., 2019a, 2020, 2019b). With less time spending, the APonlyChemReactingFoam is applied in the study to analyse the photochemical reaction process in the street canyon.

[Figure]

[Figure]

Figure 13. The comparison of (a-c) wind speed, (d-f) $O_3$, (g-i) NO, (j-l) $NO_2$ and (m-o) CO between APonlyChemReactingFoam, APreactingFoam and APSteadyReactingFoam

Table 5. The elapsed time of three solvers

|  | APonlyChemReactingFoam | APreactingFoam | APSteadyReactingFoam |
|---|---|---|---|
| elapsed time (minute) | 191 + 35 (for turbulence) | 214 | 217 |

**Specific comment 24:**

General comments regarding the associated source code. "README" has detailed instructions for compilation, but does not specify any dependencies, such as third party libraries that needed? Any requirement in terms of the version of these dependencies? In addition, I would recommend adding detailed instructions about how to run APFoam, either in "README" or in a separate user manual. Imagining I am a fresh user of APFoam, I would need to know which executable to execute. Before executing that executable, what kind of preparation work is needed. For example, no need to instruct users how to generate mesh, but would be necessary to mention that users would need to get the meshed geometry ready before using APFoam. A little bit more Instructions about how to make use the "APFoam_tutorials" would also be very helpful.

**Response:**

Thanks a lot for pointing out this. Some details has been addad in README (on Github). Since the APFoam is developed based on the openFOAM, the install recommend is similar with the original openFOAM. The necessary libraries (minimum versions) mainly include: gcc (4.8.5); cmake (3.8); boost (1.48); zlib (1.2.7); openmpi (2.1.6) or mpich (3.2.1). Additionally, a tutorial of APFoam is add in Github as well.

**3 Technical corrections**

Abstract, line 9. Change "newly" to "new".

Abstract, line 13. Change "reaction" to "reactions".

**Response:**

Thanks a lot for pointing out these problems. The revision was done in the manuscript.

In Eq. 1, please specify the meaning of $T$?

**Response:**

Thanks for your attention. $T$ is the temperature of mixture. The revision was done in the manuscript:

where $A$, $B$ and $E$ are the parameters of the reaction rates, and $T$ is the temperature of mixture in Kelvin.

In Eq. 7, what is the $h$? Specific enthalpy?

**Response:**

Thanks a lot for pointing out this. $h$ is the specific enthalpy. The revision was done in the manuscript:

where $Y_i$ is the species mass fraction; $k_i$ is the reaction rate; $T$ is the temperature; $h$ is the specific enthalpy; $u_0$ is the initial energy; $p$ is the pressure; $\rho$ is the density; $\dot{q}$ is the heat from reaction; $R$ is the gas constant and $M_{ave}$ is the average molar weight.

Fig. 9, caption. "The probe points locations."

Line 307. "... in the targeted street canyon."

Line 334. Change "obtain" to "obtained".

Line 414. "... is slightly greater than that of ..."

Line 418. Again "... the street canyon .."

Line 419. "This is because that the background "

**Response:**

Thanks a lot for pointing out these problems. The revision was done in the manuscript.

References:

Bright, V. B., Bloss, W. J. and Cai, X.: Urban street canyons: Coupling dynamics, chemistry and within-canyon chemical processing of emissions, Atmos. Environ., 68, 127–142, doi:10.1016/j.atmosenv.2012.10.056, 2013.

Chew, L. W., Aliabadi, A. A. and Norford, L. K.: Flows across high aspect ratio street canyons: Reynolds number independence revisited, Environ. Fluid Mech., 18(5), 1275–1291, doi:10.1007/s10652-018-9601-0, 2018.

Garmory, A., Kim, I. S., Britter, R. E. and Mastorakos, E.: Simulations of the dispersion of reactive pollutants in a street canyon, considering different chemical mechanisms and micromixing, Atmos. Environ., 43(31), 4670–4680, doi:10.1016/j.atmosenv.2008.07.033, 2009.

Hang, J., Chen, X., Chen, G., Chen, T., Lin, Y., Luo, Z., Zhang, X. and Wang, Q.: The influence of aspect ratios and wall heating conditions on flow and passive pollutant exposure in 2D typical street canyons, Build. Environ., 168, 106536, doi:10.1016/j.buildenv.2019.106536, 2020.

Haworth, D. C.: Progress in probability density function methods for turbulent reacting flows, Prog. Energy Combust. Sci., 36(2), 168–259, doi:10.1016/j.pecs.2009.09.003, 2010.

Kim, M. J., Park, R. J. and Kim, J. J.: Urban air quality modeling with full O3-NOx-VOC chemistry: Implications for O3and PM air quality in a street canyon, Atmos. Environ., 47(2), 330–340, doi:10.1016/j.atmosenv.2011.10.059, 2012.

Kwak, K. H., Baik, J. J. and Lee, K. Y.: Dispersion and photochemical evolution of reactive pollutants in street canyons, Atmos. Environ., 70, 98–107, doi:10.1016/j.atmosenv.2013.01.010, 2013.

Sanchez, B., Santiago, J. L., Martilli, A., Palacios, M. and Kirchner, F.: CFD modeling of reactive pollutant dispersion in simplified urban configurations with different chemical mechanisms, Atmos. Chem. Phys., 16(18), 12143–12157, doi:10.5194/acp-16-12143-2016, 2016.

Yang, H., Lam, C. K. C., Lin, Y., Chen, L., Mattsson, M., Sandberg, M., Hayati, A., Claesson, L. and Hang, J.: Numerical investigations of Re-independence and influence of wall heating on flow characteristics and ventilation in full-scale 2D street canyons, Build. Environ., 189(October 2020), 107510, doi:10.1016/j.buildenv.2020.107510, 2021.

Zhong, J., Cai, X. M. and Bloss, W. J.: Large eddy simulation of reactive pollutants in a deep urban street canyon: Coupling dynamics with O3-NOx-VOC chemistry, Environ. Pollut., 224, 171–184, doi:10.1016/j.envpol.2017.01.076, 2017.

**Response to anonymous Referee #2**

We sincerely thank the reviewers for their constructive and thoughtful suggestions, which improve the quality of this paper. We have made the revisions and responses following your comments point by point.

The referee comments are shown in black.
The responses to the comments are shown in blue. The line numbers refer to the clean version of our revised manuscript.
The changes included in the revised manuscript are shown in red.

**General comments:**

Wu et al. present the development of an open-source CFD code based on OpenFOAM for Atmospheric Photolysis calculation to study the dispersion of reactive pollutants at the microscale. Full O3-NOx-VOCs chemistry has been implemented in the CFD model and compared with data from a box model simulation. Additionally, the accuracy of the model to predict the flow field and pollutant dispersion is evaluated in a 2D street canyon against wind tunnel measurements. This coupled system is applied to perform a comprehensive sensitivity test and examine the influence of the background precursors of O3, traffic emissions, and wind speed on pollutant concentrations in the street.

I think this work provides valuable information on the field of urban air quality modeling at the microscale and I suggest carefully addressing the following comments before publication in GMD.

1. Although distinct photochemical mechanisms have been implemented in the model, this paper just present results from the full chemical mechanism "CS07A". Has the implementation of the rest of the photochemical schemes been properly evaluated? It should be mentioned in the manuscript, at least.

2. One of the limitations is that despite the model is fully coupled, the evaluation is performed separately (chemistry, flow, and dispersion of pollutants). It is probably because of the lack of measurements to validate this system; however, it should also be mentioned in the manuscript.

3. Conclusions. I would recommend improving this section and being more precise in giving the outcomes.

**Response:**

We are very thankful to reviewer for his/her constructive criticisms and valuable comments,

which were of great help in improving the quality of the manuscript.

1. The validation of the full chemical mechanism is only conducted with CS07A. The detailed description has been added in section 2.2 and 3.1.

2. Due to the rarely wind tunnel experiments with chemical reactions, the model is only validated separately. Details have been added in section 3.3 and section 3.4. We will continue to follow up the research on model accuracy in the future.

**Specific comments:**

**Specific comment 1:**

**Line 9. How can this development improve the resolution?**

**Response:**

Thanks a lot for your suggestion. The "improve the resolution" here means that the model can obtain the flow and pollutant dispersion in smaller scale. To make the expression clearer, we revised the sentence in the manuscript line 9-11:

Urban air quality issue is closely related to the human health and economic development. In order to investigate the street-scale flow and air quality, this study developed the Atmospheric Photolysis calculation framework (APFoam-1.0), an open-source CFD code based on OpenFOAM, which can be used to examine the micro-scale reactive pollutant formation and dispersion in the urban area.

**Specific comment 2:**

**Line13. The implementation of the atmospheric photochemical mechanism in the CFD model is evaluated with box model results. It should also be mentioned in the abstract.**

**Response:**

Thanks a lot for your mention. The revision was done in the Abstract line 13-15:

Additionally, the model including photochemical mechanism (CS07A), air flow and pollutant dispersion has been validated and shows the good agreement with SAPRC modeling and wind tunnel experimental data, indicating that the APFoam has sufficient ability to study urban turbulence and pollutant dispersion characteristics.

**Specific comment 3:**

**Line 130. Similar to the reaction rates depending on *T*, might the photolysis rates be modified according to an input of the intensity of the light?**

**Response:**

Thanks a lot for pointing out this. In the current version of the APFoam, the photolysis rates are calculated offline. The variation of the intensity of the light is not including in the model so far. We will keep move on to update the model in the future. The revision was done in the manuscript in line 144-146:

the model does not consider the variation of light intensity, the photolysis rates are obtained from the literature (Carter, 2010) rather than online calculation in order to improve the calculation efficiency.

**Specific comment 4:**

**Line 151-156. It should be mentioned (here or in Section 3.1.) whether the implementation of these three photochemical mechanisms has been evaluated.**

**Response:**

Thanks a lot for the hint. The revision was done in the manuscript line 164-165:

In the section 3.1, CS07A has been validated while the other two mechanisms are not verified in this study but are still the available option to the users.

**Specific comment 5:**

**Line 213. Why is the simulation time set at 24h if no diurnal variation is considered?**

**Response:**

Thanks a lot for your attention. The main purpose of setting the simulation time to 24h is to allow the reactants to fully react and to verify the stability of the model during long-term operation.
The revision was done in the manuscript line 229-231:

For the chemical mechanism, CS07A is selected for validation in this study, and simulation time is set as 24h without diurnal variation (i.e., chemical reaction rate is constant during simulation),

allowing the reactants to fully react and verifying the stability of the model.

**Specific comment 6:**

**Line 215. "Figure 2 shows the concentrations of 52 species. . ." Is that average concentration over 24h or concentration at a specific time?**

**Response:**

Thanks a lot for pointing out this. The results are the concentrations at 24h which is the last time step of the simulation. The revision was done in the manuscript line 232:

Figure 2 shows the concentrations of 52 species from two models at 24h which is the last time step of the simulation.

**Specific comment 7:**

**Line 229. The concentrations are extremely low (10^-40 ppmV). I would recommend focusing on the comparison of CFD outputs and box model results just under realistic conditions since the largest differences occur when concentrations are almost zero and are mainly related to the different processing of these two models.**

**Response:**

Thanks a lot for the hint. In the realistic conditions, the species concentrations can reach $\sim10^{-20}$ ppmv (Figure 2 results). We have also compared those results and found that the *RE*s of those species are less that 1%, showing the good agreement of two models. The revision was done in the manuscript line 243-244:

Overall, most of the $RE_{i,t}$ are less than 1% in the concentrations range between 0 to 10-20 ppmv (i.e., the concentrations under realistic conditions), indicating that simulation error of APFoam is less than 1% during the whole simulation period.

**Specific comment 8:**

**Line 254. ". . ., the prediction accuracy is better in simulating the low-wind-speed region". Please add a reference.**

**Response:**

Thanks a lot for your attention. The revision was done in the manuscript line 269-273:

Among the RANS turbulence models, in contrast to the modified k-ε models (e.g., realizable and RNG k-ε models), although the standard k-ε model performs worse in predicting turbulence in the strong wind region of urban districts (e.g. separate flows near building corner), the prediction accuracy is better in simulating the low-wind-speed region (e.g. weak wind in 2D street canyon sheltered by buildings at both sides) (Tominaga and Stathopoulos, 2013; Yoshie et al., 2007).

**Specific comment 9:**

**Line 296. The model acceptance criteria were previously defined in Chang and Hanna (2004) and Hanna and Chang (2012). Chang, J., Hanna, S., 2004. Air quality model performance evaluation. Meteorology and Atmospheric Physics 87 (1), 167-196. Hanna, S., Chang, J., 2012. Acceptance criteria for urban dispersion model evaluation. Meteorology and Atmospheric Physics 116 (3-4), 133-146. #Line 298. ". . .the respective NMSE, FB and R are 0.06, -0.13 and 0.95 (Table 1). . .". These values do not correspond to the values presented in Table 1.**

**Response:**

Thanks a lot for pointing out this. The revision was done in the manuscript line 313-314:

In this simulation case, the respective NMSE, FB and R are 0.01, -0.04 and 0.99 (Table 2), which shows the good performance of the APFoam in flow field simulation.

**Specific comment 10:**

**Line 302. Is any photochemical mechanism used in this simulation? If not, it should be clarified that this evaluation is performed with no chemical reactions included.**

**Response:**

Thanks a lot for your mention. The photochemical mechanism is not used in this simulation in order to be consistent with the wind tunnel experiments setting. The revision was done in the manuscript line 329-330:

In order to be consistent with the wind tunnel experiments setting, photochemical mechanism is not used in the simulation.

**Specific comment 11:**

**Line325. Please use the same nomenclature for the aspect ratio (H/W). It is also referred to as H/W=1 in the abstract.**

**Response:**

Thanks a lot for the hint. The revision was done in the manuscript.

**Specific comment 12:**

**Line 331. Since the pollutant concentrations are presented in ppbv, could you also provide the emissions of NOx, VOCs and CO in ppbv s-1?**

**Response:**

Thanks a lot for pointing out this. The revision was done in the manuscript line 373-374:

In this study, the emissions of $NO_x$, VOCs and CO are $4.37 \times 10^{-8}$, $2.34 \times 10^{-8}$ and $2.03 \times 10^{-7}$ kg $m^{-3}$ $s^{-1}$ (i.e., ~35, ~200 and ~170 ppbv $s^{-1}$), respectively.

**Specific comment 13:**

**Line 351. Could you explain why that time step is selected for the chemistry?**

**Response:**

Thanks a lot for your attention. The timestep of the simulation follows the CFL condition from the previous studies (Bright et al., 2013; Garmory et al., 2009; Kim et al., 2012; Kwak et al., 2013; Sanchez et al., 2016; Zhong et al., 2017). We choose to use 0.1 s in this simulation. The chemistry is solved by the ordinary deferential equation (ODE) solvers in OpenFOAM library, in which the chemical reactions can be integrated by dividing the flow time step into serval sub-time steps, automatically. The revision was done in the manuscript line 199-202 and line 206-208:

Even so, the time step ($\Delta t$) generally follows the Courant–Friedrichs–Lewy (CFL) condition to maintain numerical stability, which is:

$$Co = \frac{U\Delta t}{\Delta x} \leq 1 \qquad (13)$$

where $\Delta x$ is the grid size.

The chemistry is solved by the ordinary deferential equation (ODE) solvers in OpenFOAM library, in which the chemical reactions can be integrated by dividing the flow time step into serval sub-

**Specific comment 14:**

**Line 368. Please also provide the percentage of VOC reduction over the total VOC emissions (as shown in Table 3).**

**Response:**

Thanks a lot for pointing out this. The revision was done in the manuscript line 408-411:

Case_EMIS_Ctrl_VOC20%, Case_EMIS_Ctrl_VOC30% and Case_EMIS_Ctrl_VOC40% are the scenarios which apply the stricter VOCs control measures (corresponding to 20%, 30% and 40% more VOCs emission reduction which are 60%, 65% and 70% reduction of total VOCs emission, respectively) on the vehicles with traffic control policies.

**Specific comment 15:**

**Line 385. I do not agree with the sentence "While on the windward side, NOx concentrations are more affected by the background conditions rather than emissions". NOx concentration on the windward side is more affected by the background conditions than that on the leeward side. However, the influence of the NOx emission on NOx concentration on the windward side is still larger than the background concentrations. Based on the results in Section 4.3., the influence of background concentrations on NOx concentration on the windward side is just around 10-20%.**

**Response:**

Thanks a lot for the hint. The revision was done in the manuscript line 453-455:

While on the windward side, $NO_x$ concentrations are less than that on the leeward side. This is because that the $NO_x$ from emission source first affects the leeward side which leads to the high concentrations in this area. As the wind flows, the concentrations of $NO_x$ gradually decrease due to the wind diffusion and dilution effect.

**Specific comment 16:**

**Line 396. "...NO could be up to 90%" higher in the simple chemistry case.**

**Response:**

Thanks a lot for pointing out this. The revision was done in the manuscript.

**Specific comment 17:**

**Line 409-412. ". . .from the oxidation of background VOCs with OH will consume". Why is that oxidation only occur with background VOCs? Not sure how to distinguish the background VOCs from the emitted VOCs on the concentration in the street since pollutants are already well mixed and chemical reactions are non-linear.**

**Response:**

Thanks a lot for your attention. The NO not only reacts with background VOCs but also emitted VOCs. The main reason why this sentence only concerning the background VOCs here is that this section analyzes the influence of background condition. The method to distinguish the VOCs source is by comparing the results from different zero out cases in this study. The method is presented in the manuscript line 394-398:

To investigate the effect of chemical mechanism, background condition of the precursors (BC), emission (EMIS) and wind condition (Uref) on the reactive pollutant concentrations in the street canyon, the cases of BC_zero_out, EMIS_zero_out and Uref0.5 are set up in numerical simulations. In the Case_BC_zero and Case_Emis_zero, the precursors of $O_3$ (i.e. NOx and VOCs) are removed from domain inlet (background boundary conditions) and pollutant source emissions, respectively, and then we compare the results with Base.

**Specific comment 18:**

**Line 420-423. I think it occurs in the Base case as well.**

**Response:**

Thanks a lot for your mention. This also occurs in the Base case. As mentioned in the above question, this section analyzes the influence of background condition. Therefore, the expression only focuses on the impact of background conditions.

**Specific comment 19:**

**Line 446. "In summary. . .". Please explain this better.**

**Response:**

Thanks a lot for the hint. Some details of the explanation are add. The revision was done in the manuscript line 514-518:

As mentioned above, $RO_2$ (the production of VOCs and OH) will cosume the NO and weaken the $O_3$ titration effect with NO. In Base case (Figure 17g), the reaction rate of $RO_2$ is negative, which means that $RO_2$ consumes the NO. However, in Case_Emis_zero, reaction rate of $RO_2$ is positive during the whole simulation period which means that there is not enough NO to react with $RO_2$ or even $O_3$ without the vehicular source. Therefore, the source emissions provide a large amount of NO which enhances the $O_3$ depletion in the street canyon.

**Specific comment 20:**

**Line 468.  Is average or total concentration in the street? Please modify the caption in Fig. 15 as well.**

**Response:**

Thanks a lot for your attention. The results are concentrations at 90 minutes. The revision was done in the manuscript line 539 (Figure 19 is Figure 15 in the original manuscript):

Figure 19 shows the concentrations of $O_3$, NO and $NO_2$ in different $NO_x$ and VOCs emission control scenarios at 90 minutes.

Figure 19. The (a) $O_3$, (b) NO and (c) $NO_2$ concentrations at 90 minutes in different emission control scenarios

**Specific comment 21:**

**Line 490. This paper presents results from the coupling of the chemical mechanism CS07A and CFD model. It should also be clarified in this section.**

**Response:**

Thanks a lot for the hint. The revision was done in the manuscript line 568-569:

By applying the APFoam with CS07A mechanism in the simulation of reactive pollutants in typical street canyon ($H/W = 1$), key factors of chemical processes are investigated.

**Specific comment 22:**

**Line 494. ". . ., 2D and 3D pollutant dispersion. . .". The simulations are only performed in a 2D street canyon.**

**Response:**

Thanks a lot for the mention. The 3D simulation is added in section 3.4. The revision was done in the manuscript line 340-364:

3.4 Pollutant dispersion in 3D street canyon

As mentioned in section 3.3, 3D pollutant dispersion validation with tracer gas is conducted in this study, following the pervious study (Zhang et al., 2019b). Simulation results also compares with the wind tunnel experimental data (Chang and Meroney, 2001). CFD domain configuration is presented in Figure 9a. In this case, six buildings are set in the domain. Building height (H) and street canyon width (W) is 0.08 m with the $H/W = 1$. Building length (Lx) and building width (Ly) is 0.276 m and 0.184 m, respectively. The distance between buildings and domain inlet, side boundary, top boundary and domain outlet are respective $5H$, $5H$, $10H$ and $15H$, for simulating a realistic results (Tominaga et al., 2008). Within the target street canyon, there are also 8 measurement points (4 of which on the leeward side and 4 on the windward side) for measuring the concentrations (Figure 9b). Besides, 6 more measurement points are also set on the top of the downstream building. Pollutant concentrations at each measurement point in this simulation case are normalized with respect to the P5 ($C_i/C_5$) within the street canyon. The source of the $C_2H_6$ is set as an inlet at the bottom of the target street canyon. The size of the source is 0.005 m in width and 0.092 m in length setting in the middle of canyon. The release velocity is 0.01 m s$^{-1}$ toward top boundary and the mass fraction of the $C_2H_6$ is 1 (pure gas of $C_2H_6$). For 3D pollutant dispersion simulation, APreactingFoam solver with Standard k-ε model is applied to solver compressible unsteady-state turbulent flow and pollutant dispersion as well. Photochemical mechanism is not used in the simulation. The minimum grid size in this case is 0.0005 m with expansion ratio of 1.1 from the wall surface toward surrounding. Time step of simulation is set as $1 \times 10^{-4}$ s and ODE solvers for chemistry is used in this validation case as well. Meanwhile, the inlet velocity and TKE profile are also retrieved from and fitted by the experimental data (Figure 10).

Figure 11 shows the comparison results between CFD simulation and experimental data. Overall, CFD simulation in 3D dispersion case slightly overestimates the concentrations in the street canyon. As for P23 and P24, the simulated results also overestimate, effected by the higher concentrations predicted within street canyon. Similarly, Statistical variables such as NMSE, FB and R are calculated to evaluate the performance of the model. As shown in Table 3, the value of NMSE, FB and R are 0.16, -0.21 and 0.93 in the 3D dispersion case, respectively, which agrees with acceptance criteria. In general, APFoam also shows the good performance in the 3D pollutant dispersion simulation.

[Figure]

**Figure 9. (a) The simulation domain of 3D pollutant dispersion and (b) the measurement points setting in the street canyon**

[Figure]

**Figure 10. The inlet profile of (a) stream-wise velocity and (b) turbulent kinetic energy in 3D dispersion case**

[Figure]

**Figure 11. Normalized concentrations of CFD and experimental data at each measurement point in 3D dispersion case**

**Specific comment 23:**

**Line 497. Add aspect ratio. #Line 498. Please provide the VOC-to-NOx emission ratio used in these simulations.**

**Response:**

Thanks a lot for pointing out this. The revision was done in the manuscript line 568-569:

By applying the APFoam with CS07A mechanism in the simulation of reactive pollutants in typical street canyon ($H/W$ = 1) with the VOCs to $NO_x$ emission ratio ~ 5.7 in ppbv $s^{-1}$, key factors of chemical processes are investigated.

**Specific comment 24:**

**Line 499. "Other numerical sensitivity cases,. . .". Please clarify what cases.**

**Response:**

Thanks a lot for the mention. The revision was done in the manuscript line 571-573:

Other numerical sensitivity cases (Case_BC_zero, Case_Emis_zero and Case_Uref50%) reveal that vehicle emission is the main source of the NO and NO2, with the contribution of 82%–98% and 75%–90%, respectively.

**Specific comment 25:**

**Line 503. Due to the non-linearity of chemical reactions, how is the contribution from the boundary conditions to O3 concentration computed?**

**Response:**

Thanks a lot for pointing out this. The method to computed the contribution from boundary condition to $O_3$ is by comarping the results from BC_zeros_out and Base case. The method is presented in the manuscript line 394-398:

To investigate the effect of chemical mechanism, background condition of the precursors (BC), emission (EMIS) and wind condition ($U_{ref}$) on the reactive pollutant concentrations in the street canyon, the cases of BC_zero_out, EMIS_zero_out and Uref0.5 are set up in numerical simulations. In the Case_BC_zero and Case_Emis_zero, the precursors of $O_3$ (i.e. NOx and VOCs) are removed from domain inlet (background boundary conditions) and pollutant source emissions, respectively, and then we compare the results with Base.

**Specific comment 26:**

**Line 504. "Ventilation condition is another reason for the NOx concentrations increment, and the increase of NOx can be up to 98%". I think that is to be expected. In steady state (or quasi-steady state) conditions, the concentration of a non-reactive pollutant is double when wind speed is divided by 2 (if emissions do not change). Despite NOx is not truly a non-reactive pollutant, due to the influence of the VOC reactions with NO and NO2, it might be almost considered as non-reactive as the sum of NO and NO2.**

**Response:**

Thanks a lot for your attention. You provide us an interesting topic to discuss and analyze. It is true that if no chemical reactions exist, the change rate of pollutant concentration is 100% (i.e., the concentrations double) when the wind speed is 50% smaller. However, due to the chemical reaction, the concentration change rate of NO, $NO_2$ or $NO_x$ (sum of No and $NO_2$) are different between leeward and windward side. This is because that the windward side is nearer to and locates downwind regions of pollutant source in the flow trace. Thus, it takes shorter time for wind transport pollutants from the source to windward side. The effect of the chemical reaction has not yet been reflected and the concentration change rate is nearer to 100% (~90-98%) in the windward side. Moreover, the windward side is in the downwind regions and farther to the pollutant source in the flow trace. It takes longer time for pollutant to reach the windward side and more reactions occur in this longer period. Therefore, the change rates in this region are smaller (~80-90%).

[Figure]

The change rate of NO, $NO_2$ and $NO_x$ between different ventilation conditions

**Technical comments:**

**Line 44. "material" instead of "materiel"**

**Line 297. Change "pervious" to "previous".**

**Line 371. Add "...change rate ($CR_p$). . ."**

**Line 485. "polies" to "policies"**

**Line 392. Change "Figure 11 shows the changes rates of pollutant concentrations and NO to $NO_2$**

ratio. . ."

**Response:**

Thanks a lot for pointing out these problems. The revision was done in the manuscript.

Table 3.    Add the name of the "full chemical mechanism" used in the simulations and mentioned in the manuscript (CS07A photochemical mechanism).

Figure 11-14. Please use the defined $CRp$ (Eq. 28) to show the change rate of each pollutant in %.

**Response:**

Thanks a lot for pointing out these problems. The revision was done in the manuscript.

References:

Bright, V. B., Bloss, W. J. and Cai, X.: Urban street canyons: Coupling dynamics, chemistry and within-canyon chemical processing of emissions, Atmos. Environ., 68, 127–142, doi:10.1016/j.atmosenv.2012.10.056, 2013.

Garmory, A., Kim, I. S., Britter, R. E. and Mastorakos, E.: Simulations of the dispersion of reactive pollutants in a street canyon, considering different chemical mechanisms and micromixing, Atmos. Environ., 43(31), 4670–4680, doi:10.1016/j.atmosenv.2008.07.033, 2009.

Kim, M. J., Park, R. J. and Kim, J. J.: Urban air quality modeling with full O3-NOx-VOC chemistry: Implications for O3and PM air quality in a street canyon, Atmos. Environ., 47(2), 330–340, doi:10.1016/j.atmosenv.2011.10.059, 2012.

Kwak, K. H., Baik, J. J. and Lee, K. Y.: Dispersion and photochemical evolution of reactive pollutants in street canyons, Atmos. Environ., 70, 98–107, doi:10.1016/j.atmosenv.2013.01.010, 2013.

Sanchez, B., Santiago, J. L., Martilli, A., Palacios, M. and Kirchner, F.: CFD modeling of reactive pollutant dispersion in simplified urban configurations with different chemical mechanisms, Atmos. Chem. Phys., 16(18), 12143–12157, doi:10.5194/acp-16-12143-2016, 2016.

Zhong, J., Cai, X. M. and Bloss, W. J.: Large eddy simulation of reactive pollutants in a deep urban street canyon: Coupling dynamics with O3-NOx-VOC chemistry, Environ. Pollut., 224, 171–184, doi:10.1016/j.envpol.2017.01.076, 2017.

---

## Referee Report (RR1)

**Review of manuscript GMD- 2020-387**

The revised manuscript has been improved, and the authors have addressed most of the comments I raised previously. This study demonstrates the capability of APFoam to simulate the dispersion of reactive pollutants in urban configurations. However, some inconsistencies need to be addressed before publication. For example, the validation of pollutant dispersion in 2D and 3D street canyon is performed using APreactingFoam (compressible flow/unsteady conditions) in Section 3.3 and 3.4, and the analysis in Section 4.3-4.7 is carried out using APonlyChemReactingFoam (incompressible fluid/steady conditions), while Section 4.2 shows that these two solvers provide different wind flow patterns and therefore, pollutant dispersion changes. Additionally, there are some typos/language inconsistencies in the revised version. I suggest revising the manuscript carefully to improve readability and clarity in describing the simulations and results.

**General comments**:

**Section 4.2.** Why is this comparison performed using different fluid properties for each solver? Has the flow reached the (quasi) steady-state conditions in the simulations using APreactingFoam and APsteadyReactingFoam after 90 minutes?

My suggestion is to use incompressible fluid for all cases to perform a proper comparison of these three solvers. After reaching the quasi-steady state, the wind pattern should approximately be the same using any of these solvers (if incompressible fluid is selected for all cases). The same applies for the dispersion of a non-reactive pollutant. Having the same wind pattern, it would make sense to compare the results from these solvers and provide the different computational time required to reach the quasi-steady state including chemical reactions as well. Based on this information (i.e. computational time require), users can select the appropriate solver for their simulations.

If the same fluid properties cannot be applied, then each solver would need to be validated independently.

**Conclusions**. Please be precise in giving the details of the methodology used in this study.

Line 565. Please add that the validations presented in this paper are using APFoam with CS07A.

Line 564-567. Please clarify that the validation of the photochemical mechanism (CS07A) is carried out against the *box* modelling SAPRC and flow and dispersion are performed against wind tunnel measurements.

Line 576. Please add in which conditions the NOx increases up to 98%, "...when wind speed is reduced to the half".

Line 578-581. I suggest the authors include the percentage reduction of NO, NO2 and O3 for the most relevant scenarios (e.g. NOx50% and VOC30%) since the outcomes from the analysis in Section 4.7 are interesting to be highlighted in this section.

**Specific comments:**

Line 14. Please add "with SAPRC *box* modelling"

Line 259. Please add units "...10^-5 to 10^-6"

Line 296. This sentence "The air flow... " is repeated (line 273-274).

Line 312. The acceptance criteria were originally defined in previous studies. Please see my previous comment and add the appropriate references.

Line 332-333. Please remove this sentence. This section presents the evaluation of dispersion of a tracer (non-reactive pollutant), and therefore, chemical reactions are not solved in this simulation.

Line 355. Please remove "ODE solvers for chemistry". The same applies here since chemical reactions are not solved in this simulation.

Line 394-400. Please use the same nomenclature for the simulated cases. For the same scenario, EMIS_zero_out is first used in Line 396 and Case_Emis_zero in the following line.

Line 432. Which is the case setting? Please clarify this.

Line 478. Based on what the authors stated in their responses to my previous comment. This sentence is therefore not correct. Please remove "background" since this statement applies to all VOC (background and emitted).

---

## Author Response (AR2)

* * *
**Response to anonymous Referee #2**

We sincerely thank the reviewers for their constructive and thoughtful suggestions, which improve the quality of this paper. We have made the revisions and responses following your comments point by point.

The referee comments are shown in black.

The responses to the comments are shown in blue. The line numbers refer to the clean version of our revised manuscript.

The changes included in the revised manuscript are shown in red.

**General comments:**

**General comment 1:**

**Section 4.2.** Why is this comparison performed using different fluid properties for each solver? Has the flow reached the (quasi) steady-state conditions in the simulations using APreactingFoam and APsteadyReactingFoam after 90 minutes?

My suggestion is to use incompressible fluid for all cases to perform a proper comparison of these three solvers. After reaching the quasi-steady state, the wind pattern should approximately be the same using any of these solvers (if incompressible fluid is selected for all cases). The same applies for the dispersion of a non-reactive pollutant. Having the same wind pattern, it would make sense to compare the results from these solvers and provide the different computational time required to reach the quasi-steady state including chemical reactions as well. Based on this information (i.e. computational time require), users can select the appropriate solver for their simulations. If the same fluid properties cannot be applied, then each solver would need to be validated independently.

**Response:**

Thanks a lot for these useful suggestions. The APreactingFoam and APsteadyReactingFoam simulations have reached the (quasi) steady-state conditions after 90 minutes simulation. The wind patterns between 60 to 90 minutes are in Figure 1. As the figure shown, after 60-minute simulation, a stable single vortex structure has been formed in the street canyon. And after 80 minutes, the wind pattens has stabilized and hardly changed, which indicated that the simulations have reached quasi-steady state.

As for the comparison of the solvers, you are right and that is true that our simulated flows are incompressible. Actually, the density is almost constant in our simulation results even the compressible model is used. The reason for using the compressible fluid in APreactingFoam, APsteadyReactingFoam simulation is because that we found the calculation results can easily become unstable and divergent when the chemistry and flow field are solved simultaneously under the incompressible fluid model. It is the limitation of Openfoam code which is widely known by its users. Therefore, we designed the APreactingFoam and APsteadyReactingFoam to only solve the compressible fluid and the comparison is conducted under these conditions.

Figure 1 Wind patterns in APreactingFoam and APsteadyReactingFoam simulation

| Time (minute) | APreactingFoam | APsteadyReactingFoam |
|---|---|---|
| At time of $t$=60 min |  |  |
| At time of $t$=65 min |  |  |
| At time of $t$=70 min |  |  |

[Figure]

Some details are added in the manuscript line 207-210 and line 218-219:

The APreactingFoam is only designed to solve compressible fluids because the simulation results are more likely to be unstable and divergent when the chemistry and flow field are solved simultaneously under the incompressible fluids. It is the limitation of OpenFOAM code which is widely known by its users.

This solver is only designed to solve compressible fluids for the same reason as APreactingFoam which is mentioned above.

**General comment 2:**

**Conclusions**. Please be precise in giving the details of the methodology used in this study.

Line 565. Please add that the validations presented in this paper are using APFoam with CS07A.

**Response:**

Thanks a lot for your attention. The revision was done in the manuscript line 563-565:

Additionally, to verify the model performance, several validations, including photochemical mechanism (CS07A) with SAPRC box modelling, flow field, 2D and 3D pollutant dispersion with wind tunnel experimental data have been conducted in this study.

Line 564-567. Please clarify that the validation of the photochemical mechanism (CS07A) is carried out against the *box* modelling SAPRC and flow and dispersion are performed against wind tunnel measurements.

**Response:**

Thanks a lot for your mention. The revision was done in the manuscript line 563-567:

Additionally, to verify the model performance, several validations, including photochemical mechanism (CS07A) with SAPRC box modelling, flow field, 2D and 3D pollutant dispersion with wind tunnel experimental data have been conducted in this study. The model results show a good agreement with the SAPRC box modelling and wind tunnel experimental data, indicating that the APFoam can be applied in the analysis of micro-scale urban pollutant dispersion.

Line 576. Please add in which conditions the NOx increases up to 98%, "…when wind speed is reduced to the half".

**Response:**

Thanks a lot for pointing out this. The revision was done in the manuscript line 576-579:

Ventilation condition is another reason for the $NO_x$ concentrations increment, and the increase of $NO_x$ can be up to 98% when the wind speed is reduced to the half. If no chemical reactions, $NO_x$ concentration should rise 100% when the wind velocity decreases 50% (i.e. ventilation capacity reduces 50%) since the Re-independence requirement is satisfied.

Line 578-581. I suggest the authors include the percentage reduction of NO, NO2 and O3 for the most relevant scenarios (e.g. NOx50% and VOC30%) since the outcomes from the analysis in Section 4.7 are interesting to be highlighted in this section.

**Response:**

Thanks a lot for your mention. The revision was done in the manuscript line 579-582:

However, our results indicate that at least another 30% reduction in vehicle VOCs emissions can reduce the $O_3$ concentrations under the odd-even license plate policy with 24%-32%, 25%-28% and -6%-2% reduction rates of NO, $NO_2$ and $O_3$, respectively.

**Specific comments:**

Line 14. Please add "with SAPRC box modelling"
Line 259. Please add units "…10^-5 to 10^-6"
Line 296. This sentence "The air flow… " is repeated (line 273-274).

**Response:**
Thanks a lot for pointing out this. The revision was done in the manuscript.

Line 312. The acceptance criteria were originally defined in previous studies. Please see my previous comment and add the appropriate references.

**Response:**

Thanks a lot for pointing out this. The reference Chang and Hanna (2005) was added in the manuscript.

Line 332-333. Please remove this sentence. This section presents the evaluation of dispersion of a tracer (non-reactive pollutant), and therefore, chemical reactions are not solved in this simulation.
Line 355. Please remove "ODE solvers for chemistry". The same applies here since chemical reactions are not solved in this simulation.

**Response:**

Thanks a lot for your attention. The revision was done in the manuscript.

Line 394-400. Please use the same nomenclature for the simulated cases. For the same scenario, EMIS_zero_out is first used in Line 396 and Case_Emis_zero in the following line.

**Response:**

Thanks a lot for pointing out this. The revision was done in the manuscript. All the 'EMIS' were revised to 'Emis'.

Line 432. Which is the case setting? Please clarify this.

**Response:**

Thanks a lot for pointing out this. This case setting here refers to the simulation domain and time setting. The revision was done in the manuscript line 432:

Table 6 shows the elapsed time of these three simulations in same $H/W = 1$ street canyon for 90 minutes simulation.

Line 478. Based on what the authors stated in their responses to my previous comment. This sentence is therefore not correct. Please remove "background" since this statement applies to all VOC (background and emitted).

**Response:**

Thanks a lot for the hint. The revision was done in the manuscript.

References:

Chang, J. C. and Hanna, S. R.: Technical Descriptions and User's Guide for the BOOT Statistical Model Evaluation Software Package. [online] Available from: http://www.harmo.org/Kit/Download/BOOT_UG.pdf, 2005.